# Microtubule end stabilisation by cooperative oligomers of Ska and Ndc80 complexes

Renjith M Radhakrishnan (ID)[1], Lauren Stokes (ID)[1], Matthew Day (ID)[1], Pim J Huis in 't Veld (ID)[2,3,4] & Vladimir A Volkov (ID)[1]✉

## Abstract

**During mitosis, properly aligned chromosomes stabilise microtubule ends with the help of kinetochores to ensure timely segregation of chromosomes. Microtubule-binding components of the human outer kinetochore, such as Ndc80 and Ska complexes, are present in multiple copies and together bind several microtubule ends, creating a highly multivalent binding interface. Whereas Ndc80:Ndc80 and Ndc80:microtubule binding is crucial for interface stability, Ndc80 alone in absence of Ska is unable to support stable kinetochore-attachments. Using cryo-electron tomography, we demonstrate that oligomeric Ndc80:Ska assemblies stabilise microtubule ends against shortening by strengthening lateral contacts between tubulin protofilaments at microtubule plus-ends. We further identify a point mutation within the SKA1 microtubule-binding domain that does not affect microtubule-binding of individual Ska molecules, but does abolish Ska:Ska interactions. Finally, we report that oligomerisation of Ska, in a cooperative fashion together with the Ndc80, is necessary to maintain stable microtubule attachments both in vivo and in vitro.**

**Keywords** Mitosis; Kinetochore; Microtubule; Ndc80; Ska
**Subject Categories** Cell Adhesion, Polarity & Cytoskeleton; Cell Cycle; Structural Biology

## Introduction

During mitosis, microtubules of the mitotic spindle bind to kinetochores, large multiprotein assemblies forming at the centromeric regions of chromosomes. Having established a bioriented configuration, kinetochores mainly interact with microtubule ends (Shrestha and Draviam, 2013). These end-on attachments persist over multiple cycles of microtubule shortening and growth, supporting kinetochore motility with the filament's end, and resisting detachment (Akiyoshi et al, 2010; Nicklas and Ward, 1994; Skibbens et al, 1993; Stumpff et al, 2008). Strength of

kinetochore's attachment to a microtubule end is hypothesised to be important to prevent accidental chromosome loss, however, an appropriate balance of strength needs to be achieved in order to correct erroneous attachments before they lead to chromosome abnormalities such as aneuploidy (DeLuca et al, 2006, 2011; Liu et al, 2009; Long et al, 2017; Thompson and Compton, 2011).

The kinetochore-microtubule binding interface is highly multivalent (Fig. 1A). The Ndc80 complex, the main microtubule-binder conserved across a majority of eukaryotes, in human cells is present at each kinetochore in hundreds of copies interacting with 5–15 microtubule ends (Kiewisz et al, 2022; Suzuki et al, 2015). Each copy of Ndc80 can bind microtubules via two distinct regions: the globular calponin-homology domains of the NDC80 and NUF2 subunits, and the unstructured N-terminal tail of the NDC80 subunit (Ciferri et al, 2008, 2005; Wei et al, 2005) (Fig. 1B). The microtubule-binding regions are separated from the kinetochore-binding RWD domains of the SPC24 and SPC25 subunits by a coiled coil-rich stalk of about 58 nm.

Multivalency of the Ndc80 complexes is crucial for their ability to support motility with the shortening microtubule ends in vitro (Powers et al, 2009; Volkov et al, 2018); kinetochores multimerise Ndc80 in several ways. First, Ndc80 is recruited to the kinetochore via two complementary binding pathways: CENP-C:Mis12:Ndc80 in a stoichiometry of 1:1:1 (Huis in 't Veld et al, 2016; Petrovic et al, 2010), a complex that forms the basis for the KNL1 recruitment and checkpoint signalling (Cheeseman et al, 2006; Polley et al, 2024; Yatskevich et al, 2024), and the CENP-T:Mis12:Ndc80 pathway with a stoichiometry of 1:1:3 in humans (Huis in 't Veld et al, 2016; Kim and Yu, 2015; Rago et al, 2015) and 1:1:2 in chicken cells (Takenoshita et al, 2022) and budding yeast (Pekgöz Altunkaya et al, 2016). Both CENP-C- and CENP-T-dependent pathways contribute to the resulting Ndc80 copy number per kinetochore approximately equally (Suzuki et al, 2015).

Second, in conditions of high occupancy of its binding sites on the microtubule, Ndc80 CH-domains interact with their neighbours, with a proposed bridging role of the unstructured N-terminal tail in cross-linking them (Alushin et al, 2012, 2010). Finally, neighbouring Ndc80s interact with each other in a cooperative manner via the loop region situated about 20 nm away from the microtubule-binding site (Polley et al, 2023). We previously showed that loop-dependent Ndc80:Ndc80 binding

[1]Centre for Molecular Cell Biology, School of Biological and Behavioural Sciences, Queen Mary University of London, London, UK. [2]Department of Mechanistic Cell Biology, Max Planck Institute of Molecular Physiology, Dortmund, Germany. [3]Max Perutz Labs, Vienna BioCenter, Vienna, Austria. [4]University of Vienna, Vienna, Austria.
✉E-mail: v.volkov@qmul.ac.uk

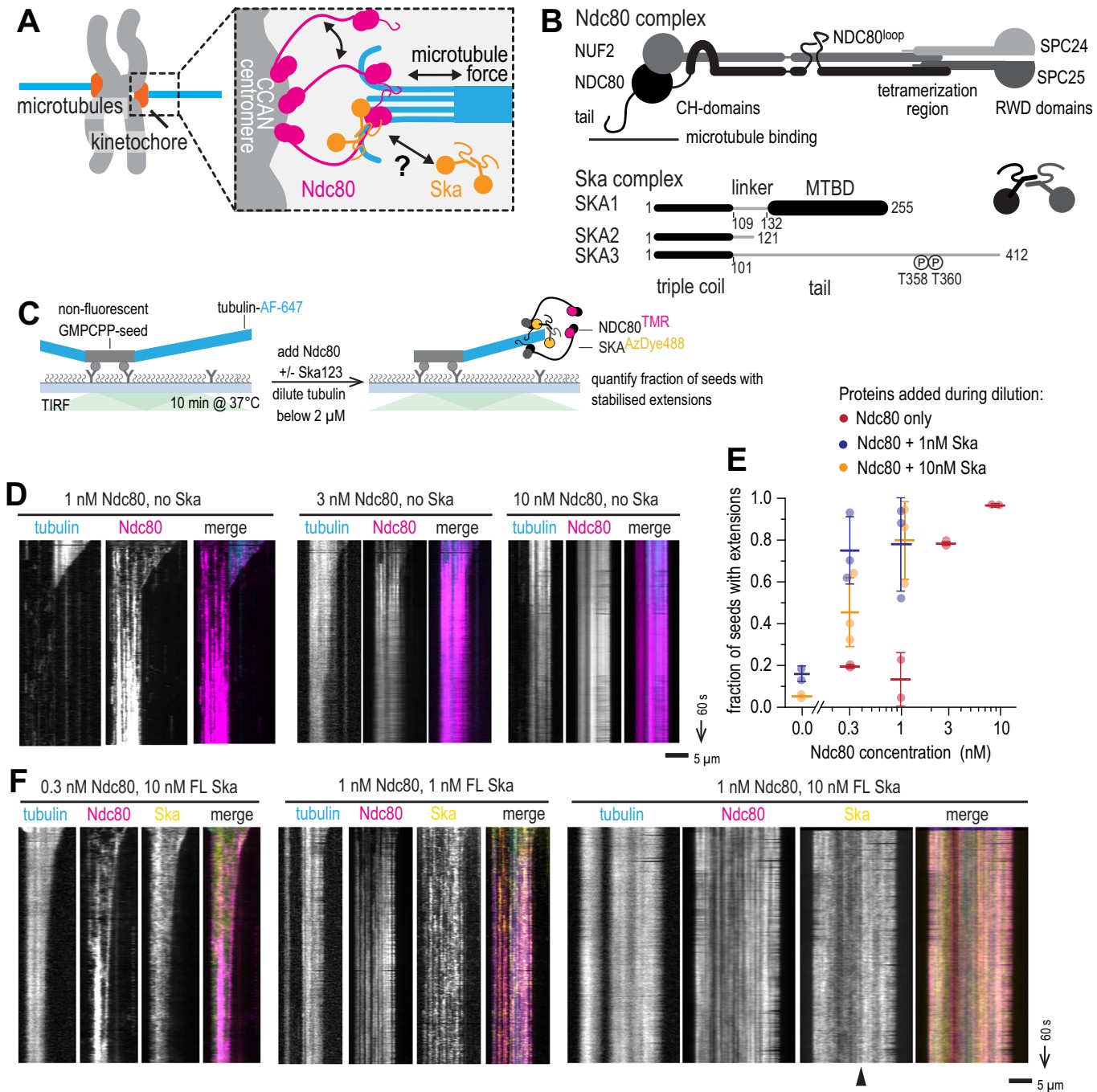

**Figure 1. Cooperative binding of Ndc80 and Ska prevents microtubule disassembly.**

(A) Schematic of the multivalent kinetochore-microtubule interface. (B) Domain organisation of human Ndc80 and Ska complexes. (C) Schematic of an experiment to test microtubule end stabilisation by Ndc80 and Ska. (D) Example kymographs showing microtubule length over time following the addition of Ndc80 at a concentration indicated. (E) Fraction of unlabelled GMPCPP-stabilised microtubule seeds with fluorescent tubulin extension following 10 min after tubulin dilution. Magenta: only Ndc80 added, blue: Ndc80 + 1 nM Ska, yellow: Ndc80 + 10 nM Ska (at least 50 seeds quantified in total over at least 5 fields of view per repeat, repeated 2–3 times). Lines and error bars represent mean ± SD. Two-way ANOVA: column factor (Ndc80 concentration) $p = 0.0002$; row factor (Ska concentration) $p = 0.0012$. (F) Example kymographs showing microtubule length over time following the simultaneous addition of Ndc80 and Ska at a concentration indicated. Arrowhead points to non-uniform microtubule decoration by Ska. Scale bars: 5 μm (horizontal), 60 s (vertical). Source data are available online for this figure.

was necessary to allow Ndc80 oligomers to rescue microtubule shortening under force, while absence of the loop region from cellular Ndc80, or point mutations in the loop, triggered the spindle assembly checkpoint (SAC) response and arrested the cells in prometaphase (Polley et al, 2023). Thus, the microtubule-binding activity of the Ndc80 is insufficient for its ability to bind and stabilise microtubule ends, and needs to be supplemented with allosteric oligomerisation.

In humans and other related animals, Ndc80 recruits an additional microtubule-binding complex called Ska. Human Ska complex consists of three polypeptides: SKA1, harbouring a C-terminal microtubule-binding domain (MTBD), SKA3 with an extended C-terminal tail that is required for Cdk1-dependent Ndc80 binding, and SKA2, which stabilises N-termini of all three subunits via formation of a triple coil in the form of a winged helix domain (Abad et al, 2014; Jeyaprakash et al, 2012; Zhang et al, 2017) (Fig. 1B). Recruitment of the Ska complex is one of the latest mitotic events, however, it is essential for proper mitotic progression, stability of kinetochore-microtubule attachments, and satisfaction of SAC (Auckland et al, 2017; Cheerambathur et al, 2017; Daum et al, 2009; Gaitanos et al, 2009; Hanisch et al, 2006; Raaijmakers et al, 2009; Sivakumar et al, 2016). While it is currently unclear how many copies of Ska can be recruited at the kinetochore, in vitro it binds Ndc80 in a stoichiometric manner (Huis in 't Veld et al, 2019) and thus is likely to be oligomerised, at least via the pathways described above leading to Ndc80 oligomerisation by its inner kinetochore receptors. Previous studies reporting Ska:Ndc80 interactions in vitro used either incomplete protein fragments (Schmidt et al, 2012), or full complexes either in homogenous micromolar mixtures (Huis in 't Veld et al, 2019), or oligomerised by beads for optical trapping (Helgeson et al, 2018; Huis in 't Veld et al, 2019). Thus it is currently unclear whether full-length Ska and Ndc80 can spontaneously oligomerise on microtubules, or they require oligomerisation imposed externally by microtubule lattices and inner kinetochore proteins.

In vitro, purified Ska was shown to follow both growing and shortening microtubule ends (Auckland et al, 2017; Maciejowski et al, 2017; Monda et al, 2017; Welburn et al, 2009), and to enhance end-tracking ability of Ndc80 with impaired intrinsic end-tracking: either non-end-tracking fragments (Schmidt et al, 2012), or tailless but otherwise full-length oligomers of Ndc80 (Huis in 't Veld et al, 2019). Microtubule end-tracking was attributed to the affinity of Ska to bent tubulin oligomers (Monda et al, 2017; Schmidt et al, 2012), conformations that are present at both growing and shortening microtubule ends (McIntosh et al, 2018). SKA1 MTBD was shown to carry multiple positively charged residues that contributed to its interaction with either growing or shortening microtubule ends, and to proper mitotic timing (Abad et al, 2014; Monda et al, 2017).

Despite the almost ubiquitous conservation of the Ndc80 complex in eukaryotes, its binding partners are divergent: while most animals have some versions of the Ska complex, Fungi lack it and instead cross-link their Ndc80 to microtubule ends with a ring-forming Dam1 complex (Miranda et al, 2005; van Hooff et al, 2017; Westermann et al, 2005). Dam1 and Ska share no sequence or domain conservation, however, they both are essential for cell viability and contribute to the mitotic fidelity through the same general mechanism: cross-linking the coiled coil of the Ndc80 complex to the tubulin flare at the end of a dynamic microtubule

(Helgeson et al, 2018; Huis in 't Veld et al, 2019; Lampert et al, 2010; Muir et al, 2023; Schmidt et al, 2012; Tien et al, 2010; Wimbish et al, 2020). Dam1 complex can assemble into ring-shaped oligomers thanks to interactions between its heterodeca-meric subunits (Jenni and Harrison, 2018; Miranda et al, 2005; Muir et al, 2023; Westermann et al, 2005), assisted by their binding to the Ndc80 complex, Bim1[EB1], and microtubules (Dudziak et al, 2021; Muir et al, 2023; Westermann et al, 2005). Mutations affecting Dam1 oligomerisation lead to a reduction in kinetochore-microtubule attachment strength, and severe mitotic phenotypes (Dudziak et al, 2021; Muir et al, 2023).

While Ska complex has not been shown to form microtubule-encircling rings, its functional similarity to the Dam1 complex has provided reasons to hypothesise that it should oligomerise as well (Jeyaprakash et al, 2012; Maciejowski et al, 2017; Welburn et al, 2009). Indeed, chemical cross-linking of the soluble Ska complex followed by mass-spectrometry have identified some regions within each of the three Ska subunits whose proximity might be sufficient for a direct interaction (Helgeson et al, 2018; Huis in 't Veld et al, 2019). However, it is unclear whether any of the previously identified molecular interfaces target specifically Ska:microtubule, or Ska:Ska interactions, or their combination.

In summary, the Ska:Ndc80:microtubule system is highly cooperative, with each component contributing to stabilising the other components: microtubules via their regular lattice structure, Ndc80 and Ska via their interactions with themselves, and with each other. The following properties appear to be essential but not individually sufficient for proper, stable kinetochore-microtubule attachments: microtubule-binding of Ndc80, cooperative oligomer-isation of Ndc80, microtubule-binding of Ska, Ska:Ndc80 interaction (Fig. 1A).

Given that Ndc80 oligomers can follow microtubule ends and stall and rescue them under force without Ska (Volkov et al, 2018), two important questions arise: first, how do Ndc80 oligomers affect microtubule end dynamics without any detectable affinity specifi-cally for microtubule ends (Schmidt et al, 2012)?; and, what is the mechanistic contribution of Ska to the microtubule end stabilisa-tion provided by oligomerised Ndc80? Here, we address both questions using a combination of biochemical reconstitution, electron cryo-tomography (cryoET), single-molecule microscopy, and cell biology. We report that Ndc80 oligomers promote lateral clustering of bent protofilaments at microtubule ends and thus stabilise microtubule plus ends against shortening, while Ska facilitates this process at lower Ndc80 concentration. We also show that Ska performs this function by oligomerising via at least two interfaces; oligomerisation of SKA1 MTBD can be disrupted by a point mutation and, while not necessary for individual molecules' microtubule binding, is indeed necessary to stabilise microtubule ends against disassembly both in vitro and in vivo.

# Results

## Cooperative binding of Ndc80 and Ska prevents microtubule disassembly

In order to dissect the specific effects of Ska:Ska and Ndc80:Ndc80 interactions within the outer kinetochore system, we sought to reconstitute the ability of Ska:Ndc80 oligomers to stabilise

microtubule ends in vitro in a minimal system of purified components. To this end, we purified full-length (FL) Ndc80 and Ska complexes, labelled them fluorescently (Fig. EV1A), and relied on their ability to form homo- and hetero-oligomers to stabilise microtubule ends. Microtubules were grown in a flow chamber from coverslip-attached stable seeds using fluorescently labelled tubulin, and then the solution was rapidly changed to the one containing low concentrations of either Ndc80, or Ska, or both, and tubulin in a concentration below 2 μM, to induce microtubule shortening (Fig. 1C). Using total internal reflection fluorescence microscopy (TIRF), we observed that Ndc80 alone was able to stabilise microtubule ends in concentrations above 3 nM; Ndc80 concentrations of 1 nM and below resulted in microtubule depolymerisation (Fig. 1D). To quantify the observed stabilising effect, we counted the fraction of seeds that carried fluorescent tubulin extensions after 10 min post-dilution, and found a sharp transition in the amount of stabilised seeds between 1 and 3 nM of added Ndc80 (Fig. 1E), consistent with the previously reported cooperative recruitment of Ndc80 to microtubules in the same concentration range (Polley et al, 2023).

Ska alone was unable to stabilise microtubules against shortening at concentrations as high as 10 nM (Fig. 1E). However, presence of 1–10 nM Ska in addition to low, non-stabilising amounts of Ndc80, resulted in stabilisation of microtubule ends and recruitment of Ska and Ndc80 oligomers to microtubule ends and lattices (Fig. 1E,F). At 10 nM Ska, we also observed higher density decoration of Ska towards the stabilised microtubule plus-end (Fig. 1F, arrowhead), which we will discuss in detail below. Two-way ANOVA confirmed that both the concentration of Ndc80, and the concentration of Ska contributed significantly to an increase in the fraction of fluorescent tubulin extensions (Fig. 1E).

Phosphorylation of SKA3 at T358 and T360 was previously shown to be essential for Ska:Ndc80 interaction in vitro (Huis in 't Veld et al, 2019) and kinetochore recruitment of Ska (Zhang et al, 2017), however, we previously demonstrated that Ska expressed in insect cells and purified without further modifications was partially phosphorylated allowing it to interact with Ndc80, unless specifically dephosphorylated using a phosphatase (Huis in 't Veld et al, 2019). To test whether the Ska:Ndc80 interaction in our microtubule end-stabilisation assay required additional phosphorylation on top of the background phosphorylation introduced during insect cell expression, we used purified Cdk1:CyclinB:CKS1 complex to phosphorylate purified FL Ska in vitro, resulting in a prominent shift in SKA3 migration on SDS-PAGE (Fig. EV1B). Repeating the microtubule end-stabilisation assay in presence of hyperphosphorylated or untreated FL Ska, we did not observe a statistically significant difference in the fraction of seeds carrying fluorescent microtubule extensions (2-way ANOVA: $p = 0.0003$ for the effect of the Ndc80; $p = 0.33$ for the effect of Ska hyperphosphorylation, Fig. EV1C). In all but one condition tested, the lengths of microtubules stabilised by Ndc80 with untreated Ska were similar or longer than those in presence of Ndc80 and hyperphosphorylated Ska (with the exception of 1 nM Ndc80 with 1 nM Ska, where microtubules were shorter in presence of untreated Ska, Fig. EV1D). On the other hand, Ska$^{SKA3\ T358/360A}$, a mutant that does not interact with Ndc80 in vitro (Huis in 't Veld et al, 2019), failed to stabilise microtubules against shortening in similar conditions (Fig. EV1C), which is also reflected in the

reduced length of few microtubules that were left in the chamber after dilution (Fig. EV1D). Both hyperphosphorylated and untreated Ska molecules were co-localising with Ndc80 on microtubule lattice (Fig. EV1E). Based on these observations, we pooled results with both hyperphosphorylated and untreated Ska.

## Ska:Ndc80 oligomers stabilise microtubule plus-ends by reinforcing lateral interactions between bent protofilaments

In order to gain a mechanistic understanding of microtubule end stabilisation by Ska and/or Ndc80 oligomers, we repeated the tubulin dilution experiments on grids suitable for cryo-electron tomography (cryoET). Microtubules were grown from stable seeds attached to the silanized and passivated holey SiO support film, and then diluted with Ska and/or Ndc80 containing solution (Fig. 2A). Examining the tomograms containing microtubule plus-ends in presence of non-stabilising (1 nM) and stabilising (3–10 nM) concentrations of Ndc80 alone, we observed oligomers of Ndc80, often forming long "trains" with their CH-domains and apparently intertwined with their C-terminal coiled coils, bound near the microtubule ends and along the lattices (Fig. 2B, Fig. EV2A). With 10 nM Ska present in addition to 1 nM Ndc80, we observed similar oligomers of Ndc80 (Fig. 2C), and, additionally, microtubule-bound oligomers that lacked the characteristic ordered "trains" of Ndc80 CH-domains with the elongated coiled-coils (Fig. EV2B,C). Although the resolution in individual denoised tomograms was not sufficient to confidently determine structural properties within these oligomers, we attribute these additional densities to the presence of Ska.

Focusing on ordered Ndc80 "trains" we picked particles for subtomogram averaging every 4.1 nm along the train in the direction from plus to minus end. Assuming the Ndc80 binds every tubulin monomer within a single protofilament (Alushin et al, 2010; 2012), we calculated the oligomerisation status of Ndc80 within these trains, and found that non-stabilising concentration of Ndc80 (1 nM) produced an average oligomer of $7 \pm 4$ Ndc80 copies (mean ± SD, $N = 26$, Fig. 2D). Both stabilising concentrations of Ndc80 produced a similar oligomer size: $17 \pm 9$ at 3 nM and $15 \pm 8$ at 10 nM ($N = 78$ and 409, respectively). Addition of 10 nM Ska to 1 nM of Ndc80 correlated with an increased Ndc80 oligomer size ($10 \pm 5$, $N = 114$, Fig. 2D). It thus appears that microtubule ends require a threshold Ndc80 oligomer size to get stabilised against shortening, which agrees with previous biophysical measurements (Volkov et al, 2018), as well as the available data on Ndc80 stoichiometry and microtubule occupancy at human kinetochores (Suzuki et al, 2015; Kiewisz et al, 2022; Dudka et al, 2018). Based on the similarity of results obtained with 3 and 10 nM Ndc80 in TIRF and cryoET experiments, we pooled them in further analyses.

We further performed subtomogram averaging to gain a better understanding of the mechanisms that allow Ndc80 and Ska oligomers to stabilise lateral contacts between tubulin protofilaments. Focusing on microtubule-bound "trains" of Ndc80 CH-domains, we obtained low-resolution subtomogram averages trains formed at 10 nM Ndc80 (Fig. 2E). We observed Ndc80 CH-domains occupying two neighbouring protofilaments within the microtubule wall. This observation was corroborated by inspection of non-averaged, denoised tomograms, where Ndc80 trains on adjacent protofilaments were clearly visible (Movies EV1–3).

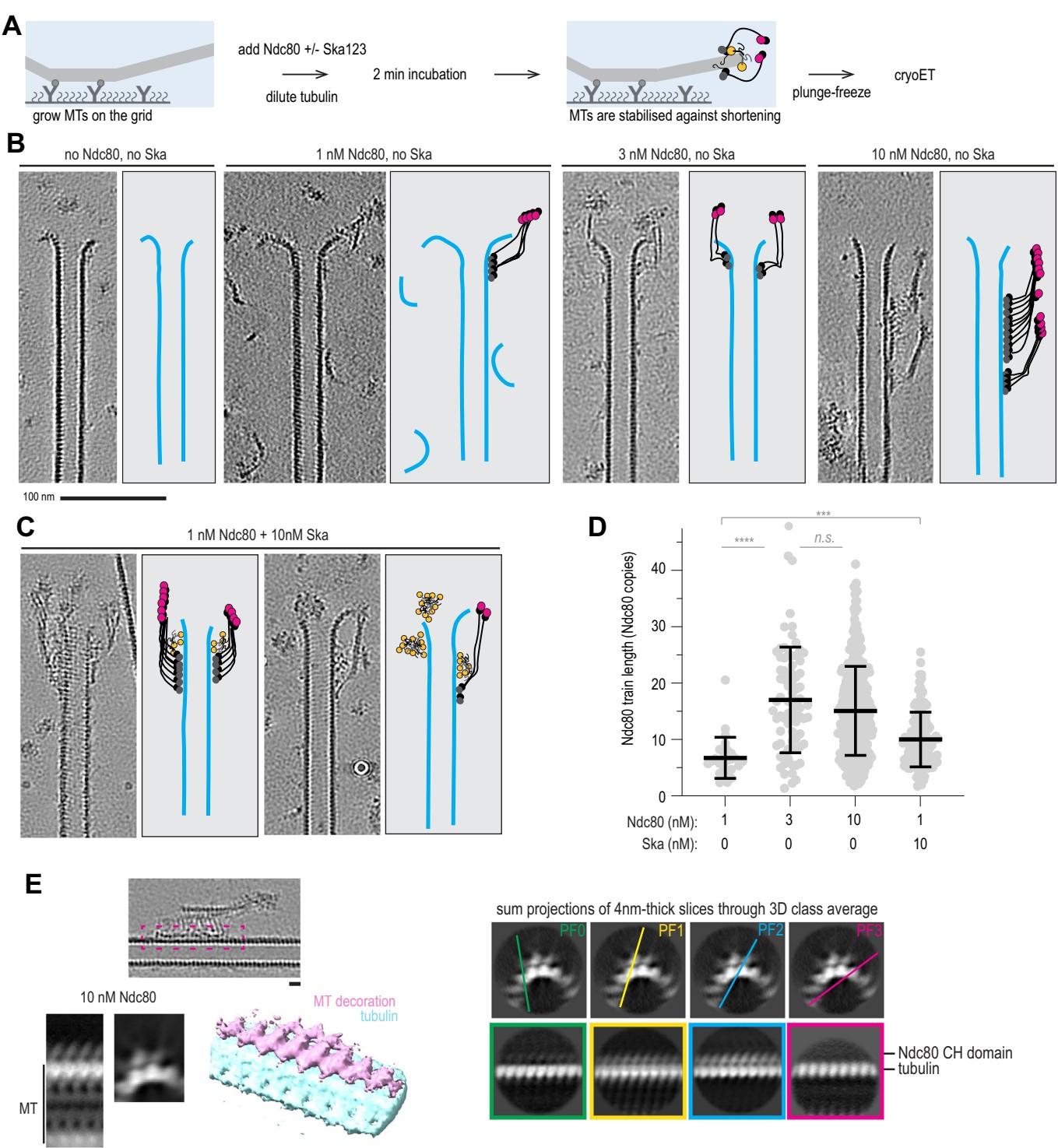

We also performed the same approach on "trains" of Ndc80 CH-domains in the samples containing 3 nM Ndc80, 1 nM Ndc80 and 10 nM Ska, obtaining similar averages (Fig. EV2D,E). However, when we repeated the same subtomogram averaging approach, picking particles from non-Ndc80 microtubule decoration in the sample containing both Ndc80 and Ska (Figs. 3B and EV2E), we did not observe any regular structures along the microtubule lattice in

our low-resolution reconstructions. Repeated rounds of 3D classification with these particles failed to yield separate classes.

How do Ndc80 and Ndc80/Ska oligomers prevent microtubules from shortening? We found a possible clue to this question by observing extended sheet-like protofilament structures stabilised at their plus-ends by Ndc80 oligomers (Figs. 3A and EV2AC). In these structures, several protofilaments appeared to be cross-linked

**Figure 2.   Characterisation of Ndc80 and Ska oligomers on microtubules using cryoET.**

(A) Schematic of an experiment to test microtubule end stabilisation by Ndc80 and Ska using cryoET. (B) Representative 0.8-nm-thick slices through cryoCARE-denoised tomograms showing microtubule end- and wall-bound oligomers of Ndc80 (greyscale), and their interpretation (colour) side by side. (C) Representative tomographic slices showing microtubule end- and wall-bound oligomers in presence of Ndc80 and Ska. (D) Length of Ndc80 trains in tomograms recorded in a condition indicated. Dots: individual values, lines: mean ± SD, N: 26 (1 nM Ndc80), 78 (3 nM Ndc80), 409 (10 nM Ndc80), 114 (1 nM Ndc80 + 10 nM Ska). Welch's t-test *p*-values: $2 \cdot 10^{-12}$ (****, 1 nM vs 3 nM), 0.09 (n.s., 3 nM vs 10 nM), 0.0004 (****, 1 nM no Ska vs 1 nM + 10 nM Ska). (E) Left: subtogram averaging of Ndc80 CH-domain trains at 10 nM Ndc80 – EMD-56085 (5766 particles obtained from 43 tomograms). White-on-black image represents a 2D projection of 3D classes (binned by 2 compared to the deposited average). Colour image shows 3D rendering of the same class. Right: sum projections through 4-nm-thick slices through the 3D class average. Top row, projection along the microtubule, with coloured lines showing corresponding projections across the microtubule shown below. Scale bars: 100 nm (B, C), 10 nm (E). Source data are available online for this figure.

laterally, an interaction that has previously been shown to distinguish shortening microtubule ends from growing ones (Kalutskii et al, 2025). We therefore set out to examine whether Ndc80 and Ska decoration of microtubule plus-ends correlates with protofilament clustering, and with stabilisation of these ends against disassembly.

To eliminate the possibility that we observed the ends of stable GMPCPP seeds, instead of GDP ends prevented from shortening by Ndc80 and Ndc80+Ska, we performed additional checks. First, we analysed the ends of those microtubules that were longer than the typical length of GMPCPP-seeds, as determined by total internal reflection fluorescence (TIRF) microscopy (Fig. EV3A). Second, building on previous observations that GMPCPP-stabilised microtubules have expanded lattices compared to GDP-bound microtubules (Manka and Moores, 2018; Zhang et al, 2015), we correlated the microtubule length to the tubulin lattice spacing near the plus ends and excluded plus ends of short expanded microtubule seeds from further analysis (Fig. EV3B). Finally, we compared the lattice spacing of microtubules shortening in absence of Ndc80, or in presence of non-stabilising Ndc80 concentration (1 nM) to microtubules stabilised by 3–10 nM Ndc80 or 1 nM Ndc80 with 10 nM Ska, and found no lattice expansion that would correlate with microtubule stabilisation (Fig. EV3C).

To get information about the shapes of microtubule plus-ends, we manually traced the 3D shapes of protofilaments at microtubule plus-ends in each of the four conditions: shortening microtubules (0 and 1 nM Ndc80), and microtubules stabilised by 3–10 nM Ndc80 alone, or by 1 nM Ndc80 with 10 nM Ska (Fig. 3B). We found that in both stabilising conditions, 3–10 nM Ndc80 alone and 1 nM Ndc80 + 10 nM Ska, microtubule plus-ends carried many more protofilaments without any bent segments than in a condition where microtubules were shortening in presence of 1 nM Ndc80 (Fig. EV3D). Increase in Ndc80 concentration from 1 nM to 3–10 nM, or addition of 10 nM Ska to 1 nM Ndc80 also correlated to a reduction in the average protofilament length (Fig. EV3E), and an increase in the raggedness of the end shape (Fig. EV3F), which reflects increased length of tapers stabilised by Ndc80 oligomers (Figs. EV2AC and 3AC).

We further tested if these bent protofilaments were interacting laterally with each other, i.e. whether they contacted neighbouring protofilaments along their length (Fig. 3C), or bent outwards without contacting their neighbours (Fig. 3D). We observed that an increase in Ndc80 concentration positively correlated with the amount of protofilaments found in a cluster, and with the appearance of large clusters containing ≥3 protofilaments (Fig. 3E). Two conditions where microtubule ends were stabilised against shortening, 3–10 nM Ndc80, and 1 nM Ndc80 with 10 nM Ska, did

not differ from each other in the distribution of protofilament clusters. On the other hand, we observed that presence of low, non-stabilising concentration of 1 nM Ndc80 already led to an increase in protofilament clustering, implying a direct effect of the end-tracking Ndc80 oligomers on the microtubule plus-end shape (Fig. 3E). Protofilaments were further clustered laterally upon an increase in the Ndc80 concentration from 1 to 3–10 nM, which coincided with microtubule stabilisation. Importantly, this effect did not depend on the threshold overlap between adjacent protofilaments: the difference between these two conditions remained significant, as judged by the Chi-squared test, within a range of overlap thresholds between 0.03 and 0.5 (Fig. EV3G). For the analysis of remaining datasets we chose the threshold of 0.1, which indicates that at least 50% of the weighted linear distances between protofilaments deviate by less than 20% (Kalutskii et al, 2025).

To further test whether there is a causal relationship between Ndc80 oligomers, and the appearance of protofilament clusters at microtubule plus-ends, we compared two ways in which Ndc80 oligomers bound to microtubules in the samples containing 3–10 nM Ndc80: within 50 nm from the protofilament flare, making possible a direct contact between Ndc80 oligomer and bent protofilaments, and farther than 50 nm from the end, making such contact impossible (Fig. 3F). We found that microtubules plus-ends carrying an Ndc80 oligomer directly at the end had fewer single protofilaments and more clusters with ≥3 protofilaments (Fig. 3E). We interpret these results, taken together, to conclude that Ndc80 oligomers stabilise microtubule ends by reinforcing lateral contacts between tubulin protofilaments, and Ska acts by lowering the Ndc80 concentration that is required to achieve this effect.

## Ska forms non-uniform coating on dynamic microtubules

What are molecular interactions that allow Ska to enrich Ndc80 oligomers on microtubules? We hypothesised that Ska could also possess cooperative binding interfaces; to dissect them, we repeated the sample preparation with dynamic microtubules in the presence of Ska, but without adding Ndc80. We observed multiple tubulin rings cross-linked to microtubules and to each other (Fig. EV4A), in agreement with a previous report (Monda et al, 2017). We also observed large microtubule end-bound oligomers which we interpreted as microtubule end-tracking Ska (Fig. EV4A). When we repeated the experiment in a flow-chamber suitable for TIRF microscopy, we observed that microtubule-bound Ska formed brighter regions near the growing microtubule ends, separated from the rest of Ska-decorated microtubule lattice by a clear stationary boundary that persisted until catastrophe (Fig. 4A). Inspired by analogous observations made with the microtubule-binding tau, we

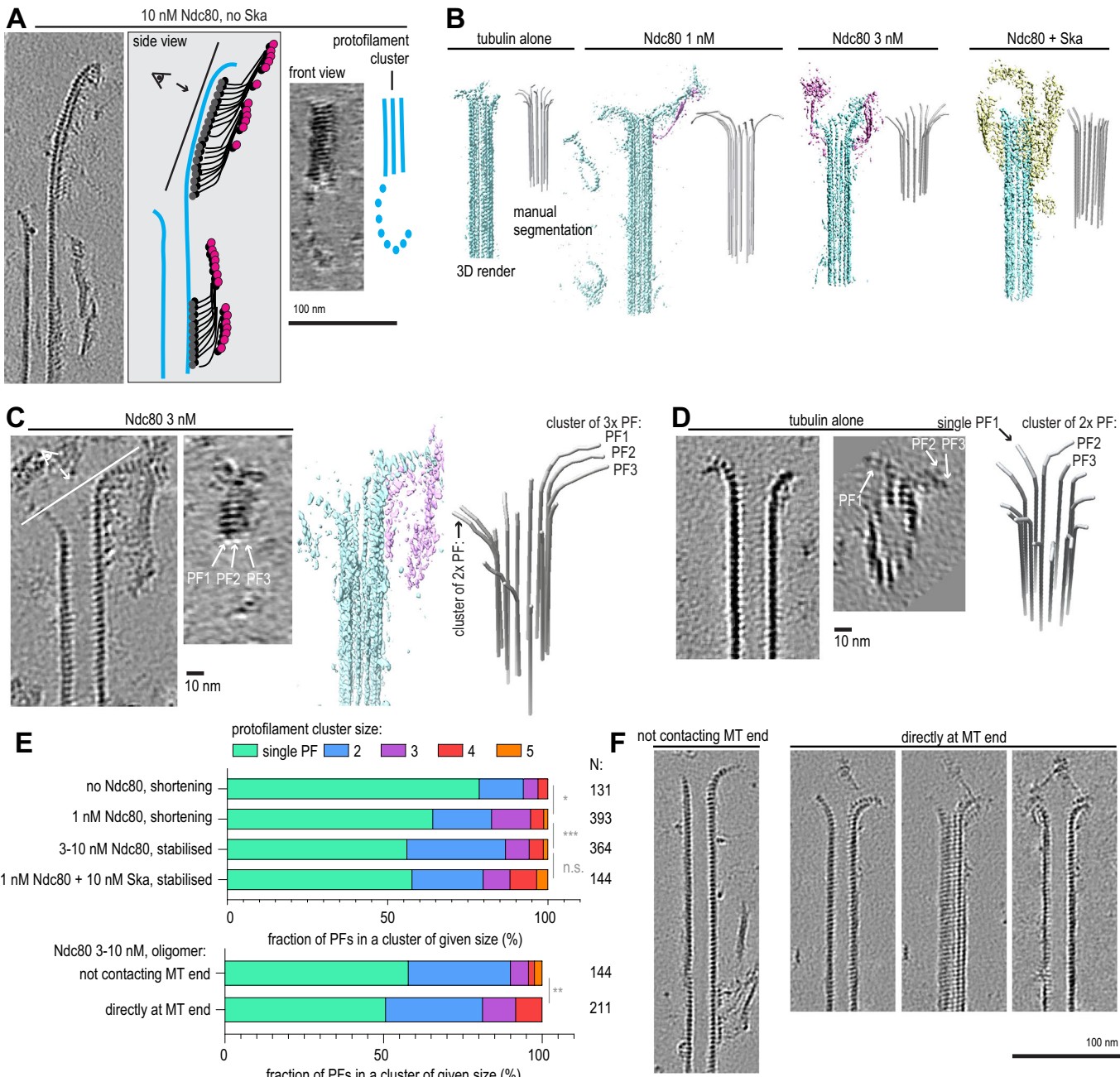

Figure 3. Ndc80 and Ska stabilise microtubule ends by promoting protofilament clustering.

(A) Example of an Ndc80 oligomer stabilising an extended, sheet-like protofilament cluster, viewed from the side (left), or from the luminal side of the cluster (right). (B) Examples of rendered 3D densities containing a microtubule plus-end, and the result of its manual segmentation to obtain the shapes of protofilaments for three conditions: undecorated shortening microtubules (15 plus-ends), microtubules shortening in presence of 1 nM Ndc80 (36 plus-ends), and microtubules stabilised against shortening using Ndc80 (41 plus ends) or Ndc80 + Ska (18 plus-ends). (C) Example of a microtubule end with 1 cluster of 2 protofilaments and 1 cluster of 3 protofilaments, viewed from the side, or from the luminal side of the 3-PF cluster, together with a 3D render and the result of the manual segmentation. (D) Same as (C), for a microtubule with only one cluster of 2 protofilaments. (E) Fraction of protofilaments engaged in laterally associated clusters, sorted by cluster size. Chi-squared test p-value: shortening with 0 vs 1 nM Ndc80: 0.0194; 1 nM vs 3–10 nM Ndc80: 0.001; 3–10 nM Ndc80 vs 1 nM Ndc80 + 10 nM Ska: 0.0905; Ndc80 not contacting plus-end vs directly at plus-end: 0.0054. (F) Additional examples of a microtubule carrying an Ndc80 oligomer farther than 50 nm from the plus-end (left); a microtubule with an Ndc80 oligomer directly contacting the protofilament flare, three individual 0.8-nm-thick slices. Scale bars: 100 nm (A, F), 10 nm (C, D). Source data are available online for this figure.

called these brighter regions "envelopes" (Siahaan et al, 2019; Siahaan et al, 2022). Non-uniform decoration of microtubules by other microtubule-associated proteins has been reported to result from cooperative interactions that stabilise oligomers of these proteins (Alushin et al, 2012, 2010; Polley et al, 2023; Tan et al, 2019; Pesenti et al, 2018; Maan et al, 2023). Following this logic, we are interpreting our results to hypothesise that non-uniform decoration of microtubules by Ska is a property arising from Ska:Ska interactions, which we dissect in detail below.

To identify domains of the Ska complex that mediate Ska:Ska interactions, we mixed a crowding agent (PEG) with purified Ska carrying deletions of either the SKA3 C-terminus, the SKA1 MTBD, or both (Fig. EV4B). Observing the resulting mixtures using fluorescence microscopy, we found round or irregular aggregates or droplets in most of the cases (Fig. EV4B,C). Ska$^{SKA3\Delta C}$ was the only construct displaying Ska:Ska interactions in absence of any additional crowding, however, the resulting structures were not spherical (Fig. EV4B,C). The only Ska construct that failed to form any self-interacting assemblies was Ska$^{SKA1\Delta MTBD\ SKA3\Delta C}$ that only retained the triple coil domains (Fig. EV4D). Thus, Ska can interact with itself via multiple regions.

We then repeated the envelope formation assay using Ska$^{SKA3\Delta C}$—the complex lacking the SKA3 tail altogether (Δ101–412), or the Ska$^{SKA3\ \Delta351-377}$—the complex lacking the peptide encompassing the region of SKA3 directly binding to the Ndc80 complex (Huis in 't Veld et al, 2019; preprint: Zhou et al, 2025) (Figs. 4AB and EV1B). We found that envelope formation was still present (Fig. 4C). We then probed the stability of these envelopes, and of their boundaries with the less densely decorated microtubule lattice, by recording fluorescence recovery after photobleaching (FRAP). We found that all three constructs supported formation of envelopes that exchanged with the soluble pool of Ska, and retained the intact boundary after FRAP (Fig. 4D). However, while Ska$^{SKA3\Delta C}$ envelopes recovered fully to pre-bleach levels, envelopes formed by FL Ska and Ska$^{SKA3\ \Delta351-377}$only recovered about 60% of their initial fluorescence (Fig. 4D). We thus conclude that envelope formation is likely mediated by an interaction mediated by SKA1 MTBDs, but envelopes that have already formed are then stabilised by regions of the SKA3 C-terminus distinct from its Ndc80-binding site.

Surprisingly, when we repeated this observation with monomeric SKA1 MTBD in isolation, we found that microtubule coating was mostly uniform with no clear boundary between microtubule plus-end proximal decoration and the rest of the microtubule lattice (Fig. 4E). Given that FL Ska is a dimer (Huis in 't Veld et al, 2019; Jeyaprakash et al, 2012; Welburn et al, 2009), we then dimerised the SKA1 MTBD using a short leucine zipper (LZ) motif from Gcn4, attached N-terminally to a construct containing SKA1 linker and MTBD (LZ-LMTBD, Fig. 4F and EV5A). Based on its migration on a size-exclusion column, we conclude that LZ-LMTBD is indeed a dimer (Fig. EV5B). Repeating the same envelope formation assay we found that LZ-LMTBD construct forms envelopes similarly to FL Ska and Ska$^{SKA3\Delta C}$ (Fig. 4F). We thus conclude that the minimal Ska construct that recapitulates the interaction of a FL Ska with a microtubule is a dimer of SKA1 MTBD.

Several lysine and arginine residues within the SKA1 MTBD were previously reported to reduce microtubule binding in vitro and extend or arrest mitotic progression in vivo (Abad et al, 2014; Monda et al, 2017). We introduced these mutations into our LZ-LMTBD and performed envelope formation assays with each of

these single or double mutants (K153A, R155A, K183/184A, K203/206A, K223/226A, R236A, and R245A, Figs. 4F,G and EV5A,C). We found that only the R236A mutation suppressed formation of non-uniform coating of SKA1 MTBD, while the other mutants formed envelopes, even when accompanied by an expected reduction in the overall density of microtubule-bound molecules (Fig. 4F,G). The R236A mutation also suppressed envelope formation in full-length SKA complexes, although we did observe Ska$^{SKA1\ R236A}$ accumulation at the shortening microtubule ends (Figs. 4F and EV5DE).

To quantify the observed effects on envelope formation, we measured the enrichment of Ska in the envelopes by calculating the ratio between the envelope fluorescence intensity, and the fluorescence intensity along the less bright region distal to the microtubule end (Fig. 4H). Using this method, we found that the only two constructs with a statistically significant reduction in the envelope/lattice intensity ratio were LZ-LMTBD R236A and Ska$^{SKA1\ R236A}$ (Fig. 4H).

To better understand the effect of the SKA1 R236A mutation on the microtubule binding of the FL Ska complex, we quantified dimensions of microtubule-bound oligomers we observed using cryoET (Fig. 5A). We found oligomers with similar dimensions in samples containing 400 nM FL Ska, and in samples in which Ndc80 (1 nM) and Ska (10 nM) were both present (Fig. 5A). However, in tomograms containing Ska$^{SKA1\ R236A}$-decorated microtubules, oligomers appeared much smaller (Fig. 5A). Using a previously established method of measuring overall dimensions of these oligomers extending in the direction perpendicular to the surface of the microtubule cylinder (Maan et al, 2023) (Fig. 5B), we found that the majority of them exceeded the expected dimensions of a single Ska dimer (Jeyaprakash et al, 2012; Janczyk et al, 2017) (Fig. 5C).

We next asked if presence of Ska$^{SKA1\ R236A}$ would interfere with the ability of Ska$^{SKA1\ wt}$ to form envelopes on microtubules. To test this, we mixed TMR-labelled Ska$^{SKA1\ R236A}$ with AzDye488-labelled Ska$^{SKA1\ wt}$ at various molar ratios in flow-chambers containing dynamic microtubules, and performed TIRF microscopy (Fig. 5D). Increasing the ratio of the R236A mutant from 10% to 50%, and keeping the total Ska concentration at the constant level of 100 nM, we observed a gradual decrease in the enrichment of the wt Ska in the envelopes (Fig. 5E,F). At 10% wt Ska mixed with 90% R236A, the envelope formation was completely abolished. Importantly, 10% wt Ska present in the chamber in absence of any R236A produced envelopes faithfully, suggesting that the observed lack of envelope formation was not caused by dilution of wt Ska (Fig. 5E,F).

We thus conclude that Ska:Ska self-interactions on dynamic microtubules that lead to non-uniform coating of microtubules require dimerization of SKA1 MTBD, and can be disrupted by a point mutation R236A. Based on various assays presented above we interpret the effects of the R236A mutation as reducing Ska oligomerisation. We then performed a series of experiments to characterise the effects of the oligomerisation-deficient Ska carrying the R236A on its microtubule binding in vitro and in vivo.

## Oligomerisation-deficient Ska has intact microtubule binding in single-molecule conditions

We first tested the microtubule-binding properties of the mutants we used above, in conditions that would prevent them from forming cooperative Ska:Ska interactions. Using taxol-stabilised

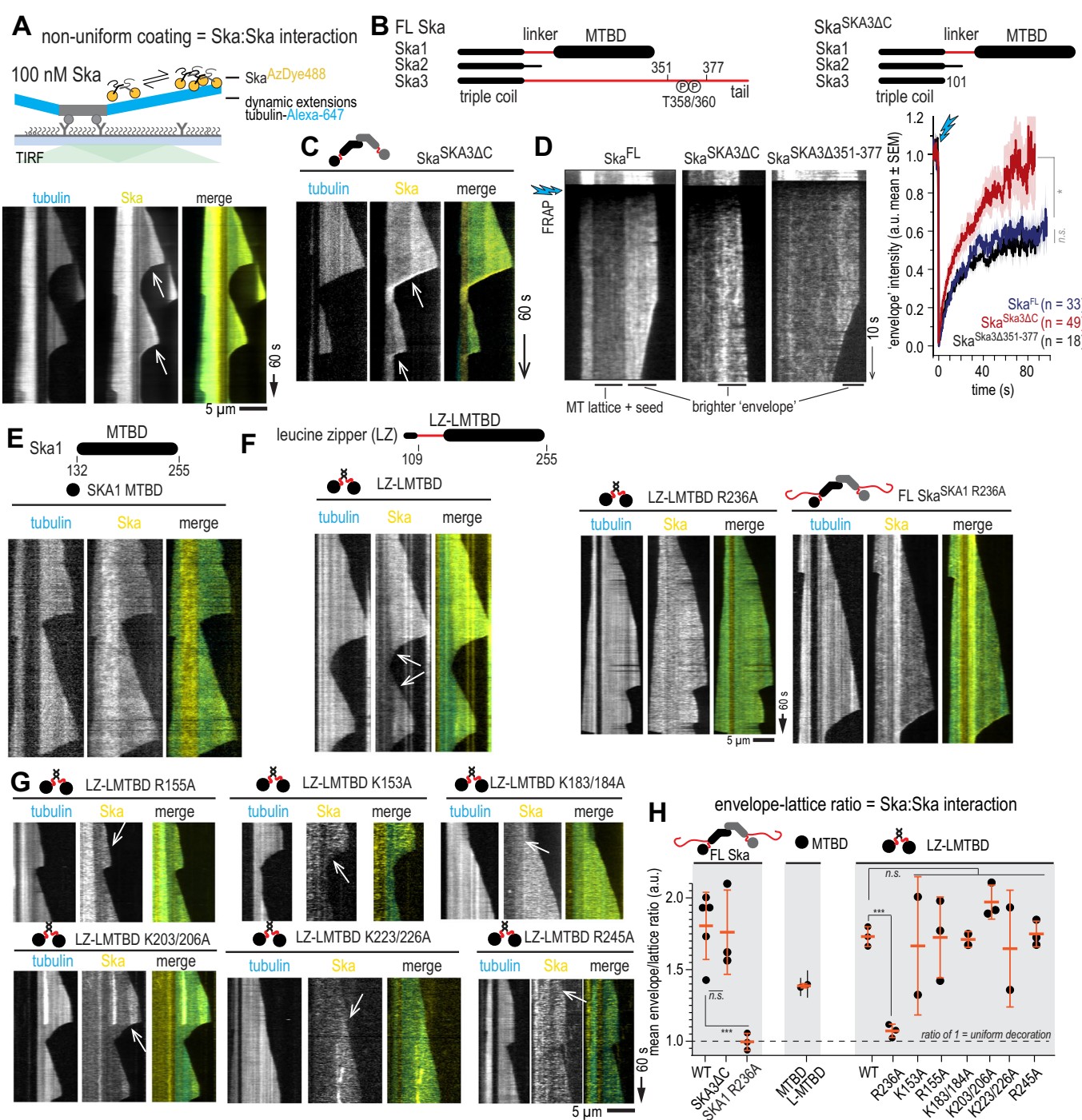

**Figure 4. Ska forms non-uniform coating on dynamic microtubules.**

(A) Schematic of the assay to assess Ska:Ska interaction on microtubules using its ability to form brighter envelopes as a readout. (B) Schematic representations of Ska123 constructs used in the study. Black represents folded domains, red represents disordered regions. (C) A kymograph showing envelope formation by Ska^SKA3 ΔC. White arrows show the persistent boundary between a brighter Ska envelope proximal to the growing microtubule end, and the rest of less brightly decorated microtubule lattice. (D) Example of a kymograph showing FRAP experiments with microtubules decorated with a construct indicated (left). The graph on the right shows mean envelope intensity (solid line) and SEM (shaded area). Welch's t-test p-values: FL Ska vs Ska^SKA3 ΔC: 0.0248 (*); FL Ska vs Ska^SKA3Δ351-377: 0.2238 (n.s.). (E) Monomeric SKA1 MTBD does not form a clear boundary between a brighter envelope and the rest of the microtubule lattice, as opposed to the dimeric LZ-LMTBD construct (F). (F) Example kymographs of LZ-LMTBD wt, LZ-LMTBD R236A, and FL Ska^SKA1 R236A. (G) Example kymographs of LZ-LMTBD mutants decorating dynamic microtubules. (H) Quantification of the extent of the envelope formation, showing mean envelope/lattice ratio of fluorescence intensity ± SD of repeated experiments. Welch's t-test: FL Ska^SKA1 wt vs FL Ska^SKA1 R236A p = 0.0009 (***); LZ-LMTBD wt vs LZ-LMTBD R236A p = 0.0003 (***); FL vs LZ-LMTBD p = 0.53; FL Ska^SKA1 R236A vs LZ-LMTBD R236A p = 0.158; 1-way ANOVA of LZ-LMTBD wt vs non-R236A mutants p = 0.97. Scale bar: 5 μm (horizontal), 60 s (vertical). Source data are available online for this figure.

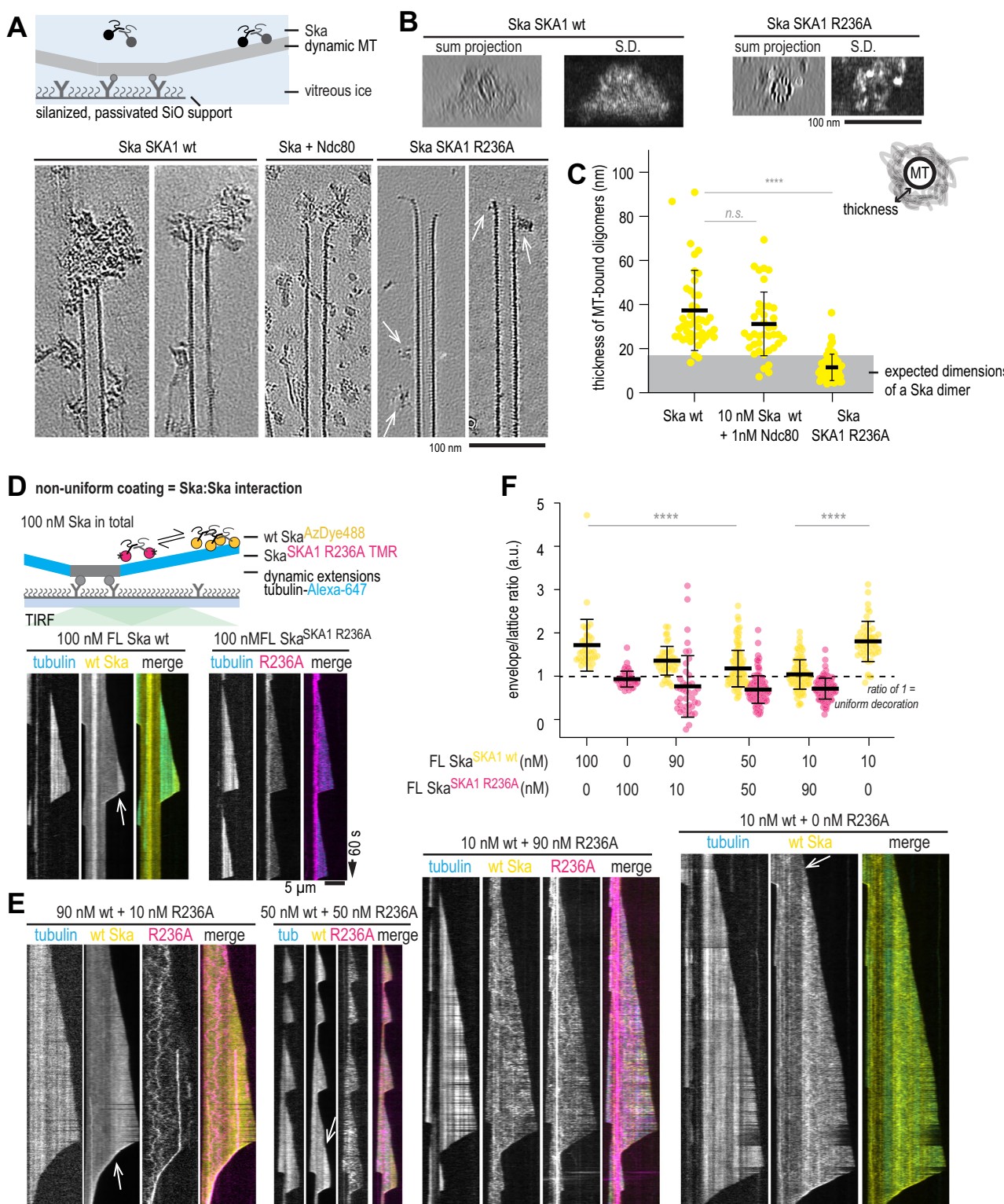

microtubules, and lowering the concentration of Ska to 0.1–1 nM, we observed that FL Ska molecules binding to microtubules had a brightness corresponding to monomers and dimers (Figs. 6A and EV6A). As expected, Ska$^{SKA3\Delta C}$ showed a similar brightness distribution, while the SKA1 MTBD only existed in the monomer form (Fig. EV6A).

We then measured the residence time of Ska molecules on microtubules in these conditions. FL Ska and Ska$^{SKA3\Delta C}$ interacted with microtubules with a wide distribution of residence times, sometimes exceeding 1 s, while the residence time of SKA1 MTBD never exceeded 0.4 s (Fig. 6B,C). The distribution of residence times of the FL Ska featured a slow and a fast component, with the fast

**Figure 5.   SKA1 R236A disrupts Ska:Ska interactions on dynamic microtubules.**

(A) Examples of 0.8-nm-thick slices through cryoCARE-denoised tomograms showing microtubule ends in presence of Ska FL Ska, Ska wt and Ndc80, or Ska$^{SKA1\,R236A}$. Arrows point to Ska$^{SKA1\,R236A}$ oligomers bound to microtubules. (B) Axial projections (sum, left; and standard deviation, right) along microtubules in presence of Ska wt or Ska$^{SKA1\,R236A}$, along with a schematic diagram showing how the thickness of Ska decoration was measured. (C) Distribution of thicknesses of Ska decorations measured in tomograms. Yellow circles show individual measurements, lines show mean ± SD. $N = 45$ (Ska), 39 (Ska + Ndc80), 57 (Ska$^{SKA1\,R236A}$). Welch's t-test $p$ value: Ska vs Ska + Ndc80: 0.0989 (n.s.), Ska wt vs Ska$^{SKA1\,R236A}$: $7.4 \cdot 10^{-12}$ (****). Grey shaded area represents the expected dimensions of a Ska dimer (18 nm). (D) Schematic of the experimental setup to study simultaneous microtubule binding of FL Ska$^{SKA1\,wt}$ and Ska$^{SKA1\,R236A}$ using TIRF imaging, along with representative kymograph obtained with each complex in isolation. (E) Example kymographs showing microtubules decorated with the two constructs in concentrations indicated. White arrows show the persistent boundary between a brighter Ska envelope proximal to the growing microtubule end, and the rest of less brightly decorated microtubule lattice. (F) Envelope/lattice brightness ratio for Ska$^{SKA1\,wt}$ (yellow) and Ska$^{SKA1\,R236A}$ (magenta), added at concentrations indicated. Black lines show mean and SD. Welch's t-test $p$-values: Ska wt 10 nM vs 100 nM (0.4656, n.s.), Ska wt 100 nM vs wt:R236 50:50 ($4.3 \cdot 10^{-6}$, ****), Ska wt 10 nM with 0 vs 90 nM R236A ($3 \cdot 10^{-15}$, ****). Scale bar: 100 nm (A, B). Scale bar: 5 µm (horizontal), 60 s (vertical) (D, E). Source data are available online for this figure.

component quite similar in its slope to the single slope observed for the SKA1 MTBD (Figs. 6C and EV6). We interpreted this observation, together with the bimodal brightness distribution, as evidence that at sub-nanomolar concentrations the Ska dimer can partially dissociate into monomers (Figs. 6A,C and EV6A). However, we could not detect a difference in the residence time of molecules with a brightness corresponding to a monomer, compared to dimers (Fig. EV6A), therefore we did not treat them separately.

We limited further analysis to measuring the fraction of residence times that were only observed for monomeric SKA1 MTBD and LMTBD constructs (short events, ≤0.32 s), and the fraction of residence times that were never or very rarely observed for MTBD/LMTBD (long events, >0.32 s, Fig. 6C). Using this approach, we found that the dimeric SKA1 LZ-LMTBD construct bound to microtubules similarly to FL Ska (Figs. 6C and EV6), which is consistent with our findings based on the envelope formation assay that SKA1 LZ-LMTBD faithfully recapitulates the microtubule interaction of FL Ska (Fig. 4).

We further used the LZ-LMTBD construct to screen mutations in the SKA1 MTBD for their effect on Ska residence time on stable microtubules. We observed that the majority of K-A and R-A mutations resulted in a reduction of the observed fraction of long microtubule-binding events, with the exception of LZ-LMTBD R236A which bound to microtubules similarly to LZ-LMTBD (Figs. 6C,D and EV6). We confirmed that this mutation did not affect binding of single molecules of FL Ska by comparing residence times of FL Ska and FL Ska$^{SKA1\,R236A}$ (Figs. 6C,D and EV6). Results obtained this way remained unaffected by the value of the threshold we chose to separate long and short events (Fig. EV6C). We thus conclude the SKA1 R236A mutation does not affect Ska:microtubule binding in single-molecule conditions.

## Oligomerisation-deficient Ska fails to support cold-stable microtubule attachments

Properly formed kinetochore-microtubule attachments, as opposed to most other cellular microtubules, are resistant to a brief treatment with a low temperature, an established assay used to demonstrate how kinetochores stabilise microtubule ends (Rieder, 1981). To assess the cold stability of kinetochore microtubules in the background of oligomerisation-deficient Ska, we depleted endogenous SKA1 using siRNA, and then expressed GFP-tagged and siRNA-resistant SKA1 in HeLa cells (Fig. 7A). In the absence of a commercially available anti-SKA1 antibody, we used immunofluorescence analysis of kinetochore-localised endogenous SKA3 after treatment of cells to monitor depletion of SKA1 by siRNA (Fig. EV7A). Using this method, we observed a strong reduction of

kinetochore-bound SKA3 following the SKA1 siRNA treatment (Fig. EV7B). Consistently, we observed an overall reduction in SKA3 levels by immunoblotting (Fig. EV7C).

We additionally tested whether SKA1 R236A formed a functional Ska complex in cells. Western blotting showed that cellular SKA3 levels were restored upon the expression of siRNA-resistant GFP-SKA1 (Fig. EV7C). We also tested whether kinetochore localisation of SKA3 was affected by expression of SKA1 wt or R236A, and did not find a difference in the relative SKA3/SKA1 fluorescence levels at kinetochores (Fig. EV7D). We have thus established a system to assess the effects of SKA1 mutants.

Depletion of SKA1 using siRNA treatment led to a dramatic increase in cells experiencing chromosome alignment defects (Figs. 7B and EV7E). Expression of GFP-SKA1 wt rescued these defects; expression of GFP-SKA1 R236A reduced the amount of cells with chromosome alignment defects, however, not to the level observed for GFP-SKA1 wt (Figs. 7B and EV7E), consistent with a previous report (Monda et al, 2017).

Incubation of cells transiently expressing GFP-SKA1 wt on ice for 10 min depolymerised majority of microtubules, but kinetochore-bound microtubules were retained (Fig. 7C). On the contrary, expression of GFP-SKA1 R236A followed by the same cold treatment resulted in a strong reduction, or a complete absence of kinetochore-microtubule attachments, despite GFP-SKA1 R236A being faithfully recruited to kinetochores (Fig. 7C). Cells transfected with GFP-SKA1 wt but with poor expression of the protein, as judged by the lack of the GFP signal at kinetochores, were also characterised by unstable kinetochore-microtubule attachments following the cold treatment (Appendix Fig. S1A). We further confirmed that despite an unchanged GFP signal at kinetochores colocalised with the Anti-centromere antibody (ACA) staining (Fig. 7D, $p = 0.77$), expression of GFP-SKA1 R236A resulted in severely reduced fluorescence intensity of kinetochore-proximal tubulin staining (Fig. 7D, $n = 3$ repeats, $p = 0.0064$).

## Oligomerisation-deficient Ska fails to stabilise microtubule ends against disassembly in vitro

Finally, we used the in vitro microtubule-stabilisation assay we designed earlier (Figs. 1C and 8A) to test the effect of SKA1 R236A mutation. We repeated the experiment with FL Ska wt and FL Ska$^{SKA1\,R236A}$, and found that 1–10 nM FL Ska$^{SKA1\,R236A}$, in presence of 0.3–1 nM Ndc80, failed to stabilise microtubules against disassembly, unlike FL Ska$^{SKA1\,wt}$ in similar conditions (Fig. 8B). Notably, the oligomerisation-deficient FL Ska$^{SKA1\,R236A}$ was able to follow shortening microtubule ends in this experiment, as we also

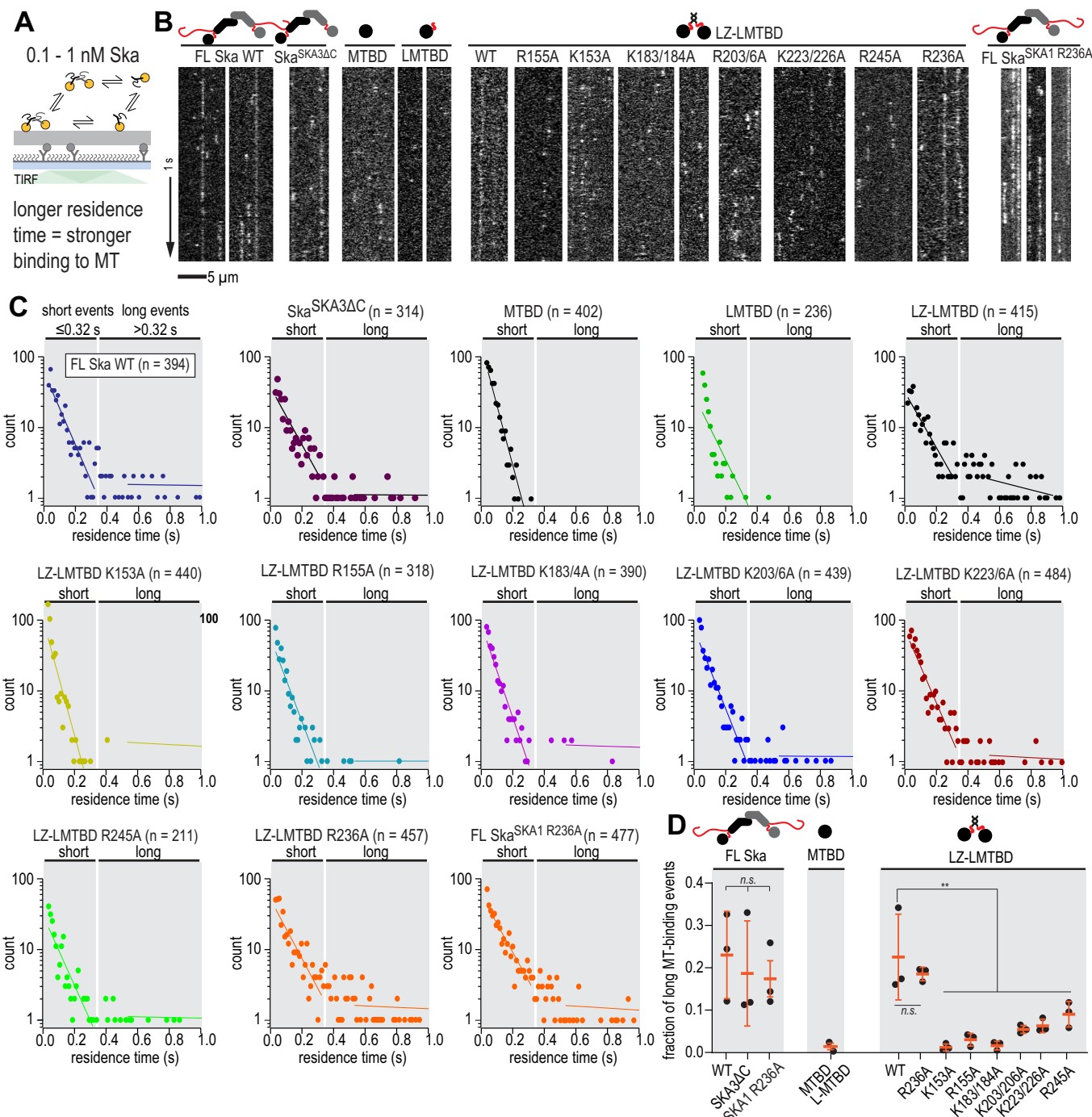

**Figure 6. Oligomerisation-deficient Ska has intact microtubule binding in single-molecule conditions.**

(A) Schematic of the experimental setup with Ska in the concentration of 0.1–1 nM interacting with coverslip-attached taxol-stabilised microtubules. (B) Example kymographs showing single molecule binding events of the constructs indicated. (C) Residence time distributions of the constructs indicated. Shaded areas represent events considered 'short' and 'long' during quantification. Rare events exceeding 1 s in duration not plotted, but included in the analysis. (D) Fraction of 'long' binding events for the constructs indicated. Orange lines show mean ± SD of repeated experiments. Welch's t-test: FL Ska$^{SKA1 wt}$ vs FL Ska$^{SKA1 R236A}$ $p = 0.49$ (n.s.); LZ-LMTBD wt vs LZ-LMTBD R236A $p = 0.96$ (n.s.); FL vs LZ-LMTBD $p = 0.95$ (n.s.); FL Ska$^{SKA1 R236A}$ vs LZ-LMTBD R236A $p = 0.82$ (n.s.); 1-way ANOVA of LZ-LMTBD wt vs non-R236A mutants $p = 0.0061$ (**). Scale bar: 5 μm (horizontal), 1 s (vertical). Source data are available online for this figure.

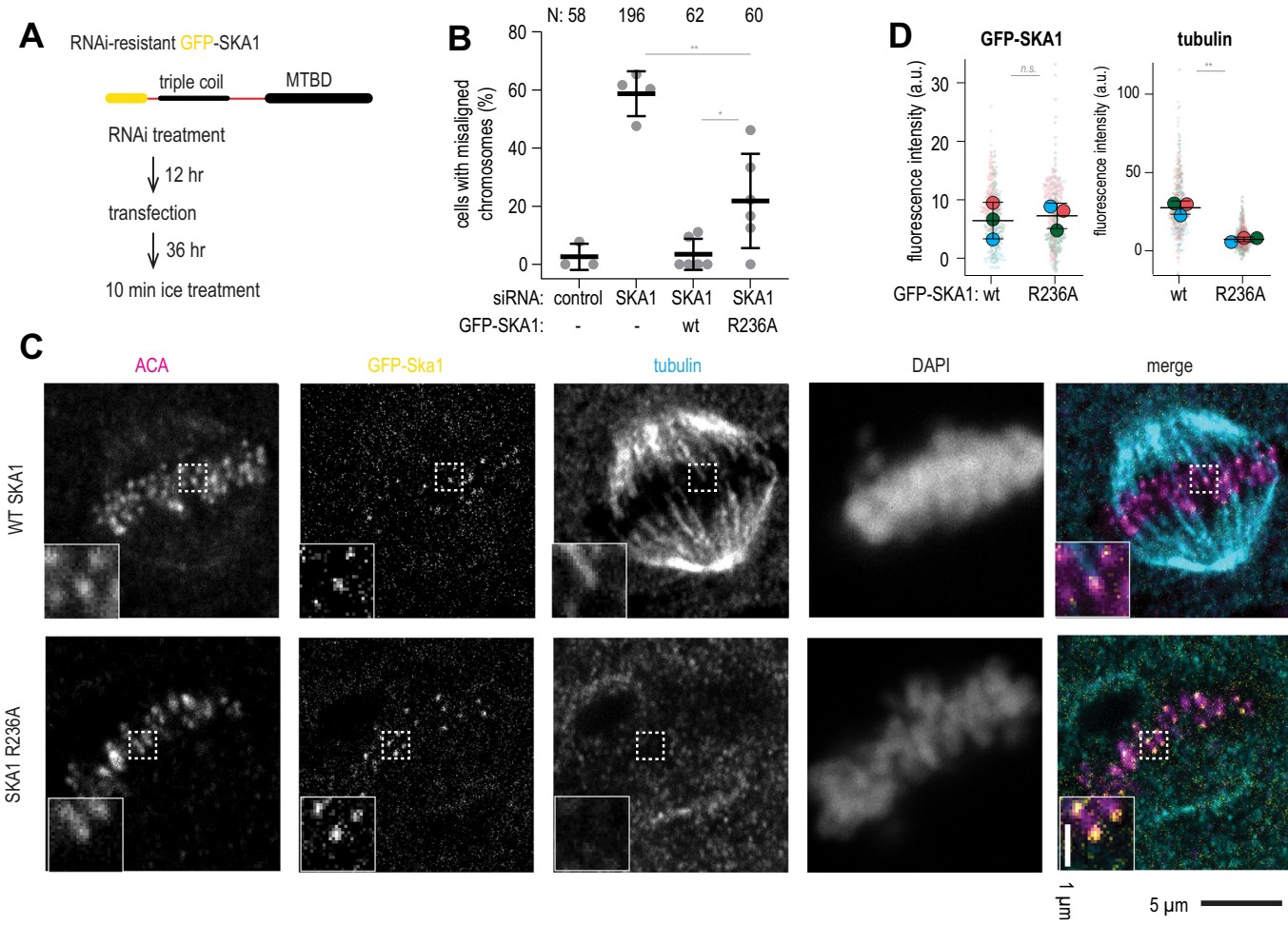

**Figure 7. Oligomerisation-deficient Ska fails to support cold-stable microtubule attachments.**

(A) Experimental setup to express RNAi-resistant GFP-SKA1 in HeLa cells treated with RNAi to deplete endogenous SKA1, followed by the 10 min incubation of cells on ice. (B) Fraction of cells with chromosome alignment defects in conditions indicated. Dots: individual values, lines: mean ± SD. Welch's t-test p-values: control siRNA vs SKA1 siRNA ($8.5 \cdot 10^{-5}$); SKA1 siRNA with or without GFP-SKA1 wt ($6.5 \cdot 10^{-5}$); SKA1 siRNA with or without GFP-SKA1 R236A (0.0016, **); GFP-SKA1 wt vs GFP-SKA1 R236A (0.0383, *). (C) Single planes from z-stacks of confocal images of cells stained for ACA (magenta), GFP (yellow), tubulin (cyan), and DAPI. Insets show magnified regions boxed from each panel. (D) GFP and tubulin intensity at kinetochores. Small circles represent individual kinetochores, large circles represent mean values per repeat, black lines show mean ± SD. N = at least 20 kinetochores per cell, 22 (wt) or 23 (R236A) cells in total, repeated 3 times. Paired t-test for repeated measurements (n = 3): p = 0.0064 (**). Source data are available online for this figure.

observed earlier in conditions that did not force microtubules to depolymerise (Figs. 4F and 8B). As a consequence of the oligomerisation-deficient mutant's inability to stabilise microtubule ends, we did not observe an enrichment of fluorescent microtubule extensions following 10 min of tubulin dilution in the presence of Ndc80 and FL Ska^SKA1 R236A, compared to Ndc80 alone (2-way ANOVA: p = 0.11; Fig. 8C). We thus conclude that Ska oligomerisation promotes cooperative Ska:Ndc80 interactions, thereby preventing disassembly of kinetochore-attached microtubules of the mitotic spindle.

## Discussion

The binding between outer kinetochores and microtubule ends is highly multivalent and confined in space and time to the middle of

a mitotic cell. It is therefore difficult to systematically disentangle the interactions of microtubule-binders with microtubules from hetero- or homotypic interactions between microtubule-binders. Here, we developed experimental strategies to (1) produce samples with microtubule ends stabilised by Ndc80 and Ska, and suitable for cryoET analysis; and (2) distinguish Ska:Ska interactions on microtubules from direct Ska:microtubule binding.

The first of these approaches allowed us to study the structure of microtubule ends in a state of induced rescue, and compare it to the structure of freely shortening microtubule ends (Figs. 1–3). We found that Ndc80 alone was able to stabilise microtubules against shortening, and that presence of Ska reduced the concentration of Ndc80 necessary for this effect—both findings consistent with previous assays quantifying effects of Ndc80 and Ska on microtubule dynamics with and without applied force (Volkov et al, 2018; Huis in 't Veld et al, 2019; Helgeson et al, 2018; Umbreit et al, 2012).

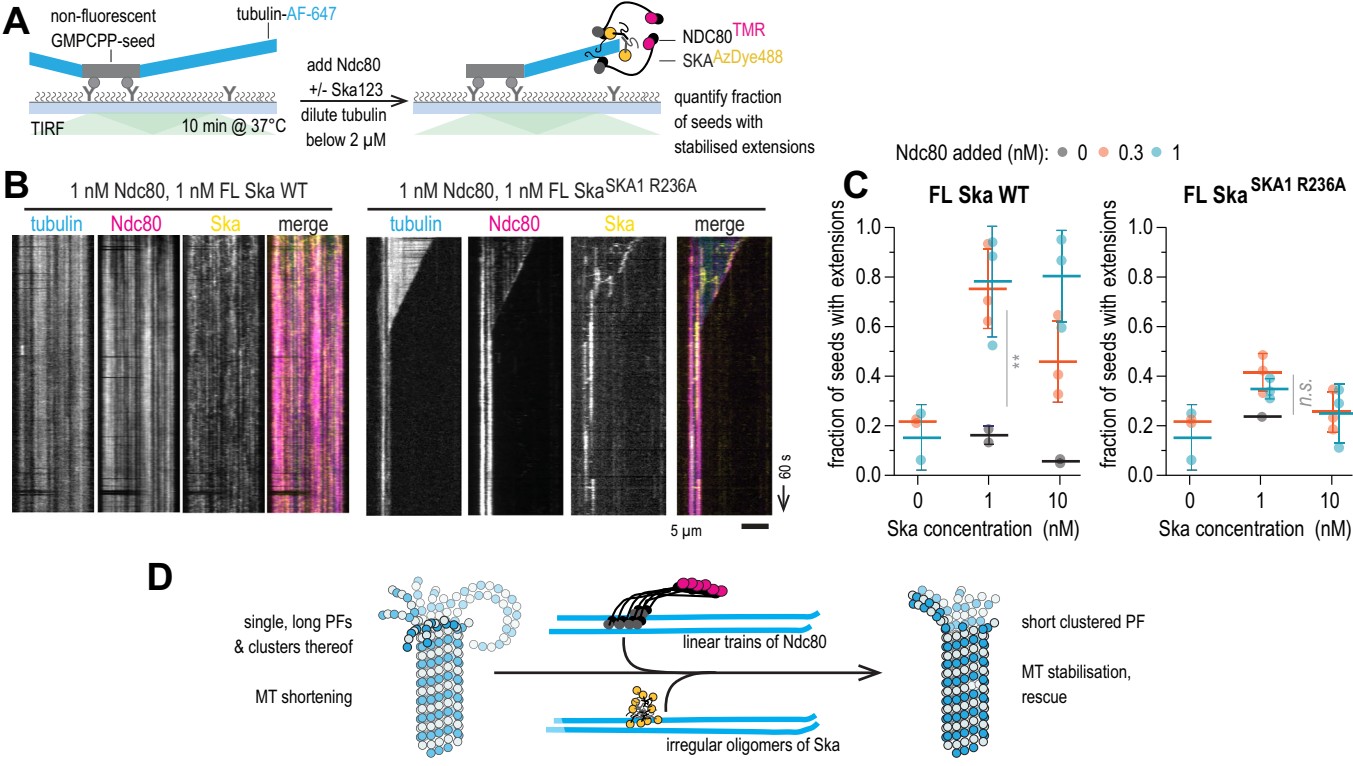

**Figure 8. Oligomerisation-deficient Ska fails to stabilise microtubule ends against disassembly in vitro.**

(A) Schematic of an experiment to test microtubule end stabilisation by Ndc80 and Ska. (B) Example kymographs showing microtubule length over time following the simultaneous addition of 1 nM Ndc80 with 1 nM Ska (wt or SKA1 R236A). (C) Fraction of unlabelled GMPCPP-stabilised microtubule seeds with fluorescent tubulin extensions following 10 min after tubulin dilution. Grey: no Ska added, blue: FL Ska$^{SKA1wt}$, FL Ska$^{SKA1R236A}$ (at least 50 seeds quantified in total over at least 5 fields of view per repeat, repeated 2–3 times). Lines and error bars represent mean ± SD. Two-way ANOVA for FL Ska$^{SKA1 wt}$: row factor (Ska concentration) $p = 0.0012$; column factor (Ndc80 concentration) $p = 0.0002$ (**). Two-way ANOVA for FL Ska$^{SKA1 R236A}$: row factor (Ska concentration) $p = 0.0084$; column factor (Ndc80 concentration) $p = 0.11$. (D) Cooperative oligomers of Ndc80 and Ska promote microtubule stabilisation via clustering of tubulin protofilaments. Source data are available online for this figure.

We also found that microtubule end-binding of self-assembled Ndc80 oligomers correlated with increased lateral clustering of tubulin protofilaments; however, this clustering started to increase at Ndc80 concentrations that were not sufficient to fully stall a shortening microtubule end (Fig. 3). We also report an increase in protofilament clustering at plus-ends in direct contact with an Ndc80 oligomer, compared to other plus-ends in the same sample that did not directly interact with Ndc80. We interpret our data to postulate that oligomers of Ndc80 bound near microtubule ends promote lateral association of bent protofilaments at the end; this property was previously identified as the crucial difference between growing and shortening microtubule ends (Kalutskii et al, 2025). Currently available data do not allow us to unambiguously identify the mechanism underlying the stabilising effect of the Ndc80 or Ndc80:Ska oligomers on the microtubule end beyond an increase in protofilament clustering. One possibility is that formation of Ndc80 "trains" on neighbouring protofilaments that we observe frequently (Fig. 2, Videos EV1–3) promotes lateral association between these protofilaments, which in turn converts a shortening microtubule end into a conformation compatible with growth. There may be other, not yet identified mechanisms of Ndc80-mediated microtubule end stabilisation that will require careful

further study using higher resolution and separation of function mutants.

In search of oligomerisation-affecting mutants that would allow us to dissect the microtubule end-stabilisation mechanisms of Ndc80:Ska oligomers, we performed screening of point mutants of SKA1 MTBD using a minimal construct that recapitulates the FL Ska microtubule-binding properties. Comparing residence time of SKA1 MTBD mutants in single-molecule conditions to uniformity of their microtubule decoration at high concentrations, we have identified the R236A mutation as the one that impairs the normally observed non-uniform microtubule decoration at high Ska concentrations without affecting individual molecules' residence time on microtubules (Figs. 4–6). We further confirmed that this mutation has similar effects on Ska:Ska and Ska:microtubule interactions both in the minimal LZ-LMTBD construct, and in the FL Ska complex.

A number of alanine substitutions of lysine and arginine residues, in various combinations, were reported to affect the amount of Ska sedimented after centrifugation with microtubules, the microtubule end-tracking properties of purified Ska, as well as the duration of mitotic progression (Abad et al, 2014; Monda et al, 2017). This also applies to the R236A mutation, which was shown

to reduce bulk microtubule binding and extend the duration of mitosis, alone and in combination with R245A and R155A (Abad et al, 2014; Monda et al, 2017), although without major mitotic alignment defects. Results presented here allow us to conclude that contrary to the rest of the R/A and K/A mutations tested, which affect the Ska:microtubule interaction directly, the R236A mutation does not affect affinity of individual Ska molecules to the microtubule lattice—likely through the remaining K/R residues in SKA1 MTBD making direct contacts with tubulin (preprint: Zhou et al, 2025). Based on our cryoET imaging and envelope assays, we interpret the effect of the R236A mutation as impairing Ska:Ska interactions. It is important to note that FL Ska$^{SKA1\ R236A}$ in our in vitro experiments was able to follow the shortening microtubule ends (Figs. 4F and 8B), further reinforcing our interpretation that this mutation does not affect Ska's ability to interact with microtubule lattices or ends per se.

We further showed that SKA1 R236A, despite interacting with microtubules normally in vitro, led to a severe reduction in the cold-stability of kinetochore fibres, a hallmark of reduced kinetochore-microtubule binding strength (Fig. 7). This experiment is directly supplemented by our in vitro test for microtubule end stabilization by the cooperative oligomers of Ndc80 and Ska (Fig. 8). Ndc80 and Ska at low nanomolar concentrations stabilise microtubules against disassembly triggered by sudden dilution of tubulin, however, substitution of Ska$^{SKA1\ wt}$ to Ska$^{SKA1\ R236A}$ leads to loss of this stabilisation.

Results of our cryoET analysis of individual oligomers of Ska binding to dynamic microtubules in presence of soluble tubulin (Fig. 5C) are consistent with the formation of multi-layered Ska oligomers, that are held together, at least partially, by Ska:Ska interactions, because their dimensions exceed the expected dimensions of a single layer of microtubule-bound Ska. These multi-layered assemblies are not typical for microtubule-bound proteins: we only observed a single microtubule-bound layer of Ndc80 oligomers (Fig. 2), and the same was true for a previously described microtubule plus-end tracking protein network, which formed comets that were also consistent with a single-layer microtubule decoration (Maan et al, 2023). Given that Ska was previously reported to interact with soluble tubulin (Monda et al, 2017), it may be possible that SKA1 R236 is involved in an indirect Ska:Ska interaction via tubulin, in a direct interaction with another region or domain of the Ska complex, or is part of an indirect binding of another kind. Our low-resolution subtomogram averages did not detect regular SKA1 MTBD densities on microtubule lattice; however, a recent preprint reports such structures using single-particle cryoEM approach using fully saturated, GMPCPP-stabilised microtubules (preprint: Zhou et al, 2025).

It is possible that Ska:Ska self-interactions stabilise microtubule ends by cross-linking neighbouring tubulin protofilaments and prevent them from splaying apart. This conclusion is corroborated by our low-resolution averages of "trains" of Ndc80 CH-domains on two neighbouring protofilaments, in all conditions tested, with and without Ska present. In this context, we propose that Ska may be required to further oligomerise Ndc80 when its concentration or availability is insufficient, or else to shift Ndc80 "trains" from the microtubule lattice towards microtubule ends. We thus propose that a combination of Ndc80 oligomerisation via the loop region and the microtubule-bound CH-domains (Alushin et al, 2010; Polley et al, 2023), and Ska cross-linking via the SKA1 MTBD, and

potentially also SKA3 tail, are all necessary to achieve the required level of end stabilisation (Fig. 8D).

The kinetochore-microtubule binding has been previously shown to be stabilised by the applied force (Akiyoshi et al, 2010; Nicklas and Ward, 1994). We previously demonstrated how presence of Ska at the interface between the Ndc80 oligomer and a shortening microtubule end during force production can stabilise the stalled conformation of the microtubule ends, and in this way to increase the rate of force-dependent microtubule rescue (Huis in 't Veld et al, 2019). Our direct imaging of microtubule ends stabilised by Ndc80:Ska oligomers brings new information about the functioning of this system, such as the lateral clustering or protofilaments by Ndc80, and the effect of Ndc80 and Ska on the shortening of the protofilaments (Figs. 2, 3 and EV3). Future work will be necessary to understand how the structures of microtubule ends, and Ndc80 and Ska oligomers binding to them, are affected by the applied force.

It also remains to be tested whether Ska recruitment to the interface between the Ndc80 and the microtubule end is force sensitive. Although our previous results with isolated Ndc80 trimers argue against this possibility (Nick Maleki et al, 2023), higher order oligomers of Ndc80, such as observed in this study, may be able to recruit Ska more efficiently in response to the tension generated across the kinetochore-microtubule interface. Thus, it still remains unclear what triggers an increased recruitment of Ska to the kinetochore so late before the metaphase-to-anaphase transition (Auckland et al, 2017; Cheerambathur et al, 2017), since the only other known determinant of the direct Ska:Ndc80 interaction, namely the Cdk1 phosphorylation of SKA3 T358 and T360 (Huis in 't Veld et al, 2019; Zhang et al, 2017), should be present throughout mitosis due to high levels of Cdk1 activity. It remains to be tested whether effects of MPS1 (Maciejowski et al, 2017), PP1 (Conti et al, 2019; Sivakumar et al, 2016), or other regulatory events affect the assembly of cooperative Ndc80:Ska oligomers at the correct time after the kinetochore biorientation.

# Methods

**Reagents and tools table**

| Reagent/Resource | Reference or Source | Identifier or Catalog Number |
|---|---|---|
| **Experimental models** | | |
| HeLa (H. sapiens) | | ATCC CCL-2 |
| **Recombinant DNA** | | |
| pBIG1 Ndc80 | Volkov et al, 2018 | |
| pBIG1 Ska | Huis in 't Veld et al, 2019 | |
| pBIG1 Ska SKA1 ΔMTBD | Huis in 't Veld et al, 2019 | |
| pBIG1 Ska SKA1 ΔMTBD SKA3 ΔC | Huis in 't Veld et al, 2019 | |
| pBIG1 Ska SKA3ΔC | This study, Huis in 't Veld et al, 2019 | Addgene 253354 |
| pBIG1 Ska SKA3 T358/ 360 A | This study, Huis in 't Veld et al, 2019 | Addgene 253355 |
| pBIG1 Ska SKA3Δ351-377 | This study | Addgene 253356 |
| pBIG1 Ska SKA1 R236A | This study | Addgene 253353 |

| Reagent/Resource | Reference or Source | Identifier or Catalog Number |
|---|---|---|
| pET28a-SKA1 MTBD | This study | Addgene 253359 |
| pET28a-SKA1 LMTBD | This study | Addgene 253360 |
| pET28a-SKA1 LZ-LMTBD | This study | Addgene 253361 |
| pET28a-SKA1 LZ-LMTBD K153A | This study | Addgene 253362 |
| pET28a-SKA1 LZ-LMTBD R155A | This study | Addgene 253363 |
| pET28a-SKA1 LZ-LMTBD K183/184 A | This study | Addgene 253364 |
| pET28a-SKA1 LZ-LMTBD K203/206 A | This study | Addgene 253365 |
| pET28a-SKA1 LZ-LMTBD K223/226 A | This study | Addgene 253366 |
| pET28a-SKA1 LZ-LMTBD R236A | This study | Addgene 253367 |
| pET28a-SKA1 LZ-LMTBD R245A | This study | Addgene 253368 |
| pGFP-N-GW GFP-SKA1 | This study | Addgene 253357 |
| pGFP-N-GW GFP-SKA1 R236A | This study | Addgene 253358 |
| **Antibodies** | | |
| anti-Digoxigenin | Roche | 11333089001 |
| anti-Tubulin | Abcam | Ab6046 |
| ACA | Antibodies Inc | 15-234 |
| anti-SKA3 | Santa Cruz Biotech. | SC 390966 |
| anti-rabbit Alexa-Fluor568 | Abcam | Ab175471 |
| anti-mouse Alexa-Fluor488 | Invitrogen | A106680 |
| anti-mouse Alexa-Fluor594 | Abcam | Ab150116 |
| anti-human Alexa-Fluor647 | Jackson | 609-604-213 |
| **Oligonucleotides and other sequence-based reagents** | | |
| anti-SKA1 siRNA | Horizon | D-015917-04 |
| control siRNA | Horizon | D-001210-01 |
| **Chemicals, Enzymes and other reagents** | | |
| GGGGK-TMR peptide | Thermo Fisher | |
| GGGK-AzDye488 peptide | Vector Labs | |
| **Software** | | |
| MotionCor2 | Zheng et al, 2017 | |
| IMOD 4.11 | Kremer et al, 1996 | |
| relion-5.0 | Burt et al, 2024 | |
| CTFfind 4.1 | Rohou and Grigorieff, 2015 | |
| napari | Sofroniew et al, 2025 | |
| Chimera X 1.8 | Pettersen et al, 2021 | |
| cryoCARE | Buchholz et al, 2019 | |
| protofilament clustering determination | Kalutskii et al, 2025 | |

## Cloning, expression, and purification of full-length Ndc80 and Ska complexes

Full-length Ndc80 and Ska complexes were expressed using previously described vectors (Huis in 't Veld et al, 2019; Volkov et al, 2018). SKA1 R236A mutation, and SKA3 Δ101-412 (SKA3ΔC), Δ351-377, and T358/360 A mutations were introduced into relevant pLIB vectors using conventional site directed mutagenesis techniques; a pBIG1 vector containing SKA3, SKA2, and SKA1 was generated using Gibson assembly. Baculovirus generation, and protein expression were both carried out in Sf9 insect cells. Cells were harvested 2–3 days after infection, washed in PBS and stored at −80 °C. Cells were thawed, resuspended in lysis buffer (20 mM Tris-HCl, pH 8.0, 150 mM NaCl, 5% v/v glycerol, 2 mM DTT, 20 mM imidazole, 0.5 mM PMSF, and protease-inhibitor mix (Pierce), and DNAse I (Roche)), lysed in a dounce glass homogenizer on ice (using 15–30 passes of the pestle), and the lysate was cleared by centrifugation at ca. $88,000 \times g$ for 45 min. The cleared lysate was mixed with pre-equilibrated HIS-Select Ni affinity beads (Merck) and incubated for 1–2 h at 4 °C with rotation. Beads were washed using 50 mL of lysis buffer without protease inhibitors, collected in a gravity flow column, and the protein was eluted manually in 1 mL fractions using the same buffer with 250 mM imidazole.

FL Ska wt was further purified by ion-exchange chromatography. Ni-purified fractions were diluted 5-fold in buffer A (20 mM Tris-HCl, pH 8.0, 30 mM NaCl, 5% v/v glycerol, 2 mM DTT), applied to a Capto HiRes Q 5/50 column (Cytiva) equilibrated in the same buffer, and eluted using a linear gradient from 30 to 500 mM NaCl in 80 mL. This step was omitted for FL Ska[SKA1 R236A], Ska[SKA3ΔC], and Ska[SKA3Δ351-377] which eluted from Ni beads in a clean enough form, and the protein was instead desalted using a PD-10 desalting column. In both cases, relevant fractions were pooled, concentrated in 50 kDa molecular weight cut-off concentrators (Thermo Fisher) and applied to a Superose 6 Increase 10/300 column (Cytiva) equilibrated in 20 mM Tris-HCl, pH 8.0, 150 mM NaCl, 5% v/v glycerol, 2 mM DTT. Relevant fractions were pooled, concentrated, flash-frozen in liquid nitrogen, and stored at −80 °C.

FL Ndc80 was expressed and purified as described previously (Huis in 't Veld et al, 2019) with slight modifications. Baculovirus generation and protein expression were both carried out in Sf9 insect cells. Cells were resuspended in lysis buffer (50 mM Hepes, pH 8.0, 250 mM NaCl, 5% v/v glycerol, 2 mM DTT, 20 mM imidazole, 0.5 mM PMSF, and protease-inhibitor mix (Pierce)), lysed in a dounce glass homogenizer on ice (using 15–30 passes of the pestle), and the lysate was cleared by centrifugation at ca. $88,000 \times g$ for 45 min. Ni-affinity and SEC were performed as described above using a buffer with 50 mM Hepes and pH 8.0, 250 mM NaCl 5% v/v glycerol, 2 mM DTT. For separation on a Capto HiRes Q 5/50 column, protein was eluted with a 40 mL linear gradient in the same buffer supplemented with 1 mM EDTA and containing NaCl concentration increasing from 25 to 300 mM.

To label the proteins fluorescently, the C-terminal 6xHis tag on SKA1, or the C-terminal 6xHis tag on SPC24, were replaced with a fluorescent peptide GGGK-AzDye488 (Vector labs), or GGGGK-TMR (Thermo Fisher) in a 30 min reaction at 25 °C, immediately followed by size-exclusion chromatography. Sortase 7 M (Hirakawa et al, 2015), the protein complex, and the peptide were used in an approximate molar ratio of 1:10:100.

## Purification of Cdk1 and in vitro phosphorylation of Ska

Cdk1, Cyclin B, and CKS1 were coexpressed from the same vector (a kind gift from Antony Oliver) in SF9 insect cells. Cell pellets were resuspended in the lysis buffer containing 25 mM HEPES pH 7.5, 200 mM NaCl, 0.5 mM TCEP, protease inhibitor cocktail, and DNAse, lysed by sonication, and the lysate was cleared by centrifugation at $36,000 \times g$ for 60 min. Cleared lysate was applied to a 5 ml HiTrap TALON FF column, washed with buffer A containing 25 mM HEPES pH 7.5, 200 mM NaCl, 0.5 mM TCEP, then the same buffer containing 5 mM imidazole, and finally eluted in 25 ml of the same buffer containing 250 mM imidazole in a single step. Eluted material was diluted two-fold with lysis buffer and applied to a 1 ml HiTrap StrepXT column. The column was washed with buffer A, and the protein was eluted in 5 ml of the same buffer containing 50 mM Biotin in a single step. Strep elution fraction was concentrated to 500 µl and loaded onto a Superdex 200 10/300 column equilibrated in 10 mM HEPES pH 7.5, 200 mM NaCl, 0.5 mM TCEP, 5% (v/v) glycerol, 0.002% Tween-20. Relevant fractions were pooled, flash-frozen in liquid nitrogen and stored at −80 °C.

For in vitro phosphorylation, Ska complexes were exposed to CDK1:Cyclin-B in presence of 5 mM ATP and 10 mM $MgCl_2$ for 30 min at 25 °C, followed by another 30 min with the addition of Sortase and fluorescent peptide GGGK-AzDye488 in an approximate ratio of 1/10 and 10x to the Ska, respectively. Phosphorylated and labelled Ska was immediately separated from the rest of components using Superose 6 Increase 10/300 column (Cytiva) equilibrated in 20 mM Tris-HCl, pH 8.0, 150 mM NaCl, 5% v/v glycerol, 1 mM DTT. Relevant fractions were pooled, concentrated, flash-frozen in liquid nitrogen, and stored at −80 °C.

## Cloning, expression, and purification of SKA1 fragments

Fragments of SKA1 were subcloned from pLIB-SKA1 into pET28a +, and the Gcn4 leucine zipper fragment RMKQLEDK-VEELLSKNYHLENEVARLKKLVGER was inserted using two rounds of PCR. Site directed mutagenesis of SKA1 MTBD was performed in pET28a+ using conventional methods.

Expression of SKA1 fragments was performed in *E. coli* BL21(DE3) Rosetta cells grown at 37 °C in presence of Chloramphenicol and Kanamycin to an OD600 of 0.6. Protein expression was induced by the addition of 200 mM IPTG, and continued for 14–20 h at 18 °C. Cells were washed in PBS and pellets were stored at −80 °C. Cells were thawed on ice and resuspended in a lysis buffer (20 mM Tris-HCl, pH 8.0, 150 mM NaCl, 5% v/v glycerol, 1 mM DTT, 20 mM imidazole, 0.5 mM PMSF, and protease-inhibitor mix (Pierce), and DNAse I), lysed by sonication and cleared by centrifugation at $75,000 \times g$. The cleared lysate was mixed with pre-equilibrated HIS-Select Ni affinity beads (Merck) and incubated for 1–2 h at 4 °C with rotation. Beads were washed three times using lysis buffer without protease inhibitors, collected in a gravity flow column, and the protein was eluted manually in 1 mL fractions using the same buffer with 400 mM imidazole. Peak fractions were desalted using a PD-10 desalting column and concentrated in 10 kDa molecular weight cut-off concentrators (Thermo Fisher).

To label the proteins fluorescently, the C-terminal 6xHis tag on SKA1 was replaced with a fluorescent peptide GGGK-AzDye488

(Vector labs) using sortase in a 30 min reaction at 25 °C as described above, immediately followed by size-exclusion chromatography using a Superdex 75 Increase 10/300 column pre-equilibrated in 20 mM Tris-HCl, pH 8.0, 150 mM NaCl, 5% v/v glycerol, 1 mM DTT. Relevant fractions were pooled, concentrated in 10 kDa molecular weight cut-off concentrators, aliquoted and stored at −80 °C.

## Tubulin, microtubules, and preparation of flow chambers

Porcine brain tubulin was purified and labelled in house using standard protocols (Castoldi and Popov, 2003; Hyman et al, 1991) using amine-reactive Digoxigenin (Merck) or AlexaFluor-647 (Invitrogen). DIG-labelled GMPCPP-stabilised seeds were produced using two cycles of polymerization to fully replace GTP, as described previously (Volkov et al, 2018).

Taxol-stabilised microtubules were made by incubating 70 µM tubulin (with or without DIG- and fluorescent labels) in presence of 25% glycerol and 1 mM GTP for 20 min at 37 °C, and then stabilized by an addition of 25 µM taxol followed by another 20 min incubation. Polymerised microtubules were sedimented by centrifugation at $100,000 \times g$ in a TLA100 rotor over 60% glycerol cushion, and the pellet was resuspended in a buffer containing 80 mM K-PIPES pH 6.9, 1 mM EGTA, 4 mM $MgCl_2$, supplemented with 40 µM taxol. Polymerised microtubules were stored at 25 °C for up to 3 days.

Glass coverslips were silanised using PlusOne Repel Silane (Cytiva), coated with anti-DIG antibody (Roche), and passivated using 1% Pluronic F-127 (Merck) as described previously (Volkov et al, 2018). All experiments with dynamic microtubules were performed using the following buffer: 80 mM K-PIPES pH 6.9, 1 mM EGTA, 4 mM $MgCl_2$, 1 mM GTP, 1 mg/ml κ-casein, 0.1% methylcellulose, oxygen-scavenging mix consisting of 4 mM DTT, 0.2 mg/ml catalase, 0.4 mg/ml glucose oxidase and 20 mM glucose, 10–11 µM tubulin (3–5% fluorescently labelled), at 30 °C. All experiments with taxol-stabilised microtubules were performed using the following buffer: 80 mM K-PIPES pH 6.9, 1 mM EGTA, 4 mM $MgCl_2$, 1 mg/ml κ-casein, 0.1% methylcellulose, 40 µM taxol, oxygen-scavenging mix consisting of 4 mM DTT, 0.2 mg/ml catalase, 0.4 mg/ml glucose oxidase and 20 mM glucose, at 25 °C.

Microtubule end-stabilisation experiments were performed by first growing microtubules in a flow chamber using a solution containing 80 mM K-PIPES pH 6.9, 1 mM EGTA, 4 mM $MgCl_2$, 1 mM GTP, 1 mg/ml κ-casein, 0.1% methylcellulose, oxygen-scavenging mix consisting of 4 mM DTT, 0.2 mg/ml catalase, 0.4 mg/ml glucose oxidase and 20 mM glucose, 10 µM tubulin (3% fluorescently labelled), at 37 °C. After 10 min of incubation, the solution was exchanged to the one containing the same buffer without soluble tubulin (pre-warmed at 37 °C), but with addition of Ndc80 and/or Ska in the required concentration, the chamber was then sealed with VALAP, placed on the microscope stage pre-warmed to 30 °C, and imaging was commenced immediately. The time between the change of solution and start of imaging was typically 30–60 s.

## TIRF microscopy

Experiments were performed using a custom microscope based on the OpenFrame system (Lightley et al, 2023) and manufactured by

Cairn Research. The instrument is equipped with the following lasers: a 150 mW Omicron LuxX 488 nm, a 200 mW Omicron LuxX 638 nm, and a 150 mW OBIS 561 nm, steered through the GATACA Ilas$^2$ TIRF/FRAP illumination module into a Nikon CFI Apochromat TIRF 100x objective with NA 1.49. The other components include an ASI motorized XY stage, a Cairn MonoLED for brightfield imaging, a custom Cairn autofocus system, and an Okolabs temperature controller with an objective heating collar. Images were acquired using an ET 405/488/561/640 nm Laser Quad Band Set (Chroma) and an iXon Life 897 EMCCD (Andor) using the MicroManager 2.0 software (Edelstein et al, 2014).

## Electron cryo-tomography

Samples with dynamic microtubules decorated with FL Ska were prepared as described previously (Maan et al, 2023). SiO holey grids (SPI supplies) were treated with oxygen plasma and immediately immersed in PlusOne Repel Silane solution (Cytiva) for 2 min, followed by sequential rinsing in Ethanol and water. Each grid was taken by Leica EM GP2 plunger forceps, incubated for 1 min with a 7 µL drop of 0.2 µM anti-DIG antibody (Roche), and sequentially rinsed with MRB80, 1% Pluronic F-127, and MRB80. A grid passivated in this way was put in the chamber of the plunger equilibrated at 25 °C and 95% humidity, and further incubated with 9 µL of GMPCPP-stabilised microtubule seeds, followed by rinsing and addition of 5 µL of a buffer containing 25 µM tubulin, 1 mM GTP, and 1 mM DTT in 80 mM K-PIPES pH 6.9, 1 mM EGTA, 4 mM MgCl$_2$. Microtubules were allowed to grow for 5 min, after which the solution on the grid was replaced with the one containing 400 nM FL Ska and 5 nm gold particles, in addition to 25 µM tubulin, 1 mM GTP, and 1 mM DTT. Grids were blotted from the back side, plunge-frozen in liquid ethane, and stored in liquid nitrogen.

Tilt series were recorded using a Titan Krios microscope (FEI) equipped with a Gatan K2 electron detector. Automated image acquisition was performed using Tomography software (Thermo Fisher). Images were recorded at 300 kV with a nominal magnification of 33,000×, resulting in the pixel size of 4.24 Å at the specimen level. Energy filtering was performed at post-processing using a 30 eV-wide slit. Bi-directional tilt series ranged from 0° to ±60° with a tilt increment of 2°, with the total electron dose of 100 e − /Å$^2$ and the target defocus set to 4 µm.

Samples with Ndc80 and Ska oligomers stabilising the shortening microtubules were prepared using SiO Quantifoil R2/2 grids (EMS) passivated as described above. Microtubules were grown on grids suspended in the chamber of a Leica EM GP2 plunger equilibrated at 37 °C and 95% humidity. Microtubules were grown for 10 min in presence of the buffer containing 80 mM K-PIPES pH 6.9, 1 mM EGTA, 4 mM MgCl$_2$, 12 µM tubulin, 1 mM GTP, 1 mM DTT, and 1 mg/ml κ-casein. After this incubation, the ca. 4 µL of solution on the grid was mixed by pipetting with ca. 30 µL of a pre-warmed solution containing the same components, with tubulin substituted to Ndc80 +/− Ska at a given concentration, and 5 nm gold nanoparticles. Following the mixing, majority of the solution was removed, leaving 3–4 µL on the grid. After a 2 min incubation in the chamber of the Leica EM GP2 plunger, the grids were blotted from the back side and plunge-frozen in liquid ethane. Grids containing shortening microtubules in presence of 0 or 1 nM Ndc80 were blotted and plunge-frozen 45 s after tubulin dilution.

Tilt series were recorded using Titan Krios microscopes (FEI) equipped with a Gatan K3 electron detector. Grids with 0, 1, 3, and 10 nM Ndc80, respectively, were imaged at LonCEM (London Consortium EM, Francis Crick Institute), with data acquisition using TOMO5 software (FEI), at 300 kV with 42,000× magnification resulting in the pixel size of 2.11 Å at the specimen level. One grid with Ndc80 and Ska was imaged at the National Cryo-EM facility at eBIC, located within the Diamond Light Source, using TOMO5 software (FEI), at 300 kV with 53,000× magnification resulting in the pixel size of 1.63 Å at the specimen level. In all cases, bi-directional tilt series were recorded using a dose-symmetric scheme in the range from 0° to ±60° with a tilt increment of 3°, the total electron dose of 100 e − /Å$^2$ and the target defocus set in a range from 2.5 to 4.5 µm.

Tomograms were reconstructed, denoised, and analysed for microtubule polarity and oligomer dimensions as described previously (Maan et al, 2023), with motion correction using MotionCor2 (Zheng et al, 2017), tilt series alignment and back projection of tomograms using IMOD (Kremer et al, 1996), and denoising using cryoCARE (Buchholz et al, 2019). Protofilament length and clustering were determined as described previously (Kalutskii et al, 2025).

## Subtomogram averaging

Unprocessed raw movies were re-processed using the tomography workflow in Relion 5.0 (Burt et al, 2024). Motion correction was performed using MotionCor2, CTF correction was performed using CTFfind 4.1 (Rohou and Grigorieff, 2015) and tomograms were reconstructed in an automated mode using IMOD. Particles were picked as filaments with a repeat of 4.1 Å, using denoised tomograms in napari (Sofroniew et al, 2025). Microtubule polarity during particle picking was determined using inflection of the Ndc80 coiled coils, which pointed towards microtubule plus-ends. Subtomograms were extracted using a binning factor 4, and a box size of 192 unbinned pixels, then cropped to 96 unbinned pixels. Initial models were generated de novo using the variable-metric gradient descent algorithm over 100 iterations, without masking, and without any prior orientation of particles, using C1 symmetry. Initial models generated this way were used for automated 3D refinement, which was followed by removal of duplicate particles, and 3D classification and additional rounds of 3D refinement. In all samples, with or without addition of Ska, we did not find separate classes of microtubule-decorating proteins within a sample, over several rounds of classification with various number of classes from 3 to 9, and with various masks intended to dampen the contribution of the microtubule signal. Particles obtained from Ndc80-only samples were further refined with binning 2. Particles obtained from the sample with both Ndc80 and Ska present were only refined at binning 4. 3D rendering of post-processed particle averages was performed in Chimera X 1.8 (Pettersen et al, 2021).

## In vivo expression of SKA1

HeLa cells were grown in DMEM containing 4.5 g/L Glucose, 2 mM L-Glutamine, 110 mg/L sodium Pyruvate supplemented with 10% FBS and 100 µg/ml each of penicillin and streptomycin. Cultures were maintained at standard conditions at 37 °C in a humidified incubator with 5% CO$_2$. For SKA1 knockdown, HeLa cells at

approximately 65% confluence were transfected with siGENOME Ska1-targeting siRNA (sequence: 5′-CCCGCTTAACCTATAAT-CAAA-3′; Horizon Cat # D-015917-04). A non-targeting control siRNA (sequence: 5′-GCCAUUCUAUCCUCUAGAGGAUG-3′; Horizon Cat # D-001210-01) was used for comparison. Transfections were performed using Lipofectamine RNAiMAX (Invitrogen).

siRNA-resistant SKA1 DNA fragment was produced by gene synthesis (Genewiz). The synthesized SKA1 wt DNA fragment was further inserted into a pGFP-N-GW vector backbone amplified from a vector encoding GFP-PMF1 (kind gift from Viji Draviam). GFP-Ska1 R236A mutant was generated by using site directed mutagenesis by using siRNA resistant GFP SKA1 wt as template. For rescue assays, siRNA-resistant GFP-SKA1 constructs (wild-type and the R236A mutant) were transfected into cells after 12 h of treatment with siRNA. Plasmids were expressed for 36 h and transfection was carried out using Lipofectamine 3000 (Invitrogen). Microtubule cold stability assay was performed by incubating cells expressing GFP SKA1 wt and GFP SKA1 R236A mutant on ice for 10 min, followed by rinsing with ice-cold PBS; the cells were then immediately fixed in −20 °C methanol.

### Immunofluorescence and confocal microscopy

Coverslips fixed in −20 °C methanol were blocked and permeabilised using PBS containing 1% bovine serum albumin and 0.5% Triton X-100 for 1 h. Primary antibody, in a dilution specified below, was incubated for 2 h at room temperature. Fluorophore-conjugated secondary antibody diluted 1:1000 were incubated for 1 h. The coverslips were stained with DAPI for 45 s and mounted on glass slides using Prolong Diamond antifade reagent (Invitrogen). The following antibodies and dilutions were used: β-Tubulin 1:1000 (Abcam #ab6046), ACA 1:1000 (Antibodies Inc. #15-234), SKA3 1:700 (Santa Cruz Biotechnology #SC 390966), anti-rabbit AlexaFluor 568 (Abcam #ab175471), anti-Human AlexaFluor-647 (Jackson #609-604-213), anti-mouse AlexaFluor-594 (Abcam #ab150116), anti-mouse AlexaFluor 488 (Invitrogen #A10680).

Confocal microscopy images were obtained with a Leica Stellaris SP8 laser scanning confocal microscope using an HC PL APO CS2 63X/1.40 Oil immersion objective. Excitation was provided by a white light laser in the range of 440–790 nm for GFP, AlexaFluor-488, and AlexaFluor-568, and a diode laser 405 nm for DAPI. Fluorescence emission was collected from 430–484 nm for DAPI; 494–573 nm for GFP and AlexaFluor-488; 585–643 nm for AlexaFluor-568 and 663–800 nm for AlexaFluor-647. The confocal pinhole was set as airy 1 for all images. Optical sections in the z-direction of ~0.25 μm, and images were acquired at a scan speed of 400 Hz with a pixel dwell time of 2.825 μs, 2 × line averaging and a 512 × 512 pixel format.

### Image analysis

All image analyses were performed in Fiji (Schindelin et al, 2012). Kymographs were generated using a custom ImageJ script (https://github.com/volkovdelft/kymo.jl). Lifetimes in single-molecule conditions were measured manually in kymographs. For clarity of representation, rare binding events exceeding 1 s in duration have been omitted from plots of lifetime distributions shown in Figs. 6C and EV6B; however, these events were included in further analysis shown in Figs. 6D and EV6C.

Ratios of envelope intensities were measured using boxed regions in kymographs and calculated as $(I_{envelope} - I_{BG})/(I_{lattice} - I_{BG})$, where $I_{BG}$ represents background intensity next to the microtubule, measured using the same region of interest (ROI).

Intensities of SKA1, SKA3, and tubulin at kinetochores were measured in individual Z-planes of confocal imaging stacks, by manually selecting ROIs encompassing the ACA signal and projecting several pixels towards the relevant spindle pole, to capture the end of a k-fibre. The same ROI was used for SKA and tubulin channels. At least 20 kinetochores were quantified per cell and averaged to create a mean value per cell, which were in turn averaged to generate a mean value per repeat. Statistical analyses were performed in GraphPad Prism 10. Blinding was not performed.

### Statement on the use of AI

The use of machine learning was limited to cryoCARE denoising of tomograms. No other AI or ML algorithms were used for image generation or analysis. At no point in this study artificial intelligence or large language models were used for literature review, logical organization, writing, data analysis, and proofreading.

### Data availability

Subtomogram averages obtained in this study have been deposited to EMDB with the following accession numbers: EMD-56085 (10 nM Ndc80, Fig. 2E), EMD-56086 (3 nM Ndc80, Fig. EV2D), EMD-56087 (1 nM Ndc80 and 10 nM Ska, non-Ndc80 microtubule-bound densities, Fig. EV2E), and EMD-56088 (1 nM Ndc80 and 10 nM Ska, Ndc80 oligomers, Fig. EV2E). Plasmids generated in this study are available via Addgene and are listed in the Reagents and Tools Table.

The source data of this paper are collected in the following database record: biostudies:S-SCDT-10_1038-S44318-026-00749-5.

### Peer review information

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

## Acknowledgements

VAV is grateful to Andrea Musacchio (Max-Planck Institute of Molecular Physiology) and Marileen Dogterom (Delft University of Technology) for supporting the initial phase of this work. We also thank Charlotte Millership (insect cell facility, QMUL) for help with insect cell expression, Hui Zhang (EM facility, QMUL) for help with grid preparation and screening, Christoph Diebolder (NeCEN), Nora Cronin (LonCEM) and Emma Buzzard (Diamond Light Source) for help with cryoET data collection, Michaela Egertova (QMUL) for help with confocal microscopy, and Hanjin Liu for help with data analysis. We thank Antony Oliver (University of Sussex) for a gift of a plasmid encoding for Cdk1:CyclinB:CKS1, Viji Draviam (QMUL) for a gift of a plasmid encoding for GFP-PMF1, and Michel Steinmetz (Paul Scherer Institute) for a gift of a plasmid encoding for MACF-GCN4. PJH acknowledges funding from the Austrian Science Fund (FWF PAT1893225). VAV acknowledges grant support from the Biotechnology and Biological Sciences Research Council (BBSRC, BB/X014975/1) and the Wellcome Trust (308895/Z/23/Z). Confocal imaging using the Leica Stellaris 8 system was supported by BBSRC grant BB/W019698/1.

## Author contributions

**Renjith M Radhakrishnan**: Conceptualization; Data curation; Formal analysis; Validation; Investigation; Methodology; Writing—review and editing. **Lauren Stokes**: Investigation; Methodology; Writing—review and editing. **Matthew Day**: Resources; Investigation; Writing—review and editing. **Pim J Huis in 't Veld**: Conceptualization; Resources; Writing—review and editing. **Vladimir A Volkov**: Conceptualization; Resources; Data curation; Formal analysis; Supervision; Funding acquisition; Validation; Investigation; Visualization; Methodology; Writing—original draft; Project administration; Writing—review and editing.

Source data underlying figure panels in this paper may have individual authorship assigned. Where available, figure panel/source data authorship is listed in the following database record: biostudies:S-SCDT-10_1038-S44318-026-00749-5.

## Disclosure and competing interests statement

The authors declare no competing interests.

# Expanded View Figures

**Figure EV1. Purification of Ndc80 and Ska complexes, and Cdk1 phosphorylation of Ska.**

(A) SDS-PAGE of full-length Ndc80 and Ska complexes, using Coomassie staining (top), and in-gel fluorescence of TMR (left) or AzDye-488 (right). (B) SDS-PAGE of the CyclinB/Cdk1/ CKS1 complex (left) and Ska complex treated with CyclinB/Cdk1/CKS1 next to an untreated control, and Ska$^{SKA3\ T358/360A}$, Ska$^{SKA3\ \Delta351-377}$, and Ska$^{SKA3\Delta C}$. (C) Comparison of the microtubule-stabilising activity of Cdk1-treated and untreated Ska, and Ska$^{SKA3\ T358/360A}$ in presence of an indicated concentration of Ndc80 (mean ± SD). Two-way ANOVA comparing hyperphosphorylated and untreated Ska: row factor (Ndc80 concentration) $p = 0.0003$ (**); column factor (Ska treatment) $p = 0.33$ (n.s.). Two-way ANOVA comparing Ska$^{SKA3\ wt}$ and Ska$^{SKA3\ T358/360A}$: row factor (Ndc80 concentration) $p = 0.0092$ (**); column factor (Ska wt vs mutant) $p = 0.0015$ (**). At least 50 seeds quantified in total over at least 5 fields of view per repeat, repeated 2–3 times. (D) Length of residual fluorescent microtubule extensions after tubulin dilution in presence of the proteins in concentrations indicated. Dots: individual measurements, lines: mean ± SD. (E) Microtubule decoration by Ndc80 and Cdk1-treated and untreated Ska. Scale bar: 5 μm.

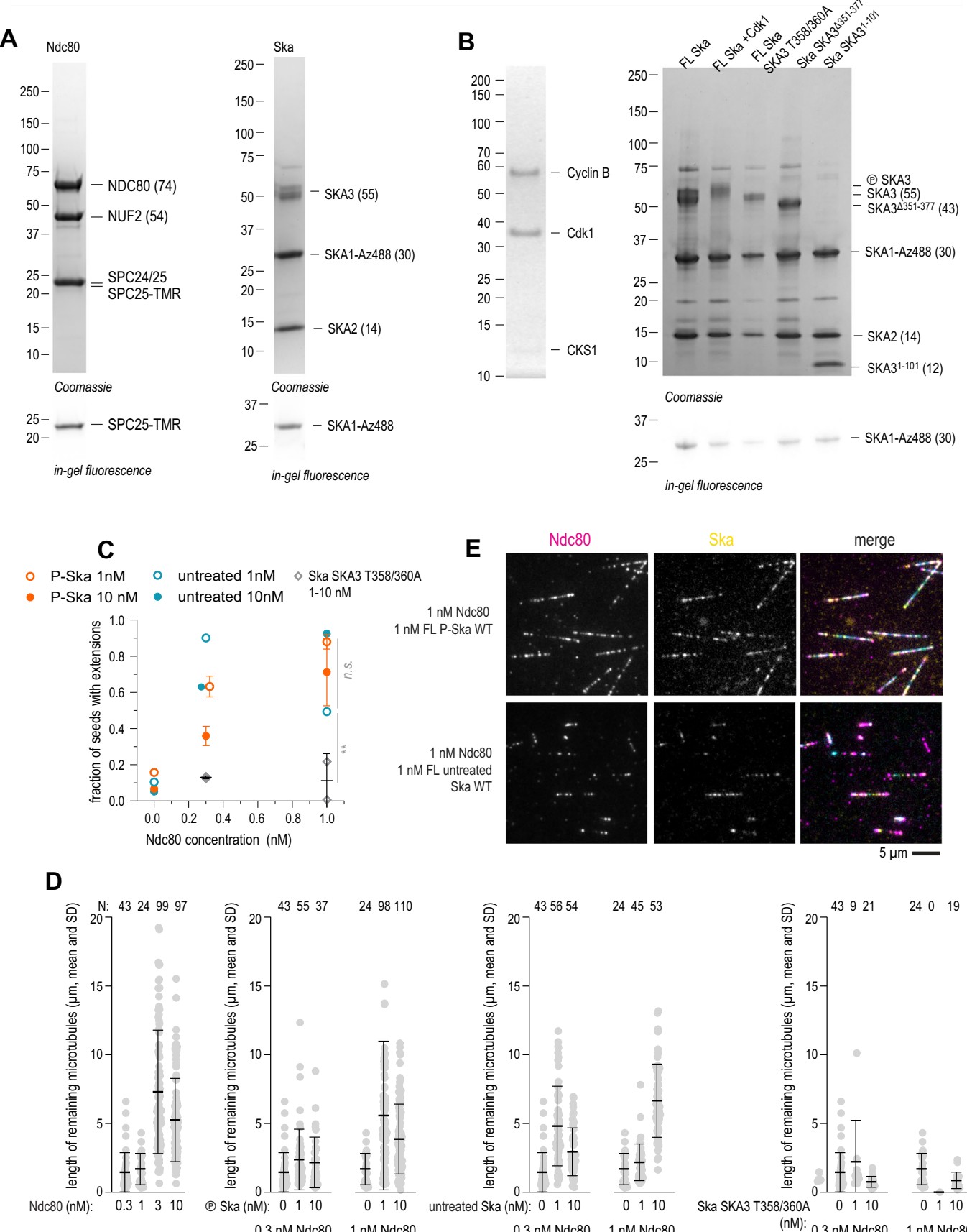

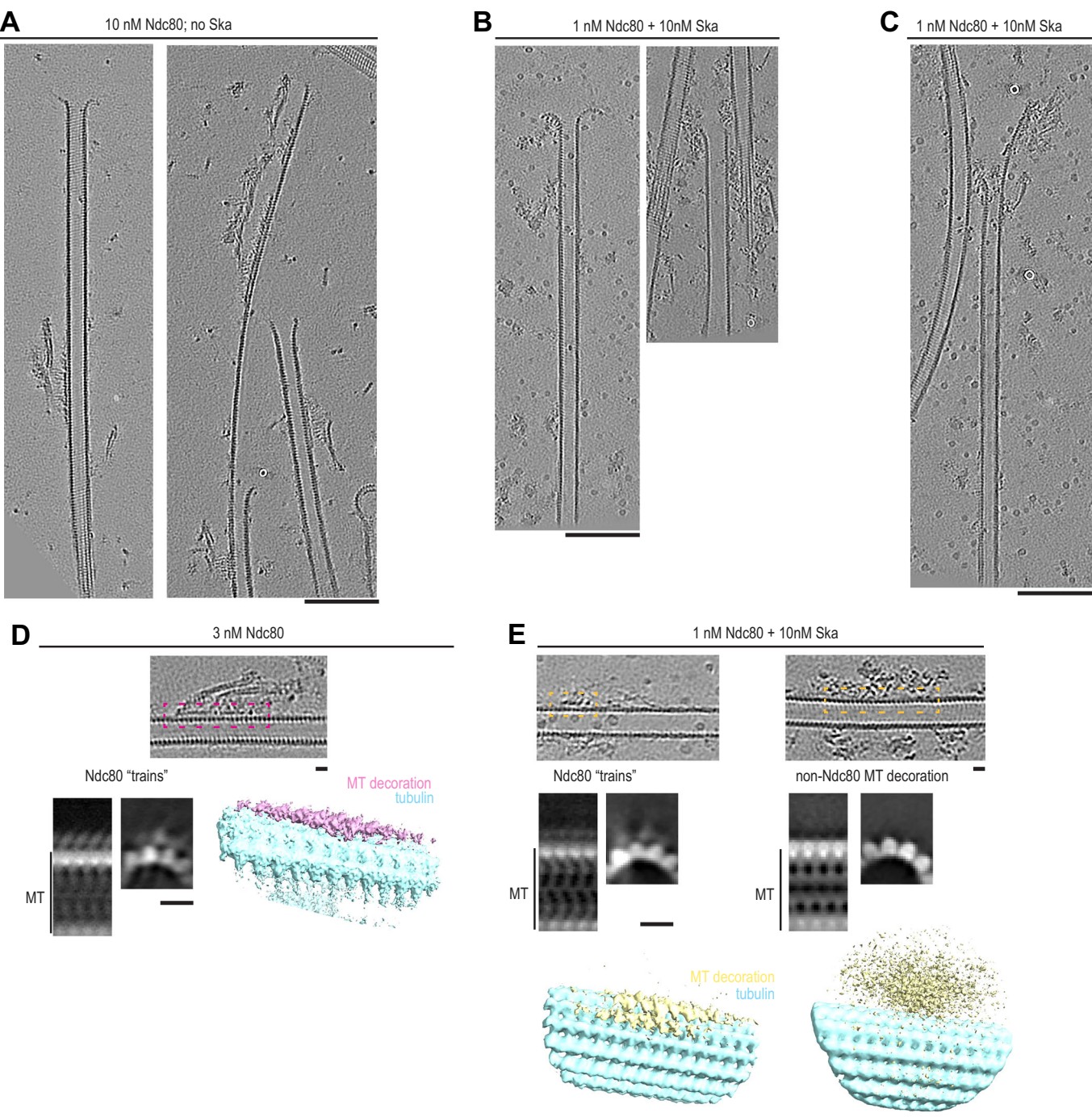

**Figure EV2.  Characterisation of Ndc80 and Ska oligomers on microtubules using cryoET.**

Additional examples of microtubules decorated with Ndc80 trains at 10 nM Ndc80 (**A**), with non-Ndc80 oligomers in presence of 1 nM Ndc80 and 10 nM Ska (**B**), and with Ndc80 trains stabilising extended sheet-like protofilaments in presence of 1 nM Ndc80 and 10 nM Ska (**C**). (**D**) Subtomogram average of Ndc80 trains in a sample with 3 nM Ndc80: 1285 particles obtained from 26 tomograms, deposited to EMBD as EMD-56086. Images shown are binned by 2 compared to the deposited map. (**E**) Subtomogram averaging of CH-domain trains (left), and non-Ndc80 microtubule decorations in the sample containing 1 nM Ndc80 and 10 nM Ska. White-on-black images represent 2D projections of 3D classes. Colour images show 3D rendering of the same classes. Ndc80 "trains": 822 particles obtained from 42 tomograms (EMD-56088). Non-Ndc80 densities: 5026 particles from the same 42 tomograms (EMD-56087).

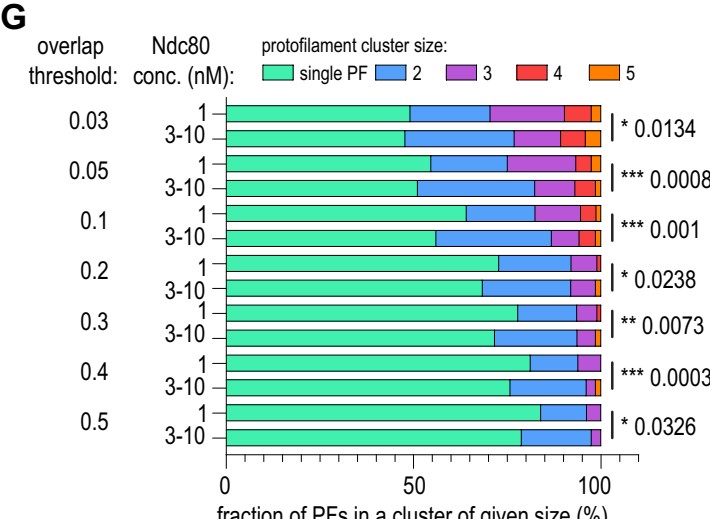

**A** — TIRF / cryoEM

MT length (μm)

| Ndc80 (nM): | 1 | 3 | 3 | 10 | 1 |
| Ska (nM): | 0 | 0 | 0 | 0 | 1 |

**B** MT length vs lattice spacing

seeds | dynamic extensions

lattice spacing near plus-end (nm)

○ 3–10 nM Ndc80
○ 1 nM Ndc80 + Ska

MT length (μm)

**C** plus-ends selected for analysis

lattice spacing near plus-end (nm)

shortening   stabilised with:
● Ndc80
● Ndc80 + Ska

free MTs | Ndc80 1 nM | Ndc80 3–10 nM | Ndc80 +Ska

N: 27   41   53   20

**D**

fraction of zero-length PF (%)

Ndc80 1 nM | Ndc80 3–10 nM | Ndc80 +Ska

**E**

N: 404   358   158

protofilament length (nm)

**   ***

Ndc80 1 nM | Ndc80 3–10 nM | Ndc80 +Ska

**F**

N: 36   41   18

raggedness (nm)

*
*   n.s.

Ndc80 1 nM | Ndc80 3–10 nM | Ndc80 +Ska

raggedness = SD of protofilaments' first bent segment position along the MT axis

**G**

overlap threshold: | Ndc80 conc. (nM): | protofilament cluster size:
■ single PF  ■ 2  ■ 3  ■ 4  ■ 5

| 0.03 | 1 / 3–10 | * 0.0134 |
| 0.05 | 1 / 3–10 | *** 0.0008 |
| 0.1 | 1 / 3–10 | *** 0.001 |
| 0.2 | 1 / 3–10 | * 0.0238 |
| 0.3 | 1 / 3–10 | ** 0.0073 |
| 0.4 | 1 / 3–10 | *** 0.0003 |
| 0.5 | 1 / 3–10 | * 0.0326 |

fraction of PFs in a cluster of given size (%)

◀ **Figure EV3.   Criteria to select microtubule plus-ends for analysis, and effect of the protofilament overlap parameter on the observed differences if protofilament clustering.**

(A) Microtubule length determined using TIRF microscopy, or low-magnification cryoEM in conditions used to detect target positions for tomography, in presence of Ndc80 and Ska in indicated concentrations. Grey circles: individual microtubules, lines: median. N: TIRF 1 nM Ndc80 (68), 3 nM Ndc80 (54); cryoEM 3 nM Ndc80 (15), 10 nM Ndc80 (41), 1 nM Ndc80 with 10 nM Ska (19). (B) Correlation of tubulin lattice spacing and microtubule length in the samples containing Ndc80 only (magenta), or Ndc80 + Ska (yellow). (C) Lattice spacing near plus-ends of microtubules selected for further analysis of protofilament shapes (mean ± SD). N: free MT (27), 1 nM Ndc80 (41), 3–10 nM Ndc80 (53), 1 nM Ndc80 with 10 nM Ska (20). (D) Fraction of protofilaments without a bent part in a microtubule plus-end. (E) Length of bent protofilament segments at microtubule plus-ends in presence of Ndc80 or Ndc80 + Ska. N: 1 nM Ndc80 (404), 3–10 nM Ndc80 (358), 1 nM Ndc80 with 10 nM Ska (158). Kolmogorov-Smirnov $p$-values: Ndc80 1 nM vs 3–10 nM: 0.0014 (**); Ndc80 3–10 nM vs Ndc80 + Ska: 0.4486 (n.s.); Ndc80 1 nM with vs without Ska: 0.0004 (***). (F) Raggedness of microtubule ends in conditions indicated. N: 1 nM Ndc80 (36), 3–10 nM Ndc80 (41), 1 nM Ndc80 with 10 nM Ska (18). Welch's t-test $p$-values: Ndc80 1 nM vs 3–10 nM: 0.0225 (*); Ndc80 3–10 nM vs Ndc80 + Ska: 0.9637 (n.s.); Ndc80 1 nM with vs without Ska: 0.0238 (*). (G) Protofilament cluster distributions obtained using indicated threshold value for neighbouring protofilament overlap. Chi-squared $p$-values: threshold of 0.03: 0.0134; threshold of 0.05: 0.0008; threshold of 0.1: 0.001; threshold of 0.2: 0.0238; threshold of 0.3: 0.0073; threshold of 0.4: 0.0003; threshold of 0.5: 0.0326.

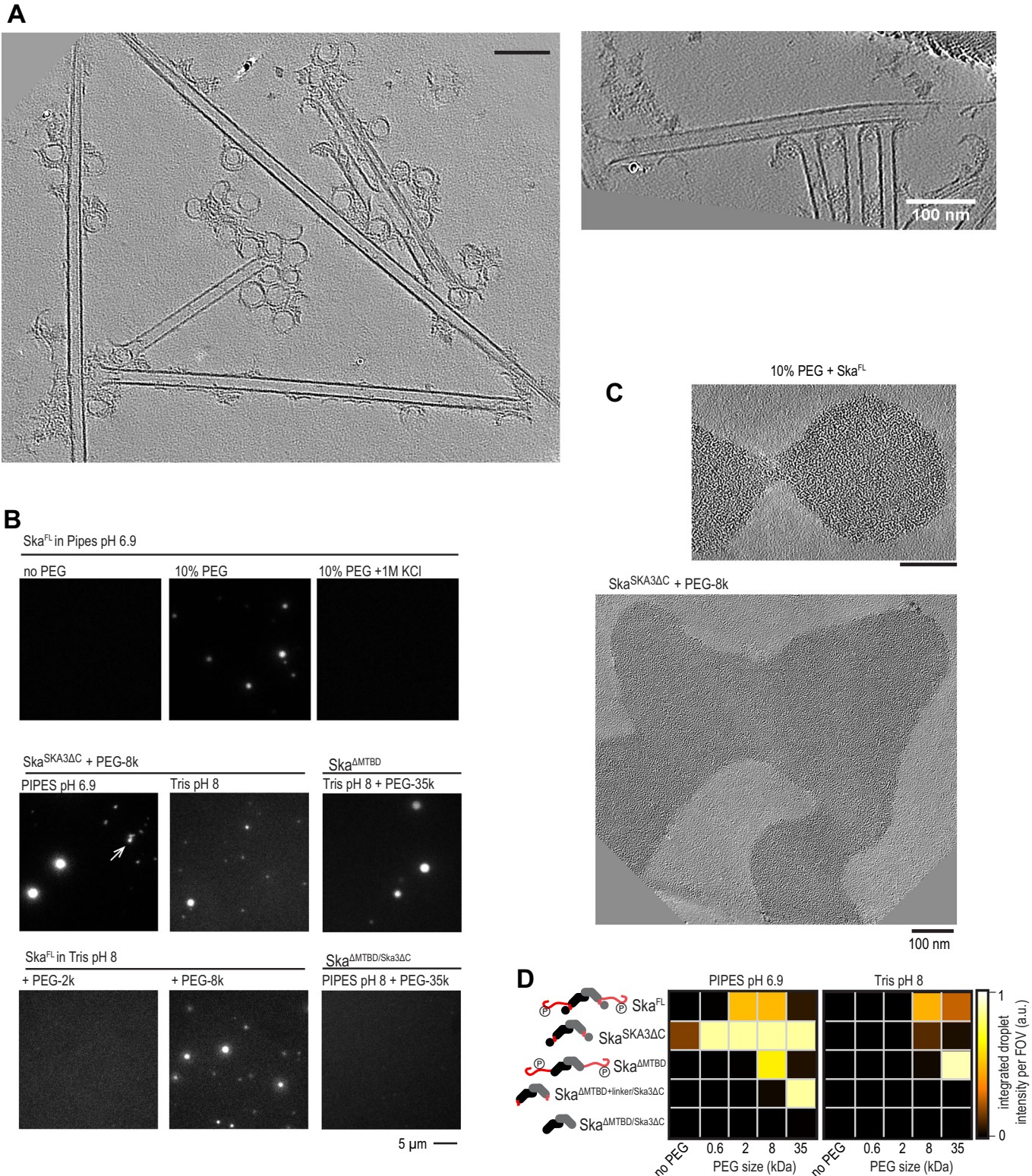

**Figure EV4. Self-interaction of the Ska complex with and without microtubules.**

(A) Representative slices through tomograms obtained in a sample containing Ska and dynamic microtubules. (B) Fluorescence microscopy images of Ska in presence or absence of crowding agents and various buffer compositions indicated on the panel. Arrow points to non-spherical aggregates of Ska^SKA3ΔC. (C) Slices through tomograms of FL Ska or Ska^SKA3ΔC in presence of PEG. (D) Formation of fluorescent self-assembling oligomers of Ska in two various buffers, with the deletion construct indicated, and using a crowding agent indicated.

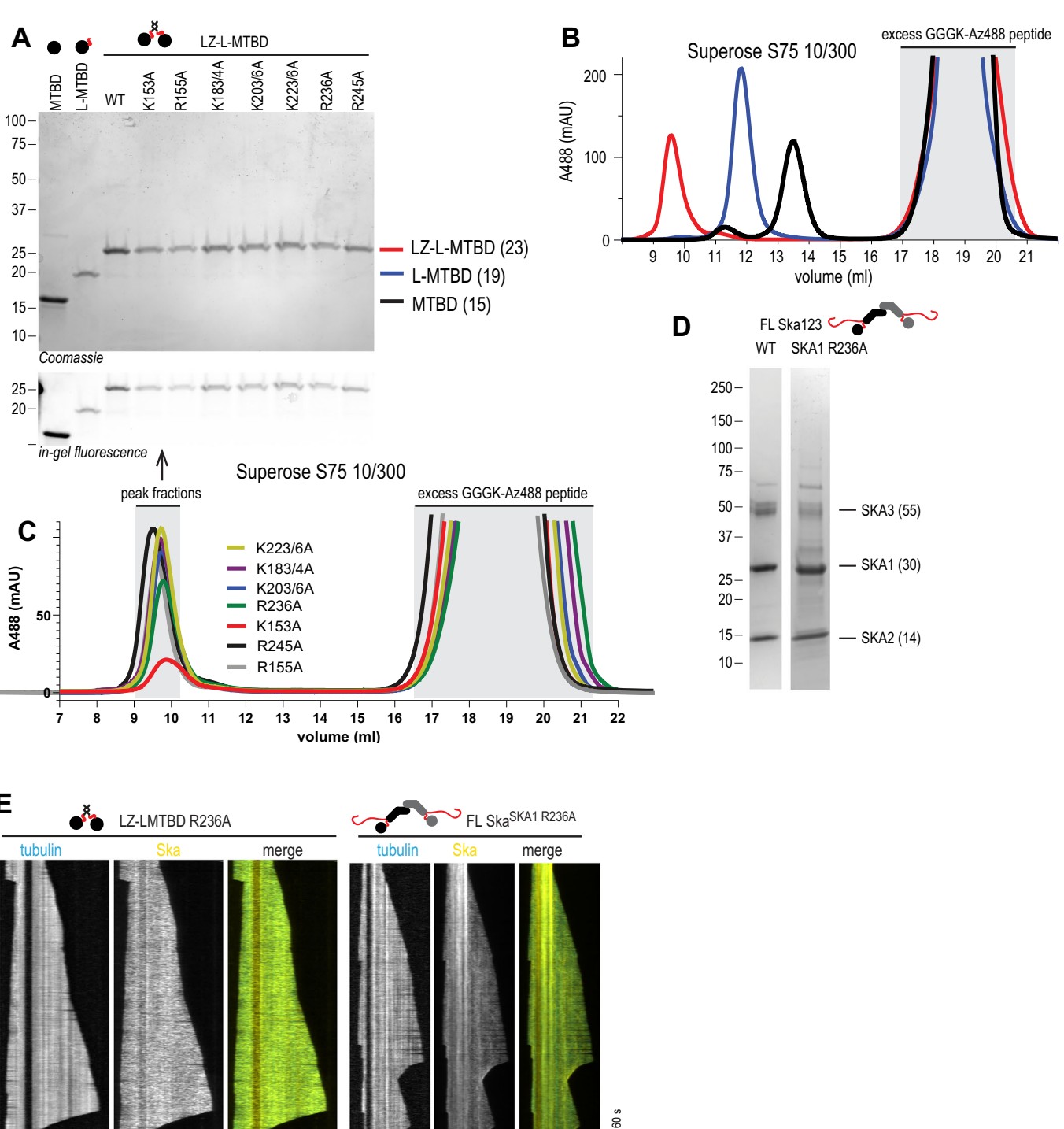

**Figure EV5. Purification of SKA1 fragments, and additional kymographs demonstrating uniform microtubule coating by SKA1 R236A.**

(A) SDS-PAGE of SKA1 MTBD constructs used in the study. (B) SEC profiles of dimeric LZ-LMTBD (red), and monomeric L-MTBD (blue) and MTBD (black) following fluorescent labelling with sortase. (C) SEC profiles of all point mutants of LZ-LMTBD following fluorescent labelling with sortase. (D) SDS-PAGE of FL Ska^SKA1 wt and FL Ska^SKA1 R236A. (E) Additional examples of uniform coating of microtubules by SKA1 LZ-LMTBD R236A and FL Ska^SKA1 R236A.

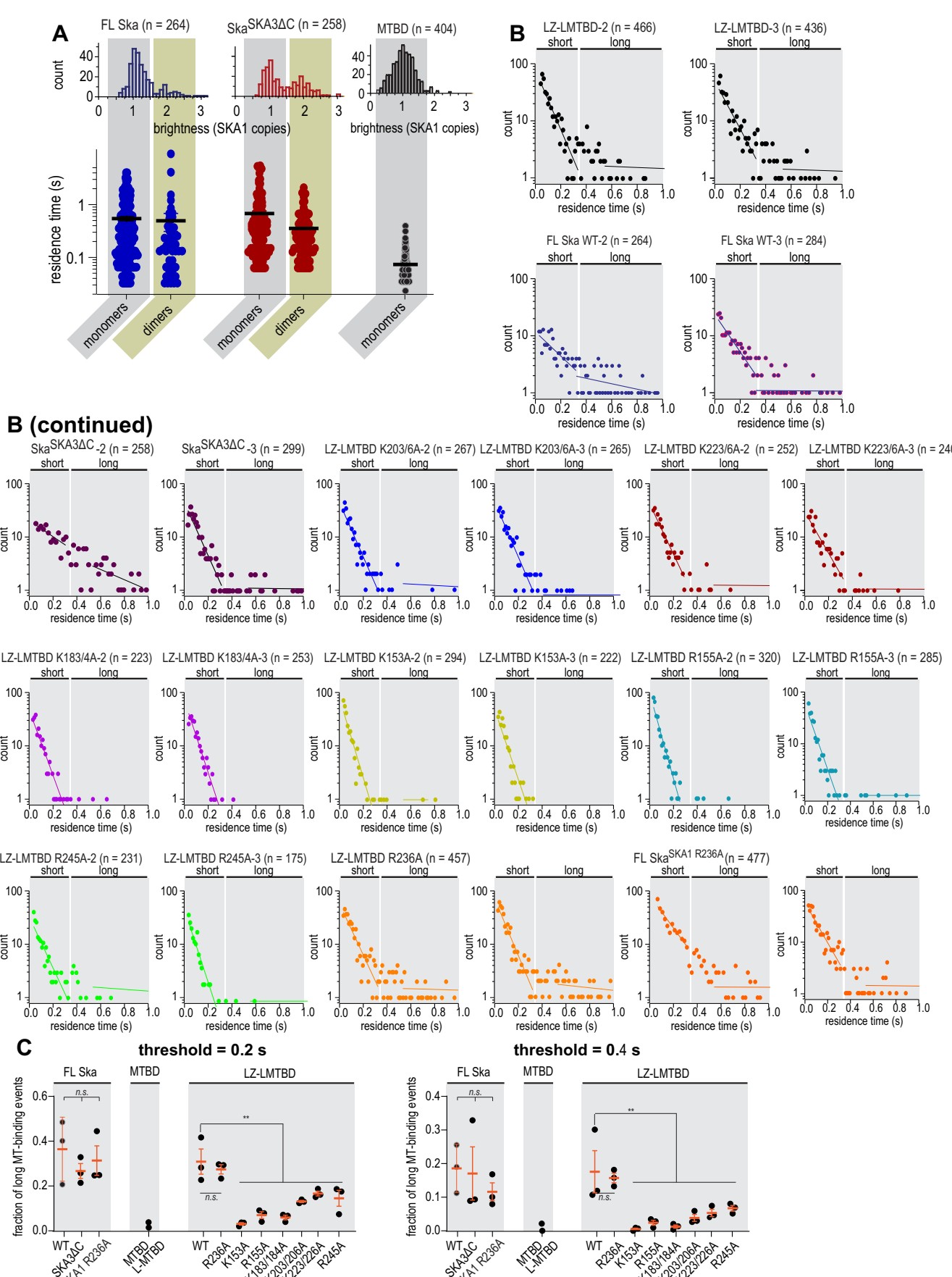

◀ **Figure EV6. Distributions of residence time of SKA mutants on stable microtubules.**

(A) Residence times of FL Ska, Ska$^{SKA3\Delta C}$, and SKA1 MTBD molecules assigned into "monomers" and "dimers" based on their fluorescence intensity (circles), with mean and SEM (lines). FL Ska monomers ($n = 212$) vs dimers ($n = 52$): $p = 0.0988$. Ska$^{SKA3\Delta C}$ monomers ($n = 151$) vs dimers ($n = 102$): $p = 0.0098$. (B) Additional repeats of residence time measurements of constructs indicated. (C) Fraction of long microtubule-binding events of constructs indicated determined at two threshold values for the short/long boundary. Black dots: individual repeats, orange lines: mean and SD. Threshold $= 0.2$ s. Welch's t-test: FL Ska$^{SKA1wt}$ vs FL Ska$^{SKA1R236A}$ $p = 0.65$ (n.s.); LZ-LMTBD wt vs LZ-LMTBD R236A $p = 0.98$ (n.s.); FL vs LZ-LMTBD $p = 0.40$ (n.s.); FL Ska$^{SKA1 R236A}$ vs LZ-LMTBD R236A $p = 0.62$ (n.s.); 1-way ANOVA of LZ-LMTBD wt vs non-R236A mutants $p = 0.0066$ (**). Threshold $= 0.4$ s. Welch's t-test: FL Ska$^{SKA1wt}$ vs FL Ska$^{SKA1R236A}$ $p = 0.24$ (n.s.); LZ-LMTBD wt vs LZ-LMTBD R236A $p = 0.98$ (n.s.); FL vs LZ-LMTBD $p = 0.48$ (n.s.); FL Ska$^{SKA1 R236A}$ vs LZ-LMTBD R236A $p = 0.27$ (n.s.); 1-way ANOVA of LZ-LMTBD wt vs non-R236A mutants $p = 0.0069$ (**). Scale bar: 5 μm (horizontal), 1 s (vertical).

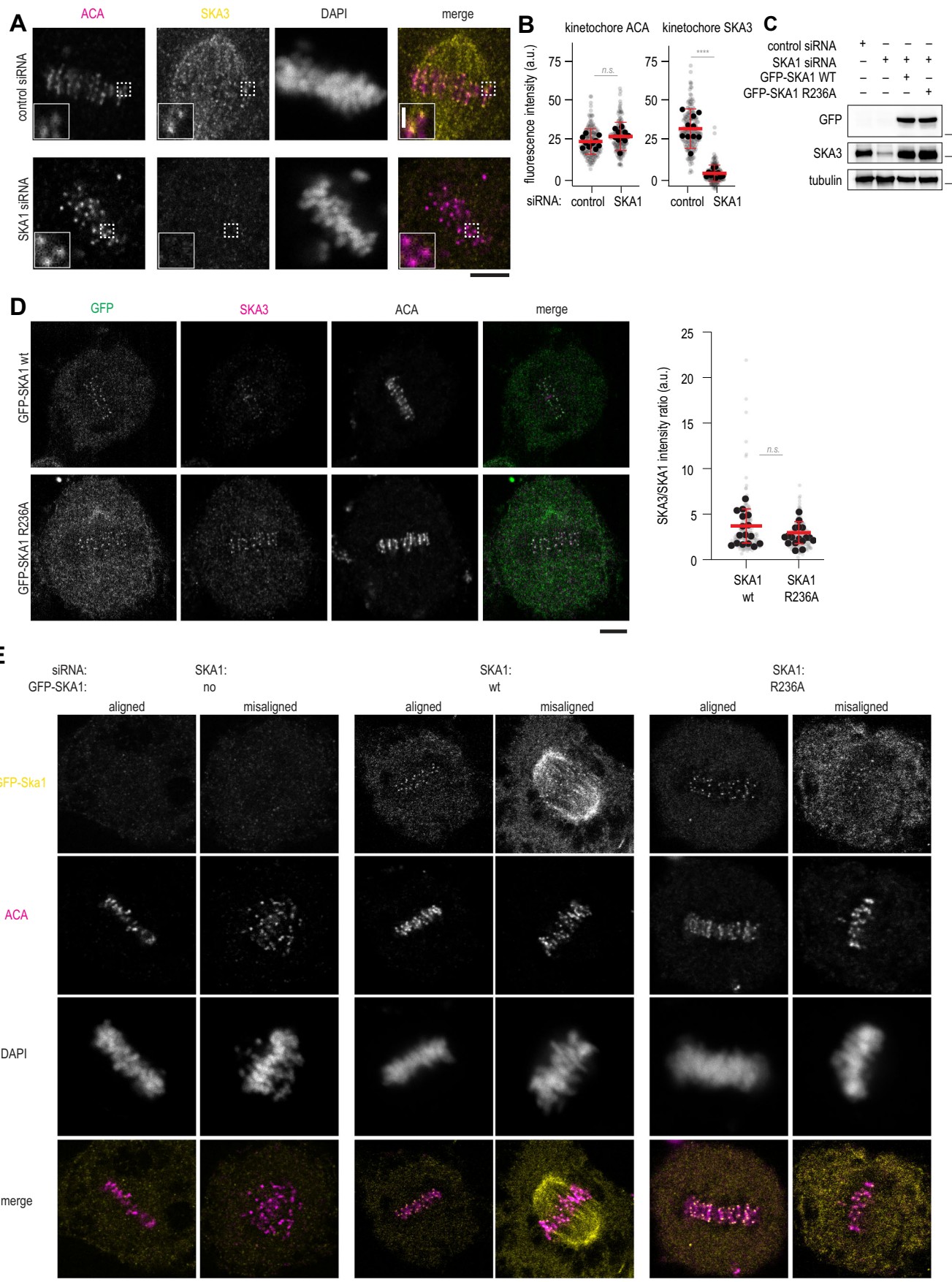

◀ **Figure EV7. Additional characterisation of cells depleted of SKA1 and expressing GFP-SKA1 with and without the R236A mutation.**

(**A**) Single planes from z-stacks of confocal images of cells treated with control siRNA or SKA1 siRNA and stained for ACA (magenta), SKA3 (yellow), and DAPI. (**B**) Quantification of SKA3 fluorescence intensity at kinetochores following treatment with the indicated siRNA. N = at least 20 kinetochores per cell (small symbols), 10 cells per condition (large symbols). Welch's t-test $p$ value: ACA (control vs SKA1 siRNA): 0.0652; SKA3 (control vs SKA1 siRNA): $1.2 \cdot 10^{-6}$ (****). Lines show mean and S.D. (**C**) Western blots probing for GFP, SKA3, and tubulin as a loading control, using cells treated according to conditions indicated at the top of the panel. (**D**) Single planes from z-stacks of confocal images of cells expressing GFP-SKA1 wt or R236A following SKA siRNA treatment, and stained for GFP (green) ACA (magenta), SKA3, and DAPI. The graph shows SKA3 intensity values normalised to SKA1 intensity per kinetochore (grey) or per cell (black). Red lines show mean and SD. N = 248 kinetochores, 16 cells (SKA1 wt); 266 kinetochore, 17 cells (SKA1 R236A). Welch's t-test: 0.1973. (**E**) Examples of cells considered having aligned or misaligned metaphase plates following the treatment indicated and stained for GFP-SKA1, ACA, and DAPI. Scale bars: 5 μm.

