## [Peer Review File · The EMBO Journal]

Microtubule end stabilisation by cooperative oligomers of Ska and Ndc80 complexes

Renjith Radhakrishnan, Lauren Stokes, Matthew Day, Pim Huis in 't Veld, and Vladimir Volkov

Corresponding author(s): Vladimir Volkov (v.volkov@qmul.ac.uk)

Review Timeline:

Submission Date:	15th Jul 25
Editorial Decision:	9th Sep 25
Appeal Received:	19th Sep 25
Editorial Decision:	2nd Oct 25
Revision Received:	4th Jan 26
Editorial Decision:	12th Feb 26
Revision Received:	23rd Feb 26
Accepted:	27th Feb 26

Editor: Hartmut Vodermaier

Transaction Report:

Dr. Vladimir A. Volkov
Queen Mary University of London
Centre for Molecular Cell Biology, School of Biological and Behavioural Sciences
Mile End Rd
E1 4NS
United Kingdom

9th Sep 2025

Re: EMBOJ-2025-121896
Microtubule end stabilisation by cooperative oligomers of Ska and Ndc80 complexes

Dear Vladimir,

Thank you submitting your study on cooperative microtubule binding and stabilization by kinetochore protein complexes for our consideration, and my apologies for the delay in its evaluation. I sent it to three expert referees, whose reports came in while I was on vacation, and upon my return did not immediately find the time to carefully assess the comments.

I have now finally had the chance to discuss the study and the reports with my colleagues, and I am afraid we eventually concluded that we are currently not able to offer publication of this work in The EMBO Journal. As you will see, only referee 2 provides an overall supportive review, whereas neither referees 1 nor 3 find the study publishable at this stage. Although I realize that some of the specific points, e.g. in referee 3's report, could possibly be clarified through additional work, we note that especially referee 1 raises a number of substantive issues and remains generally unconvinced that key conclusions of the study are currently strongly supported by the presented data. I prefer not to repeat all individual points of criticism in detail here, but hope you understand that in light of their well-taken and overriding concerns, I am not in a position to invite (and thus to some degree commit to) a revised manuscript in this case.

Thank you again for having had the opportunity to consider this work for The EMBO Journal; I am sorry that the reports do not allow me to be more positive on this occasion, but hope that you will nevertheless find our referees' comments and suggestions helpful when considering your further proceedings with this work.

With kind regards,

Hartmut

Referee #1:

In the manuscript "Microtubule end stabilisation by cooperative oligomers of Ska and Ndc80 complexes," Radhakrishnan and colleagues characterize the oligomerization behavior of reconstituted Ska and Ndc80 complexes and examine the impact of their oligomerization on microtubule stability. The authors initially demonstrate that addition of purified Ska complexes to low concentrations of Ndc80 complexes (which alone poorly stabilize dynamic microtubules) leads to longer microtubule growth extensions off stable seeds in vitro. This confirms previous studies showing synergistic behavior of these two complexes in promoting microtubule (MT) stability. The authors then visualize Ska and Ndc80 complexes on dynamic MTs using cryo-electron tomography, observing ordered oligomers of Ndc80 complexes in the presence or absence of Ska complexes, as well as disordered clumps of Ska complexes associated with MTs under both conditions. They measure the number of "non-bent" protofilaments per MT plus end under various conditions and report that MTs incubated with either stabilizing concentrations of Ndc80C or sub-stabilizing concentrations of Ndc80C plus Ska complex contain higher numbers of non-bent MT protofilaments at plus ends compared to previously published values for plus-ends of free MTs. These experiments further confirm that Ndc80 and Ska complexes promote MT plus end stability.

In subsequent experiments, the authors investigate Ska complex oligomerization in the absence of Ndc80 complexes. On MTs that are determined to be depolymerizing, Ska densities appear to associate primarily with the plus ends and manifest in cryo-

ET images as large masses whose thickness around the MT surface exceeds the dimensions of a single Ska dimer. The conclusion is that these observed densities represent oligomeric complexes. Using TIRF-M, the authors then employ an "envelope" assay, in which dynamic MTs are incubated with fluorescent Ska complexes in vitro and Ska density along the MT is analyzed. Using full-length, wild-type Ska complexes, higher densities of Ska are observed near the MT growing ends, resulting in non-uniform density along the lattice. The authors hypothesize that this non-uniformity is specifically a property of oligomerized Ska complexes on MTs. Using this as their metric for Ska oligomerization, they describe domains and amino acids required for formation of the "envelope," or the bright region of Ska near the MT plus end. They show that a forced dimer of Ska1 MTBD is sufficient for envelope formation and that mutation of R236 in this domain perturbs its formation. Additionally, using FRAP, the authors report that the Ska3 C-terminus, which is responsible for Ndc80 complex binding, contributes to stabilizing MT-bound Ska oligomers. The authors report that Ska complexes containing the Ska1-R236A mutant retain intact MT binding capacity by analyzing the residence times of the complexes on MTs. Finally, the authors express the Ska1-R236A mutant in human tissue culture cells and, using a cold-stable MT assay, find that kinetochore-MTs are reduced in these cells compared to those expressing WT Ska1.

Overall, this paper presents interesting experiments that support the idea that oligomeric Ska and Ndc80 complexes form on MTs and play a role in stabilizing MT plus ends against depolymerization. However, the major conclusions are not strongly supported by the data. Specifically, the conclusions regarding Ska oligomerization are based on the "envelope" assay, which the authors use as a metric for oligomerization; however, there are no data supporting the premise that changes in Ska density along the lattice are indeed indicators of oligomerization. Additionally, the conclusion that Ska and Ndc80 oligomers promote increased lateral binding between plus-end protofilaments as the mechanism for MT stabilization is not supported by the data presented. These and other points are expanded upon below. In light of these issues, I do not recommend publication in The EMBO Journal.

Major points:

(1) Using TIRF microscopy to analyze Ska complexes on dynamic microtubules, the authors observe that wild-type, full-length Ska complexes decorate MTs in a non-uniform manner, with brighter regions found near the plus ends ("envelopes"). They hypothesize that this non-uniform decoration results from oligomeric Ska complexes on the MTs. While this is a reasonable hypothesis, it is not experimentally tested in the study. Even so, the authors go on to use the "envelope assay" as their metric for the presence or absence of Ska oligomerization. As such, they conclude that certain mutants prevent formation of Ska oligomers or oligomer stabilization and then use these data to define Ska oligomerization domains, which represents a major finding of the paper. However, without direct evidence that envelope formation correlates with oligomerization, these conclusions are not adequately supported by the data.

(2) The authors conclude that Ndc80 and Ska complex oligomerization stabilizes MT plus ends by promoting lateral association of MT protofilaments. This conclusion is drawn, in part, on a previous study by the authors demonstrating that MT plus ends with GTP-bound tubulin exhibit pronounced protofilament clustering, while this clustering is reduced at MT plus ends with GDP-bound tubulin. Since Ska and Ndc80 complexes stabilize MT plus ends, it is expected that these plus ends should exhibit high protofilament clustering, which is observed on stable, growing MTs (Kalutskii et al., 2025). However, the authors do not establish causality: there is no evidence that Ndc80 and Ska complexes actively drive protofilament clustering to promote stability, as opposed to a scenario in which these complexes promote stability through other mechanisms, with clustered protofilaments observed as a downstream consequence. Given that MTs with GTP-bound tubulin naturally exhibit clustering in the absence of external factors, the latter interpretation appears more likely.

(3) Related to the above point: For the protofilament clustering experiments, the authors rely on control data and measurements from their previously published work rather than generating independent controls for this study. Given that protofilament clustering is a newly documented phenomenon, it seems important to carry out independent controls in this new study. Furthermore, the authors do not specify the distance threshold used to define protofilaments as "clustered," or how this was determined. The manuscript would benefit from additional representative images comparing protofilaments that meet their clustering criteria versus those that are adjacent but do not qualify as clustered, as this would help readers evaluate the data.

(4) The EM images are striking; however, they do not necessarily further our understanding of how these complexes functionally impact MT biology or assemble on MTs. A revealing functional experiment would be to assess the formation of Ska oligomers/aggregates in the presence of the anti-oligomerization mutants which would potentially support their findings (see point #1).

(5) In Figure 3, the authors conclude from the EM images that NDC80 CH domains occupy two neighboring protofilaments; however, the resolution of the presented images appears insufficient to support this point.

(6) The FRAP data nicely show that the Ska3 C-term plays a role in stabilizing Ska on MTs (independently of Ndc80). Does the stabilizing domain overlap with the NDC80 binding domain?

(7) In Supp Figure 5A, it is interesting that the dimer population is increased in Ska3 C-term mutant condition compared to WT.

Since the C-term was defined in this paper as a Ska-Ska oligomerization stabilizing region, do the authors have an explanation for the puzzling data in the figure?

(8) Figure 5: The authors are using residence time as correlative indicator of "direct MT binding" but oligomerization can also affect this parameter (e.g. artificially dimerizing NDC80 complexes with an antibody significantly increasing residence time without altering the binding domains). They state that they are using single molecule conditions (a scenario where considering oligomers would not be necessary), but Supp Figure 5A shows that there are both dimers and monomers present under the experimental conditions.

(9) For the cell work in Figure 6, it would be helpful to include some analysis for non-cold treated cells to have some indication of the mitotic phenotype for cells expressing WT Ska1 vs. Ska1 R236A with no perturbation (possibly chromosome alignment phenotype, timing through mitosis, or chromosome segregation errors). Related to this, the supplemental data for this figure are somewhat confusing. The figure shows two very different phenotypic outcomes in cells presumably depleted of Ska1 and not rescued. Which is more frequently observed? Does one have stable attachments and the other doesn't? Do either of these phenocopy the Ska1 R236A mutant? How, overall, do the phenotypes of Ska1 depletion compare to expression of the mutant?

(10) In Figure 7, the authors state that for the 1 nM and 10 nM Ska conditions (with either 0, 0.3, or 1 nM Ndc80 added), the fractions of seeds with extensions are not significantly different from each other, but both are significantly different from the "no Ska added" condition. It is a little hard to see how the 0 vs. 10 nM Ska conditions are different. It would be helpful to add the calculated significance scores to these.

Minor points:

(1) The rationale for the experiments is not altogether clear. In the introduction, the authors state that "...it is currently unclear whether full-length Ska and Ndc80 can spontaneously oligomerize on microtubules, or they require externally imposed oligomerisation." Is the question if the complexes can oligomerize together in the absence of the NDC80 kinetochore recruitment factors (e.g. is this the "externally imposed oligomerization")? Or, if Ska alone can oligomerize independently of NDC80? It would be helpful if the issue being addressed in the paper was a bit clearer for the reader.

(2) Figure 2G: I think the significance values are switched (or, alternatively, the description of results doesn't match the figure).

(3) In Figure 1E, why does 1 nM Ska enhance MT stabilization with 0.3 nM NDC80C better than 10 nM?

(4) Line 235: bot should be "both"

(5) Figure 5: There is no "E" label on the figure

(6) Line 298 text: I think this should be the subheading for this section. (Oligomerisation-deficient Ska has intact microtubule binding in single-molecule conditions).

(7) In Figure 5, it would be helpful to compare the data for WT, FL Ska and the FL Ska R236 mutant, with the data side-by-side. The WT FL data are included in Panel C, but is tough to see what is going on with the overlaying plots and data points. Also, in Supplement for Figure 5, the replicates for FL Ska are displayed with a single curve fit while the mutant FL R236A is shown with fits for both the long and short. Was this intentional?

(8) In Figure 7C, the blue vs. green dots are difficult to distinguish on the graphs.

Referee #2:

In this manuscript, Radhakrishnan and colleagues combined in-vitro reconstitution assays with cryoET and TIRF microscopy to address the individual and synergistic effects of Ndc80 and Ska complexes in the stabilization of microtubule plus-ends. The authors showed that neither Ska alone nor non-stabilizing concentration of Ndc80 (<1nM) can prevent the microtubule plus-end shortening, but that combining Ska with non-stabilizing concentration of Ndc80 successfully stabilized microtubule plus-ends. Based on these results and the visualization of Ndc80 and Ska molecules along the neighboring microtubule protofilaments, they propose that the Ndc80-Ska oligomers facilitate lateral contacts between microtubule protofilaments, thereby protecting the microtubule plus-ends from depolymerization. Using site-directed mutagenesis, they also identified a residue of SKA1 (R236) required for Ska oligomerization and showed both in-cellulo and in-vitro that the oligomerization of Ska cooperates with Ndc80 to maintain stable kinetochore-microtubule attachments. In addition to the contribution of the Ndc80:Ndc80 interactions in the stabilization of the kinetochore-microtubule attachments, this study highlights the importance of Ndc80:Ska and Ska:Ska interactions in this important process and provides molecular insight into these interactions. By elucidating the role of these

interactions in facilitating lateral contacts between microtubule protofilaments, the study provides a novel insight into how microtubule plus-ends are stabilized at kinetochores. The authors have clearly demonstrated their expertise in the in-vitro reconstitution of kinetochore-microtubule interactions. The manuscript is well-written, and the study is overall well-designed, convincingly executed and properly controlled. I find it suitable for broad readership of the EMBO Journal and support its publication. Below are a few comments that the authors may consider implementing.

Major points:

1) It was shown before that SKA3 phosphorylation at T358 and T360 is required for Ndc80:Ska complex formation. The authors used both untreated Ska FL and in-vitro phosphorylated Ska FL to show that these Ska molecules cooperate with Ndc80 to stabilize microtubule plus-ends by facilitating lateral interactions between microtubule protofilaments. An additional control that the authors could use to strengthen their claims that Ndc80:Ska oligomerize to facilitate microtubule stabilization would be to replace SKA3 WT with SKA3 T358A and T360A mutants within the Ska complex and then assess its ability to cooperate with non-stabilizing concentrations of Ndc80 to stabilize the microtubule plus-ends.

2) Even though both GFP-SKA1 WT and GFP-SKA1 R236A correctly localized at kinetochores, the cold-treated GFP-SKA1 R236A-expressing cells displayed a strong reduction in kinetochore-microtubule attachments. Taking their results together, the authors conclude that this is due to the inability of SKA1 R236A to form oligomers. But is the SKA1 R236A-containing Ska complex functional? The authors could provide SKA3 immunofluorescence staining in GFP-SKA1 WT and GFP-SKA1 R236A cells to partially address this question.

Minor points:

- 1) Line 167: it should read "we" instead of "were".
- 2) Line 188: remove "of" at the end.
- 3) Line 235: it should read "both" instead of "bot".
- 4) Line 336: it should read "depletion" instead of "depeletion".
- 5) Line 472: it should read "Cells" instead of "Cell".

Referee #3:

Accurate segregation of aligned chromosomes during mitosis is essential for eukaryotic cells and ensured by the coordinated action of Ndc80 and Ska proteins. In this study, Radhakrishnan et al. investigated the function of the Ndc80-Ska complex using cryo-electron tomography (cryo-ET), total internal reflection fluorescence microscopy (TIRFM), and in vivo fluorescence imaging. Radhakrishnan et al. found that Ndc80-Ska complex stabilizes the microtubule plus ends by reinforcing the lateral interaction between protofilaments. Their analyses further revealed that the oligomerization of Ska molecules is mediated by the interaction of SKA1 microtubule binding domain (MTBD) and C-terminal tail and the Ska proteins form an envelope-like distribution on microtubules. The authors also identified a residue in SKA1 MTBD which is important for Ska oligomerization but dispensable for microtubule binding. Mutation of this residue in Ska hinders the stable kinetochore-microtubule attachment in cold conditions in vivo as well as microtubule end-tracking ability of Ska complex in vitro. Overall, this research provides insights into the microtubule stabilizing mechanism in mitosis with thorough analyses and proposes the possible role of Ska molecule in stabilizing microtubule ends. However, there are some fundamental issues in the current manuscript, and therefore it cannot be published in its current form. This reviewer thinks that the manuscript could be suitable for publication in The EMBO Journal, provided the authors appropriately address the points listed below.

Major comments:

In Supplementary Figure 1C and D, the authors claim that the hyperphosphorylation of Ska did not affect the fraction of seeds with extensions. However, the lengths of microtubules in the condition of untreated Ska appear shorter than the hyperphosphorylated condition in Supplementary Fig. 1D. Thus, even the fraction of extensions might be the same, hyperphosphorylation might have effects on the degree of extensions. Did the authors compare the lengths of extensions between these conditions?

In this work, the authors were not able to show the clear structure of Ska complex bound on the microtubule. There is, however, a previous report using cryo-ET to visualize each Ska molecule bound to the microtubule (Janczyk et al., 2017, "Mechanism of Ska Recruitment by Ndc80 Complexes to Kinetochores, Developmental Cell). This paper is not cited in this manuscript. The authors are encouraged to discuss the reason why similar structures were not obtained in this study with citing the paper.

In Fig. 2D, it appears that a protofilament sheet is covering a microtubule (MT1) and forms another microtubule (MT2), but this is not mentioned in the main text. Is this something often observed with Ndc80? What is the possible mechanism of the covering of the microtubule and branching? Please describe.

In the graph in Fig. 4D, the authors compare the recovery of fluorescence of SkaFL and SkaSka3ΔC in the graph. However, only the kymograph of SkaFL is shown. For completeness and to allow readers to directly compare the two conditions, it would

be helpful to also include a representative kymograph for SkaSka3ΔC.

In the graph in Fig. 4H, there are some constructs with data from a single experiment without repeated experiments. The authors are encouraged to check at least if these results are reproducible.

Regarding Fig. 5, there are several issues that need to be addressed. First, in Fig. 5E, several constructs only have data without repeated experiments and included in the statistical analysis. This case is more serious since the values of constructs like R245A and K223/226A are comparable to some data points of FL Ska constructs. The authors need to perform three independent replicates since this directly affects the conclusion that R236A is the only point mutation that did not affect the long binding events.

Furthermore, the binding events up to 1.0 s are plotted in the graphs in Fig. 5C and D. In the kymographs in Fig. 5B, however, there are some binding events clearly longer than 1 second. The details of the analysis are not described, either. Please describe how binding events exceeding 1 second were treated.

Another problem with Fig. 5C and D is that, although the short events are indicated as <0.3 s and long events as >0.3 s, the boundaries of shaded areas in the graphs are set around 0.38 s on the X-axis. This must be fixed. Furthermore, in the main text, short events are defined as {less than or equal to}0.3s (Line 314). Please clarify whether the definition is <0.3 s or {less than or equal to}0.3 s.

The authors are also saying in the main text that "the residence time of SKA1 MTBD never exceeded 0.4 s (Figure 5B, C)" (Line 307-308) and "the fraction of residence times that were never or very rarely observed for MTBD/LMTBD (long events, >0.3s, Figure 5C)". However, there is clearly one MTBD data point that exceeds 0.4 s in Fig. 5C. In fact, the data presented in Fig. 5C labeled as MTBD (n = 404) looks a lot like LMTBD (n=243) data in Fig. S5B rather than MTBD data (n = 404) in Fig. S5B. Please check if the authors have used the correct dataset when preparing Fig. 5C.

Minor comments:

In contrast to the figures where superscripts are used for the construct names, such as SkaSKA3ΔC and FL Ska SKA1 R236A, they are not shown as superscripts in the main text as well as in the figure legends. Furthermore, in all cases of MgCl₂, the "2" is not in subscript. It seems that this happened when the authors were preparing the manuscript file for the submission. Please check if these are as the authors intended.

For the readers who are not familiar with Ska-Ndc80, the models in Fig. 1B can be improved. First, RWD domain should be indicated in the Ndc80 complex. Second, the schematic of Ska complex is too simplified, and it's hard to understand which part corresponds to SKA1, 2 and 3, respectively. Please indicate which part correspond to which region as well as phosphorylation sites.

The authors used many different truncated constructs as well as point mutations to reveal functions of Ska complex. It's easy to lose track which region and/or residues are important for which functions. Nevertheless, there is only a summary figure of whole Ska complex and Ndc80 in Fig. 7D. Adding a summary figure focusing on Ska complex's domains and residues would be helpful for the readers.

Fig. 1E legend, "1nM Ska" and "10nM Ska": there should be spaces between numbers and units.

Fig. 3E is not cited in the main text. It should be mentioned.

Fig. 7B legend, "1 nN Ska": this should be 1 nM Ska.

Line 20: "cryoET" should be defined as an abbreviation when first mentioned.

Line 52 "Yatskevich et al., 2023": did the authors mean "Yatskevich et al., 2024" as in the references?

Line 147 "p = 0.0006 for the effect of Ska concentration": this seems to contradict the value in the legend of Fig. 1E (p = 0.0012).

Line 182: "TIRF" should be defined as an abbreviation when first mentioned.

Line 188 "and found a similar compacted spacing of (Kalutskii et al., 2025) (Supplementary Figure 2F).": the word "of" looks unnecessary here.

Line 235 "bot": this should be "both".

Line 235 "in which Ndc80 (1nM)": it lacks space between 1 and nM.

Line 244 "envelopes" (Siahaan et al., 2019): the term "envelope" is not used in Siahaan et al., (2019). Did the author intend to cite Siahaan et al., 2022, "Microtubule lattice spacing governs cohesive envelope formation of tau family proteins", Nat. Chem. Biol.?

Line 356 "and found that 1 nM FL SkaSKA1 R236, in presence of 1 nM Ndc80, failed to stabilise microtubules against disassembly": "SkaSKA1 R236" should be "SkaSKA1 R236A".

Line 349: ACA should be defined as an abbreviation when first mentioned.

Line 350: Figure 6D should be Figure 6C.

Line 369: the term "tremendously" is highly subjective and should be avoided.

Line 518 "in an approximate ration of": should be ratio instead of ration.

Line 528 "e. coli BL21(DE3) Rosetta cells": it should be fixed to E. coli with capital E.

Line 651 "4.5g/L Glucose, 2mM L-Glutamine": spaces are necessary between numbers and units.

Line 653: it seems that CO₂'s O is written as 0 here. Furthermore, the "2" should be subscript.

In the "Cloning, expression, and purification of SKA1 fragments." section of Materials and Methods, all the temperature conditions are written as "C" instead of "{degree sign}C". All of them should be fixed.

From line 557-574, the temperature conditions are written as "C" instead of "{degree sign}C" again.

*** As a service to authors, The EMBO Journal offers the possibility to directly transfer declined manuscripts to another EMBO Press title (EMBO Reports, EMBO Molecular Medicine, Molecular Systems Biology) or to the open access journal Life Science Alliance launched in partnership between EMBO Press, Rockefeller University Press and Cold Spring Harbor Laboratory Press. The full manuscript (including reviewer comments, where applicable and if chosen) will be automatically forwarded to the receiving journal, to allow for fast handling and a prompt decision on your manuscript. For more details of this service, and to transfer your manuscript to another EMBO title please follow this link:
Link Not Available

Point-by-point response/revision plan. Responses to reviewers' comments are added in blue font.

Referee #1:

In the manuscript "Microtubule end stabilisation by cooperative oligomers of Ska and Ndc80 complexes," Radhakrishnan and colleagues characterize the oligomerization behavior of reconstituted Ska and Ndc80 complexes and examine the impact of their oligomerization on microtubule stability. The authors initially demonstrate that addition of purified Ska complexes to low concentrations of Ndc80 complexes (which alone poorly stabilize dynamic microtubules) leads to longer microtubule growth extensions off stable seeds in vitro. This confirms previous studies showing synergistic behavior of these two complexes in promoting microtubule (MT) stability. The authors then visualize Ska and Ndc80 complexes on dynamic MTs using cryo-electron tomography, observing ordered oligomers of Ndc80 complexes in the presence or absence of Ska complexes, as well as disordered clumps of Ska complexes associated with MTs under both conditions. They measure the number of "non-bent" protofilaments per MT plus end under various conditions and report that MTs incubated with either stabilizing concentrations of Ndc80C or sub-stabilizing concentrations of Ndc80C plus Ska complex contain higher numbers of non-bent MT protofilaments at plus ends compared to previously published values for plus-ends of free MTs. These experiments further confirm that Ndc80 and Ska complexes promote MT plus end stability.

In subsequent experiments, the authors investigate Ska complex oligomerization in the absence of Ndc80 complexes. On MTs that are determined to be depolymerizing, Ska densities appear to associate primarily with the plus ends and manifest in cryo-ET images as large masses whose thickness around the MT surface exceeds the dimensions of a single Ska dimer. The conclusion is that these observed densities represent oligomeric complexes. Using TIRF-M, the authors then employ an "envelope" assay, in which dynamic MTs are incubated with fluorescent Ska complexes in vitro and Ska density along the MT is analyzed. Using full-length, wild-type Ska complexes, higher densities of Ska are observed near the MT growing ends, resulting in non-uniform density along the lattice. The authors hypothesize that this non-uniformity is specifically a property of oligomerized Ska complexes on MTs. Using this as their metric for Ska oligomerization, they describe domains and amino acids required for formation of the "envelope," or the bright region of Ska near the MT plus end. They show that a forced dimer of Ska1 MTBD is sufficient for envelope formation and that mutation of R236 in this domain perturbs its formation. Additionally, using FRAP, the authors report that the Ska3 C-terminus, which is responsible for Ndc80 complex binding, contributes to stabilizing MT-bound Ska oligomers. The authors report that Ska complexes containing the Ska1-R236A mutant retain intact MT binding capacity by analyzing the residence times of the complexes on MTs. Finally, the authors express the Ska1-R236A mutant in human tissue culture cells and, using a cold-stable MT assay, find that kinetochore-MTs are reduced in these cells compared to those expressing WT Ska1.

Overall, this paper presents interesting experiments that support the idea that oligomeric Ska and Ndc80 complexes form on MTs and play a role in stabilizing MT plus ends against depolymerization. However, the major conclusions are not strongly supported by the data. Specifically, the conclusions regarding Ska oligomerization are based on the "envelope" assay, which the authors use as a metric for oligomerization; however, there are no data supporting the premise that changes in Ska density along the lattice are indeed indicators of oligomerization. Additionally, the conclusion that Ska and Ndc80 oligomers promote increased lateral binding between plus-end protofilaments as the mechanism for MT stabilization is not supported by the data presented. These and other points are expanded upon below. In light of these issues, I do not recommend publication in The EMBO Journal.

We thank the reviewer for a positive evaluation of the relevance of our experimental data. Below we propose a set of additional experiments aimed at providing stronger support for the statements in the paper.

Major points:

(1) Using TIRF microscopy to analyze Ska complexes on dynamic microtubules, the authors observe that wild-type, full-length Ska complexes decorate MTs in a non-uniform manner, with brighter regions found near the plus ends ("envelopes"). They hypothesize that this non-uniform decoration results from oligomeric Ska complexes on the MTs. While this is a reasonable hypothesis, it is not experimentally tested in the study. Even so, the authors go on to use the "envelope assay" as their metric for the presence or absence of Ska oligomerization. As such, they conclude that certain mutants prevent formation of Ska oligomers or oligomer stabilization and then use these data to define Ska oligomerization domains, which represents a major finding of the paper. However, without direct evidence that envelope formation correlates with oligomerization, these conclusions are not adequately supported by the data.

We thank the reviewer for raising this important point. To further support our interpretation that formation of non-uniform microtubule decoration by Ska correlates with Ska oligomerisation, we will perform cryoET on microtubules decorated with Ska^{SKA1 R236}, a mutant with disrupted ability to produce non-uniform MT decoration. We will analyse the tomograms for dimensions of Ska coating on microtubules, directly comparing them to the data currently available for oligomerisation-competent WT Ska (Figure 3C-F).

(2) The authors conclude that Ndc80 and Ska complex oligomerization stabilizes MT plus ends by promoting lateral association of MT protofilaments. This conclusion is drawn, in part, on a previous study by the authors demonstrating that MT plus ends with GTP-bound tubulin exhibit pronounced protofilament clustering, while this clustering is reduced at MT plus ends with GDP-bound tubulin. Since Ska and Ndc80 complexes stabilize MT plus ends, it is expected that these plus ends should exhibit high protofilament clustering, which is observed on stable, growing MTs (Kalutskii et al., 2025). However, the authors do not establish causality: there is no evidence that Ndc80 and Ska complexes actively drive protofilament clustering to promote stability, as opposed to a scenario in which these complexes promote stability through other mechanisms, with clustered protofilaments observed as a downstream consequence. Given that MTs with GTP-bound tubulin naturally exhibit clustering in the absence of external factors, the latter interpretation appears more likely.

We thank the reviewer for pointing out the relationship between the dynamic state of a microtubule end and the lateral clustering of the protofilaments. To establish whether Ndc80 and Ska stabilise microtubule ends via protofilament clustering, or protofilament clustering is a consequence of a different stabilisation mechanism, one would require to design and validate an experimental assay where protofilaments are prevented from interacting laterally, and probe whether Ndc80/Ska decoration restores these lateral interactions. This is exactly the experimental strategy we present in the current Figure 2: microtubules are forced to depolymerise by dilution of tubulin, a treatment we previously established to produce protofilament singles and doubles (Kalutskii et al., 2025). We further show that presence of Ndc80 or Ndc80/Ska oligomers results in increased fraction of protofilament clusters of size ≥ 2 , and a reduction in the fraction of single protofilaments.

What is the exact mechanism by which Ndc80 or Ndc80/Ska oligomers induce protofilaments to cluster is a question that so far has no clear answer, since our manuscript presents the first evidence of this effect being induced in microtubule ends by a protein complex bound to it. One hypothesis

could be that binding of Ndc80 may expand tubulin lattice, similarly to kinesin-1 (Shima et al., 2018; Peet et al., 2018), or taxol (Siahaan et al., 2022; Prota et al., 2023). Finding optimal conditions for each of these molecules to stabilise dynamic microtubule ends against shortening on EM grids, performing cryoET and image processing, and manually segmenting protofilament shapes falls outside of the scope of the current paper.

Data currently included in the manuscript argue against tubulin expansion mechanism though: microtubule plus-ends decorated with Ndc80 clusters had a lattice spacing that was indistinguishable from the same parameter in the GDP tubulin in the microtubule ends that were freely shortening (Supplementary Figure 2F). This conclusion will be reinforced by analysis of additional controls described below in the response to point 3. We will perform new control experiments discussed in detail below in point 3, and analyse them the same way as the data currently in the paper, for local lattice spacing near the plus-end, and for protofilament clustering.

(3) Related to the above point: For the protofilament clustering experiments, the authors rely on control data and measurements from their previously published work rather than generating independent controls for this study. Given that protofilament clustering is a newly documented phenomenon, it seems important to carry out independent controls in this new study. Furthermore, the authors do not specify the distance threshold used to define protofilaments as "clustered," or how this was determined. The manuscript would benefit from additional representative images comparing protofilaments that meet their clustering criteria versus those that are adjacent but do not qualify as clustered, as this would help readers evaluate the data.

We will provide two new control datasets to reinforce our conclusions: a) microtubules shortening in absence of Ndc80 or Ska, reproducing data in Kalutskii et al., 2025; and b) microtubules shortening in presence of low, non-stabilising concentration of Ndc80 (1 nM). Regarding the important question raised about the definition of "cluster": in the original paper describing protofilament clustering, Kalutskii et al., 2025 tested a range of distance thresholds to classify protofilaments into "single" or "clustered" states. They found that irrespective of the threshold used, GTP/growing microtubule ends always produced a statistically significantly higher percentage of clustered protofilaments than GDP/shortening microtubule ends. To reinforce our conclusions, we will perform a similar analysis to determine whether the outcome of our comparison of protofilament clustering in absence or presence of Ndc80 or Ndc80/Ska oligomers depends on the distance threshold used.

(4) The EM images are striking; however, they do not necessarily further our understanding of how these complexes functionally impact MT biology or assemble on MTs. A revealing functional experiment would be to assess the formation of Ska oligomers/aggregates in the presence of the anti-oligomerization mutants which would potentially support their findings (see point #1).

To address this important point, we will perform envelope formation assays using TIRF microscopy of WT Ska labelled with AzDye488, and oligomerisation-deficient Ska^{SKA1 R236A} labelled with TMR, added in various ratios. This way we will titrate the effect of the presence of the mutant on the oligomerisation/envelope formation of the WT Ska.

(5) In Figure 3, the authors conclude from the EM images that NDC80 CH domains occupy two neighboring protofilaments; however, the resolution of the presented images appears insufficient to support this point.

We appreciate the reviewer's concern about resolution in our EM images. Our 3D class averages have resolution of ca. 20Å (as judged by the gold-standard 0.143 FSC criterion applied to post-processed half-maps in relion 5.0). This resolution is sufficient to distinguish individual Ndc80

molecules occupying neighbouring tubulin monomers in the microtubule lattice (41-43Å apart). Individual adjacent protofilaments in a microtubule can be reliably distinguished even without averaging. To further support our conclusion that Ndc80 occupies adjacent protofilaments, we are including 4-nm thick summed cross-sections through four individual protofilaments in our 3D class average of a microtubule decorated with 10 nM Ndc80 (Figure 3A), clearly showing Ndc80 CH-domain density on protofilaments 1 and 2, but not protofilaments 0 and 3:

Rebuttal Figure 1. Sum projections through 4nm-thick slices through the 3D class average. Top row, projection along the microtubule, with coloured lines showing corresponding projections across the microtubule shown below. Only PFs 1 and 2 show a clear Ndc80 CHD density.

In addition, we provide three videos showing non-averaged 0.8-nm slices from three denoised tomograms, with Ndc80 oligomer densities on adjacent protofilaments. These data will be used to strengthen our conclusions in the revised manuscript. We will also upload 3D class averages we obtained to EMDB and provide accession numbers.

(6) The FRAP data nicely show that the Ska3 C-term plays a role in stabilizing Ska on MTs (independently of Ndc80). Does the stabilizing domain overlap with the NDC80 binding domain?

To the best of the authors' knowledge, the precise region of SKA3 interacting with Ndc80 is currently not reported in the literature. SKA3 T358 and T360 phosphorylated by Cdk1 were reported to be essential for this binding (Zhang et al., 2017, Huis in 't Veld et al., 2019). To address the point raised by the reviewer, we will perform the FRAP assay using Ska^{SKA3 T358A/T360A}, a mutant that was previously characterised in vitro (Huis in 't Veld et al., 2019), and which we will also use to address point 1 raised by Reviewer 2 (see below).

(7) In Supp Figure 5A, it is interesting that the dimer population is increased in Ska3 C-term mutant condition compared to WT. Since the C-term was defined in this paper as a Ska-Ska oligomerization stabilizing region, do the authors have an explanation for the puzzling data in the figure?

We thank the reviewer for noticing a slight increase in the percentage of dimers in the case of Ska^{SKA3ΔC}. However, this difference did not appear statistically significant, and thus we did not pursue an explanation. Please also see our response to the next point raised.

(8) Figure 5: The authors are using residence time as correlative indicator of "direct MT binding" but oligomerization can also affect this parameter (e.g. artificially dimerizing NDC80 complexes with an antibody significantly increasing residence time without altering the binding domains). They state that they are using single molecule conditions (a scenario where considering oligomers would not be necessary), but Supp Figure 5A shows that there are both dimers and monomers present under the experimental conditions.

We thank the reviewer for pointing out the existence of both monomers and dimers of Ska in our experimental conditions. We have previously tested whether dimers of Ska have longer residence time than monomers, and the results did not support this hypothesis, both for FL Ska and for Ska^{SKA3ΔC}. We either see no change in residence time for both species of FL Ska ($p = 0.0988$), or a reverse effect, namely shorter residence time for dimers of Ska^{SKA3ΔC} (see the figure below). These data supporting our decision to treat monomers and dimers as a single-molecule species with the similar residence time will be included in the revised manuscript.

Rebuttal Figure 2. Residence times of FL Ska and Ska^{SKA3ΔC} molecules assigned into “monomers” and “dimers” based on their fluorescence intensity (circles), with mean and SEM (lines). FL Ska monomers ($n = 212$) vs dimers ($n = 52$): $p = 0.0988$. Ska^{SKA3ΔC} monomers ($n = 151$) vs dimers ($n = 102$): $p = 0.0098$.

(9) For the cell work in Figure 6, it would be helpful to include some analysis for non-cold treated cells to have some indication of the mitotic phenotype for cells expressing WT Ska1 vs. Ska1 R236A with no perturbation (possibly chromosome alignment phenotype, timing through mitosis, or chromosome segregation errors).

We thank the reviewer for raising this important point. SKA1 R236A mutant has been extensively characterised in two independent studies, which we discuss in the manuscript, Abad et al., 2014 and Monda et al., 2017. In particular both studies report increased mitotic timing for R236A and R236/245A. In addition, Monda et al. report a slight increase in major chromosome alignment defects for the R236A mutation. Given that these data are already available in the literature, we refrained from reproducing them. We will however perform and report additional validation experiments to determine if the R236A mutation causes chromosome alignment phenotypes. Please see also our response to a related point 2 raised by the Reviewer 2.

Related to this, the supplemental data for this figure are somewhat confusing. The figure shows two very different phenotypic outcomes in cells presumably depleted of Ska1 and not rescued. Which is more frequently observed? Does one have stable attachments and the other doesn't? Do either of these phenocopy the Ska1 R236A mutant? How, overall, do the phenotypes of Ska1 depletion compare to expression of the mutant?

We will provide quantitative analysis of the phenotypes of cells depleted of SKA1, and their comparison to cells expressing SKA1 R236A.

(10) In Figure 7, the authors state that for the 1 nM and 10 nM Ska conditions (with either 0, 0.3, or 1 nM Ndc80 added), the fractions of seeds with extensions are not significantly different from each other, but both are significantly different from the “no Ska added” condition. It is a little hard to see

how the 0 vs. 10 nM Ska conditions are different. It would be helpful to add the calculated significance scores to these.

The data for various concentrations of Ska in absence of Ndc80 are plotted twice: in Figure 7C, but also Figure 1E, where the difference is more clear. We will add significance scores, and refer to both figures to clarify the difference.

Minor points:

(1) The rationale for the experiments is not altogether clear. In the introduction, the authors state that "...it is currently unclear whether full-length Ska and Ndc80 can spontaneously oligomerize on microtubules, or they require externally imposed oligomerisation." Is the question if the complexes can oligomerize together in the absence of the NDC80 kinetochore recruitment factors (e.g. is this the "externally imposed oligomerization")? Or, if Ska alone can oligomerize independently of NDC80? It would be helpful if the issue being addressed in the paper was a bit clearer for the reader.

We will provide a more focused and streamlined description of rationale in the revised manuscript.

(2) Figure 2G: I think the significance values are switched (or, alternatively, the description of results doesn't match the figure).

(3) In Figure 1E, why does 1 nM Ska enhance MT stabilization with 0.3 nM NDC80C better than 10 nM?

(4) Line 235: bot should be "both"

(5) Figure 5: There is no "E" label on the figure

(6) Line 298 text: I think this should be the subheading for this section. (Oligomerisation-deficient Ska has intact microtubule binding in single-molecule conditions).

(7) In Figure 5, it would be helpful to compare the data for WT, FL Ska and the FL Ska R236 mutant, with the data side-by-side. The WT FL data are included in Panel C, but is tough to see what is going on with the overlaying plots and data points. Also, in Supplement for Figure 5, the replicates for FL Ska are displayed with a single curve fit while the mutant FL R236A is shown with fits for both the long and short. Was this intentional?

(8) In Figure 7C, the blue vs. green dots are difficult to distinguish on the graphs.

We are grateful to the reviewer for noting typos and inconsistencies, which will all be corrected in the revised manuscript.

Referee #2:

In this manuscript, Radhakrishnan and colleagues combined in-vitro reconstitution assays with cryoET and TIRF microscopy to address the individual and synergistic effects of Ndc80 and Ska complexes in the stabilization of microtubule plus-ends. The authors showed that neither Ska alone nor non-stabilizing concentration of Ndc80 (<1nM) can prevent the microtubule plus-end shortening, but that combining Ska with non-stabilizing concentration of Ndc80 successfully stabilized microtubule plus-ends. Based on these results and the visualization of Ndc80 and Ska molecules along the neighboring

microtubule protofilaments, they propose that the Ndc80-Ska oligomers facilitate lateral contacts between microtubule protofilaments, thereby protecting the microtubule plus-ends from depolymerization. Using site-directed mutagenesis, they also identified a residue of SKA1 (R236) required for Ska oligomerization and showed both in-cellulo and in-vitro that the oligomerization of Ska cooperates with Ndc80 to maintain stable kinetochore-microtubule attachments. In addition to the contribution of the Ndc80:Ndc80 interactions in the stabilization of the kinetochore-microtubule attachments, this study highlights the importance of Ndc80:Ska and Ska:Ska interactions in this important process and provides molecular insight into these interactions. By elucidating the role of these interactions in facilitating lateral contacts between microtubule protofilaments, the study provides a novel insight into how microtubule plus-ends are stabilized at kinetochores. The authors have clearly demonstrated their expertise in the in-vitro reconstitution of kinetochore-microtubule interactions. The manuscript is well-written, and the study is overall well-designed, convincingly executed and properly controlled. I find it suitable for broad readership of the EMBO Journal and support its publication. Below are a few comments that the authors may consider implementing.

We thank the reviewer for a positive evaluation of our manuscript.

Major points:

1) It was shown before that SKA3 phosphorylation at T358 and T360 is required for Ndc80:Ska complex formation. The authors used both untreated Ska FL and in-vitro phosphorylated Ska FL to show that these Ska molecules cooperate with Ndc80 to stabilize microtubule plus-ends by facilitating lateral interactions between microtubule protofilaments. An additional control that the authors could use to strengthen their claims that Ndc80:Ska oligomerize to facilitate microtubule stabilization would be to replace SKA3 WT with SKA3 T358A and T360A mutants within the Ska complex and then assess its ability to cooperate with non-stabilizing concentrations of Ndc80 to stabilize the microtubule plus-ends.

We thank the reviewer for suggesting this interesting experiment. We will generate Ska^{SKA3 T358A/T360A}, a mutant that was previously characterised in vitro (Huis in 't Veld et al., 2019), and perform microtubule end-stabilisation assays with this mutant and Ndc80 complex added to shortening microtubules.

2) Even though both GFP-SKA1 WT and GFP-SKA1 R236A correctly localized at kinetochores, the cold-treated GFP-SKA1 R236A-expressing cells displayed a strong reduction in kinetochore-microtubule attachments. Taking their results together, the authors conclude that this is due to the inability of SKA1 R236A to form oligomers. But is the SKA1 R236A-containing Ska complex functional? The authors could provide SKA3 immunofluorescence staining in GFP-SKA1 WT and GFP-SKA1 R236A cells to partially address this question.

We are grateful to the reviewer for raising this important point. First, we would like to refer to our current Supplementary Figure 4E that shows that Ska^{SKA1 R236A} forms a stable complex when expressed and purified *in vitro*. Additionally, we show in our current Supplementary Figure 6C that siRNA depletion of endogenous SKA1 results in reduced cellular levels of SKA3, which are restored to normal levels equally well by WT SKA1 and SKA1 R236A. These data in combination support our conclusion that SKA1 R236A forms a functional Ska123 complex. To further support this interpretation, we will perform immunofluorescence staining of HeLa cells for endogenous SKA3 in presence of transiently expressed GFP-SKA1 with or without the R236A mutation.

Minor points:

- 1) Line 167: it should read "we" instead of "were".
- 2) Line 188: remove "of" at the end.
- 3) Line 235: it should read "both" instead of "bot".
- 4) Line 336: it should read "depletion" instead of "depeletion".
- 5) Line 472: it should read "Cells" instead of "Cell".

We are grateful to the reviewer for noting typos and inconsistencies, which will all be corrected in the revised manuscript.

Referee #3:

Accurate segregation of aligned chromosomes during mitosis is essential for eukaryotic cells and ensured by the coordinated action of Ndc80 and Ska proteins. In this study, Radhakrishnan et al. investigated the function of the Ndc80-Ska complex using cryo-electron tomography (cryo-ET), total internal reflection fluorescence microscopy (TIRFM), and in vivo fluorescence imaging. Radhakrishnan et al. found that Ndc80-Ska complex stabilizes the microtubule plus ends by reinforcing the lateral interaction between protofilaments. Their analyses further revealed that the oligomerization of Ska molecules is mediated by the interaction of SKA1 microtubule binding domain (MTBD) and C-terminal tail and the Ska proteins form an envelope-like distribution on microtubules. The authors also identified a residue in SKA1 MTBD which is important for Ska oligomerization but dispensable for microtubule binding. Mutation of this residue in Ska hinders the stable kinetochore-microtubule attachment in cold conditions in vivo as well as microtubule end-tracking ability of Ska complex in vitro. Overall, this research provides insights into the microtubule stabilizing mechanism in mitosis with thorough analyses and proposes the possible role of Ska molecule in stabilizing microtubule ends. However, there are some fundamental issues in the current manuscript, and therefore it cannot be published in its current form. This reviewer thinks that the manuscript could be suitable for publication in The EMBO Journal, provided the authors appropriately address the points listed below.

We thank the reviewer for supporting the publication of a revised version of the manuscript.

Major comments:

In Supplementary Figure 1C and D, the authors claim that the hyperphosphorylation of Ska did not affect the fraction of seeds with extensions. However, the lengths of microtubules in the condition of untreated Ska appear shorter than the hyperphosphorylated condition in Supplementary Fig. 1D. Thus, even the fraction of extensions might be the same, hyperphosphorylation might have effects on the degree of extensions. Did the authors compare the lengths of extensions between these conditions?

We thank the reviewer for pointing out the differences in microtubule lengths in a Supplementary Figure. We will quantify the lengths of fluorescent extensions, and also repeat this experiment with Ska^{SKA3 T358A/T360A}, a mutant of FL SKA3 that does not interact with Ndc80 (related to point 1 raised by Reviewer 2, and point 6 raised by Reviewer 1).

In this work, the authors were not able to show the clear structure of Ska complex bound on the microtubule. There is, however, a previous report using cryo-ET to visualize each Ska molecule bound to the microtubule (Janczyk et al., 2017, "Mechanism of Ska Recruitment by Ndc80 Complexes to Kinetochores, Developmental Cell). This paper is not cited in this manuscript. The authors are

encouraged to discuss the reason why similar structures were not obtained in this study with citing the paper.

We are grateful to the reviewer for spotting our regrettable omission of a citation to this important work; this will be rectified in the revised manuscript. In the paper in question, Janczyk et al observed densities they interpreted as the Ska complex on the lattice of microtubule in individual micrographs of negatively stained samples. In individual denoised tomograms, we do observe plenty of microtubule-binding densities we interpret as Ska, as shown in the current manuscript in Figures 2C, 3B, 3D, Supplementary Figures 2BC and Supplementary Figure 3A. We did not however observe regular densities of Ska after subtomogram averaging, a method that Janczyk et al. did not employ. We therefore do not think there is any contradiction, and will address a potential confusion in the revised manuscript by citing Janczyk et al., 2017 and comparing our individual tomographic images to the observations in that paper.

In Fig. 2D, it appears that a protofilament sheet is covering a microtubule (MT1) and forms another microtubule (MT2), but this is not mentioned in the main text. Is this something often observed with Ndc80? What is the possible mechanism of the covering of the microtubule and branching? Please describe.

We thank the reviewer for pointing out this interesting observation. We do observe long protofilament clusters/sheets stabilised by Ndc80 oligomers at their plus ends more frequently than in Ndc80-free samples (see other examples in Supplementary Figure 2). However, a situation depicted in Figure 2D with a sheet covering another microtubule is unique. We will clarify the difference in the revised manuscript, and add a quantitative analysis of the plus-end taper length with and without Ndc80/Ska decoration.

In the graph in Fig. 4D, the authors compare the recovery of fluorescence of SkaFL and SkaSka3ΔC in the graph. However, only the kymograph of SkaFL is shown. For completeness and to allow readers to directly compare the two conditions, it would be helpful to also include a representative kymograph for SkaSka3ΔC.

We are grateful to the reviewer for pointing out the omission of a kymograph showing recovery of Ska^{SKA3ΔC} after photobleaching. This will be rectified in the revised manuscript.

In the graph in Fig. 4H, there are some constructs with data from a single experiment without repeated experiments. The authors are encouraged to check at least if these results are reproducible.

We thank the reviewer for being diligent regarding the reproducibility of the experimental data presented. We will include additional analyses of experimental repeats.

Regarding Fig. 5, there are several issues that need to be addressed. First, in Fig. 5E, several constructs only have data without repeated experiments and included in the statistical analysis. This case is more serious since the values of constructs like R245A and K223/226A are comparable to some data points of FL Ska constructs. The authors need to perform three independent replicates since this directly affects the conclusion that R236A is the only point mutation that did not affect the long binding events.

We thank the reviewer for their valuable suggestions. In the revised manuscript, we will include additional analysis of all experimental replicates performed. Individual datapoints, in their hundreds, are already plotted in Figure 5 and Supplementary Figure 5 for the reader to make their own judgement.

Furthermore, the binding events up to 1.0 s are plotted in the graphs in Fig. 5C and D. In the kymographs in Fig. 5B, however, there are some binding events clearly longer than 1 second. The details of the analysis are not described, either. Please describe how binding events exceeding 1 second were treated.

Rare events exceeding 1 s were accounted fully in the analysis, but were excluded from residence time plots to draw attention to where majority of the effects were observed, below 1 s. Description of this analysis will be expanded in the revised manuscript.

Another problem with Fig. 5C and D is that, although the short events are indicated as <0.3 s and long events as >0.3 s, the boundaries of shaded areas in the graphs are set around 0.38 s on the X-axis. This must be fixed. Furthermore, in the main text, short events are defined as {less than or equal to}0.3s (Line 314). Please clarify whether the definition is <0.3 s or {less than or equal to}0.3 s.

To address this important point raised by the reviewer, we will compare results obtained with a range of thresholds between "long" and "short" events and report the results of this analysis in the revised manuscript.

The authors are also saying in the main text that "the residence time of SKA1 MTBD never exceeded 0.4 s (Figure 5B, C)" (Line 307-308) and "the fraction of residence times that were never or very rarely observed for MTBD/LMTBD (long events, >0.3 s, Figure 5C)". However, there is clearly one MTBD data point that exceeds 0.4 s in Fig. 5C. In fact, the data presented in Fig. 5C labeled as MTBD (n = 404) looks a lot like LMTBD (n=243) data in Fig. S5B rather than MTBD data (n = 404) in Fig. S5B. Please check if the authors have used the correct dataset when preparing Fig. 5C.

We thank the reviewer for spotting this mistake. Indeed, LMTBD was plotted in Figure 5C instead of MTBD. The manuscript text stands correct. We will correct the graph in Figure 5C in the revised manuscript.

Minor comments:

In contrast to the figures where superscripts are used for the construct names, such as SkaSKA3 Δ C and FL Ska SKA1 R236A, they are not shown as superscripts in the main text as well as in the figure legends. Furthermore, in all cases of MgCl₂, the "2" is not in subscript. It seems that this happened when the authors were preparing the manuscript file for the submission. Please check if these are as the authors intended.

For the readers who are not familiar with Ska-Ndc80, the models in Fig. 1B can be improved. First, RWD domain should be indicated in the Ndc80 complex. Second, the schematic of Ska complex is too simplified, and it's hard to understand which part corresponds to SKA1, 2 and 3, respectively. Please indicate which part correspond to which region as well as phosphorylation sites.

The authors used many different truncated constructs as well as point mutations to reveal functions of Ska complex. It's easy to lose track which region and/or residues are important for which functions. Nevertheless, there is only a summary figure of whole Ska complex and Ndc80 in Fig. 7D. Adding a summary figure focusing on Ska complex's domains and residues would be helpful for the readers.

Fig. 1E legend, "1nM Ska" and "10nM Ska": there should be spaces between numbers and units.

Fig. 3E is not cited in the main text. It should be mentioned.

Fig. 7B legend, "1 nN Ska": this should be 1 nM Ska.

Line 20: "cryoET" should be defined as an abbreviation when first mentioned.

Line 52 "Yatskevich et al., 2023": did the authors mean "Yatskevich et al., 2024" as in the references?

Line 147 " $p = 0.0006$ for the effect of Ska concentration": this seems to contradict the value in the legend of Fig. 1E ($p = 0.0012$).

Line 182: "TIRF" should be defined as an abbreviation when first mentioned.

Line 188 "and found a similar compacted spacing of (Kalutskii et al., 2025) (Supplementary Figure 2F)": the word "of" looks unnecessary here.

Line 235 "bot": this should be "both".

Line 235 "in which Ndc80 (1nM)": it lacks space between 1 and nM.

Line 244 "'envelopes' (Siahaan et al., 2019).": the term "envelope" is not used in Siahaan et al., (2019). Did the author intend to cite Siahaan et al., 2022, "Microtubule lattice spacing governs cohesive envelope formation of tau family proteins", Nat. Chem. Biol.?

Line 356 "and found that 1 nM FL SkaSKA1 R236, in presence of 1 nM Ndc80, failed to stabilise microtubules against disassembly": "SkaSKA1 R236" should be "SkaSKA1 R236A".

Line 349: ACA should be defined as an abbreviation when first mentioned.

Line 350: Figure 6D should be Figure 6C.

Line 369: the term "tremendously" is highly subjective and should be avoided.

Line 518 "in an approximate ration of": should be ratio instead of ration.

Line 528 "e. coli BL21(DE3) Rosetta cells": it should be fixed to E. coli with capital E.

Line 651 "4.5g/L Glucose, 2mM L-Glutamine": spaces are necessary between numbers and units.

Line 653: it seems that CO₂'s O is written as 0 here. Furthermore, the "2" should be subscript.

In the "Cloning, expression, and purification of SKA1 fragments." section of Materials and Methods, all the temperature conditions are written as "C" instead of " $^{\circ}$ C". All of them should be fixed.

From line 557-574, the temperature conditions are written as "C" instead of " $^{\circ}$ C" again.

We are grateful to the reviewer for noting typos and inconsistencies, which will all be corrected in the revised manuscript.

Dr. Vladimir A. Volkov
Queen Mary University of London
Centre for Molecular Cell Biology, School of Biological and Behavioural Sciences
Mile End Rd
E1 4NS
United Kingdom

2nd Oct 2025

Re: EMBOJ-2025-121896R-Q
Microtubule end stabilisation by cooperative oligomers of Ska and Ndc80 complexes

Dear Vladimir,

Thank you for your message in response to my post-review decision on your recent EMBO Journal submission. I have now finally had a chance to go through your detailed answers to the various comments of the referees, and I appreciate that they are overall well-taken and could potentially address the overriding concerns that had been raised in particular by referee 1. In this light, I would be happy to give you an opportunity to resubmit a new version of this study, revised as proposed in your letter, using the link below (and also incorporating the revision instructions below and in our author guidelines). I hope you understand that I am not in a position to predict the end result of the re-evaluation, which will naturally depend both on the outcome of the planned revision experiments and on convincing the initially critical referees 1 and 3.

Please do not hesitate to contact me with any follow-up questions or major updates, or in case the revision experiments should require more time than the default 3-month period. I look forward to reading your resubmission.

With kind regards,

Hartmut

9) To facilitate reproducibility and cross-laboratory adoption of methodologies, please structure the Materials & Methods section as outlined in our guide to authors, including a completed Reagents and Tools Table that can be downloaded from our author guidelines as well (<https://www.embopress.org/page/journal/14602075/authorguide#structuredmethods>).

10) Digital image enhancement is acceptable practice, as long as it accurately represents the original data and conforms to community standards. If a figure has been subjected to significant electronic manipulation, this must be clearly noted in the figure legend and/or the 'Materials and Methods' section. The editors reserve the right to request original versions of figures and the original images that were used to assemble the figure. Finally, we generally encourage uploading of numerical as well as gel/blot image source data; for details see: embopress.org/page/journal/14602075/authorguide#sourcedata

In the interest of ensuring the conceptual advance provided by the work, we recommend submitting a revision within 3 months (31st Dec 2025). Please discuss the revision progress ahead of this time with the editor if you require more time to complete the revisions. Use the link below to submit your revision:

Link Not Available

Dr Vladimir Volkov
Reader in Biochemistry
Queen Mary University of London
School of Biological and Behavioural Science
Centre for Molecular Cell Biology
London E1 4NS
Email: v.volkov@qmul.ac.uk

Revision EMBOJ-2025-121896R1

Microtubule end stabilisation by cooperative oligomers of Ska and Ndc80 complexes

Dear Hartmut,

many thanks to you and the reviewers for critical evaluation of our original submission. We hereby submit a revised version of the manuscript that fully addresses all concerns raised by all three referees with additional experiments, data analysis, and revisions of manuscript text and figures. Incorporating the reviewers' suggestions has significantly improved the strength of our arguments, and we sincerely hope that the revision will satisfy the reviewers sufficiently to recommend the paper for publication.

Two most important changes include: additional experiments and data analysis to strengthen the causal link between Ndc80 oligomers binding to microtubule ends and protofilament clustering, and additional experiments to strengthen the causal link between non-uniform microtubule coating by Ska and its oligomerisation, both in response to issues raised by reviewer 1.

We have also added new quantifications of our *in vivo* observations of phenotypes caused by the SKA1 R236A mutant, and additional analyses of our *in vitro* single-molecule microscopy data.

Please find detailed point-by-point responses to all reviewers' comments below (added in blue font).

With best wishes,

Vladimir.

Point-by-point response to reviewers' comments: our responses added in blue font.

Referee #1:

In the manuscript "Microtubule end stabilisation by cooperative oligomers of Ska and Ndc80 complexes," Radhakrishnan and colleagues characterize the oligomerization behavior of reconstituted Ska and Ndc80 complexes and examine the impact of their oligomerization on microtubule stability. The authors initially demonstrate that addition of purified Ska complexes to low concentrations of Ndc80 complexes (which alone poorly stabilize dynamic microtubules) leads to longer microtubule growth extensions off stable seeds *in vitro*. This confirms previous studies showing synergistic behavior of these two complexes in promoting microtubule (MT) stability. The authors then visualize Ska and Ndc80 complexes on dynamic MTs using cryo-electron tomography, observing ordered oligomers of Ndc80 complexes in the presence or absence of Ska complexes, as well as disordered clumps of Ska complexes associated with MTs under both conditions. They measure the number of "non-bent" protofilaments per MT plus end under various conditions and report that MTs incubated with either stabilizing concentrations of Ndc80C or sub-stabilizing concentrations of Ndc80C plus Ska complex contain higher numbers of non-bent MT protofilaments at plus ends compared to previously published values for plus-ends of free MTs. These experiments further confirm that Ndc80 and Ska complexes promote MT plus end stability.

In subsequent experiments, the authors investigate Ska complex oligomerization in the absence of Ndc80 complexes. On MTs that are determined to be depolymerizing, Ska densities appear to associate primarily with the plus ends and manifest in cryo-ET images as large masses whose thickness around the MT surface exceeds the dimensions of a single Ska dimer. The conclusion is that these observed densities represent oligomeric complexes. Using TIRF-M, the authors then employ an "envelope" assay, in which dynamic MTs are incubated with fluorescent Ska complexes *in vitro* and Ska density along the MT is analyzed. Using full-length, wild-type Ska complexes, higher densities of Ska are observed near the MT growing ends, resulting in non-uniform density along the lattice. The authors hypothesize that this non-uniformity is specifically a property of oligomerized Ska complexes on MTs. Using this as their metric for Ska oligomerization, they describe domains and amino acids required for formation of the "envelope," or the bright region of Ska near the MT plus end. They show that a forced dimer of Ska1 MTBD is sufficient for envelope formation and that mutation of R236 in this domain perturbs its formation. Additionally, using FRAP, the authors report that the Ska3 C-terminus, which is responsible for Ndc80 complex binding, contributes to stabilizing MT-bound Ska oligomers. The authors report that Ska complexes containing the Ska1-R236A mutant retain intact MT binding capacity by analyzing the residence times of the complexes on MTs. Finally, the authors express the Ska1-R236A mutant in human tissue culture cells and, using a cold-stable MT assay, find that kinetochore-MTs are reduced in these cells compared to those expressing WT Ska1.

Overall, this paper presents interesting experiments that support the idea that oligomeric Ska and Ndc80 complexes form on MTs and play a role in stabilizing MT plus ends against depolymerization. However, the major conclusions are not strongly supported by the data. Specifically, the conclusions regarding Ska oligomerization are based on the "envelope" assay, which the authors use as a metric for oligomerization; however, there are no data supporting the premise that changes in Ska density along the lattice are indeed indicators of oligomerization. Additionally, the conclusion that Ska and Ndc80 oligomers promote increased lateral binding between plus-end protofilaments as the mechanism for MT stabilization is not supported by the data presented. These and other points are expanded upon below. In light of these issues, I do not recommend publication in The EMBO Journal.

We thank the reviewer for their critical assessment of our work – this helped us to improve our manuscript. We believe that by addressing the raised major points, we do now present our findings in a clearer and better controlled manner. We hope that the reviewer agrees and are looking forward to their feedback.

Major points:

(1) Using TIRF microscopy to analyze Ska complexes on dynamic microtubules, the authors observe that wild-type, full-length Ska complexes decorate MTs in a non-uniform manner, with brighter regions found near the plus ends ("envelopes"). They hypothesize that this non-uniform decoration results from oligomeric Ska complexes on the MTs. While this is a reasonable hypothesis, it is not experimentally tested in the study. Even so, the authors go on to use the "envelope assay" as their metric for the presence or absence of Ska oligomerization. As such, they conclude that certain mutants prevent formation of Ska oligomers or oligomer stabilization and then use these data to define Ska oligomerization domains, which represents a major finding of the paper. However, without direct evidence that envelope formation correlates with oligomerization, these conclusions are not adequately supported by the data.

We thank the reviewer for raising this important point. First, we do now more explicitly state that we interpret the non-uniform distribution and envelopes near the end (lines 283-289 on page 7). We support this interpretation by referring to studies of microtubule-binders with a non-uniform microtubule coating that self-interact on microtubules (e.g. *Ndc80* – Alushin et al., Nature 2010 & Polley et al., EMBO J 2023, *tau* – Tan et al., Nat Cell Biol 2019; *CENP-OPQR* – Pesenti et al., Mol Cell 2018; *Mal3/Tip1/Tea2* – Maan et al., Nat Cell Biol 2023).

Second, to directly address the reviewer's concern, we analysed wild-type (wt) Ska and Ska^{SKA1 R236A}, a mutant with disrupted ability to produce non-uniform MT decoration, side-by-side using the envelope assay based on TIRF microscopy and using cryoET of Ska variants on microtubules (Figure 5). Both methods showed smaller oligomers of Ska^{SKA1 R236A} compared to wt Ska. This supports our interpretation that formation of non-uniform microtubule decoration by Ska correlates with Ska oligomerisation.

(2) The authors conclude that Ndc80 and Ska complex oligomerization stabilizes MT plus ends by promoting lateral association of MT protofilaments. This conclusion is drawn, in part, on a previous study by the authors demonstrating that MT plus ends with GTP-bound tubulin exhibit pronounced protofilament clustering, while this clustering is reduced at MT plus ends with GDP-bound tubulin. Since Ska and Ndc80 complexes stabilize MT plus ends, it is expected that these plus ends should exhibit high protofilament clustering, which is observed on stable, growing MTs (Kalutskii et al., 2025). However, the authors do not establish causality: there is no evidence that Ndc80 and Ska complexes actively drive protofilament clustering to promote stability, as opposed to a scenario in which these complexes promote stability through other mechanisms, with clustered protofilaments observed as a downstream consequence. Given that MTs with GTP-bound tubulin naturally exhibit clustering in the absence of external factors, the latter interpretation appears more likely.

We agree with the reviewer that it is important to distinguish cause and consequence when studying the impact of Ndc80 and Ska (and MT-binders in general) on the dynamic state of a microtubule end and the lateral clustering of protofilaments. To test if a) Ndc80 and Ska stabilise microtubule ends via

protofilament clustering, or b) protofilament clustering is a consequence of a different stabilisation mechanism, we set out to establish conditions with reduced lateral protofilaments contacts and probed whether Ndc80/Ska decoration restores lateral interactions. This is described in the current Figure 3 and on page 6, lines 217-223: microtubules are forced to depolymerise by dilution of tubulin, a treatment we previously established to produce protofilament singles and doubles (Kalutskii et al., 2025). Under these conditions, the presence of Ndc80 or Ndc80/Ska oligomers increased the fraction of protofilament clusters with a size ≥ 2 and reduced the fraction of single protofilaments.

To directly address the reviewer's concern regarding the causality, we have also performed and analysed additional experiments. First, we compared the shapes of microtubule plus-ends with or without an Ndc80 oligomer bound within 50 nm of the protofilament flare (both conditions found in the same dataset with 3-10 nM Ndc80 stabilising microtubules against shortening). Beyond a direct visualization of Ndc80 on dynamic microtubules, we find that the presence of an Ndc80 oligomer at the plus-end correlates with increased protofilament clustering (Fig 3EF). Second, we performed cryoET of shortening microtubules with and without a low, non-stabilising concentration of Ndc80 (1 nM) and found that protofilaments were more clustered in the presence of Ndc80 but in absence of microtubule end stabilisation (Fig 3E). We use these data to postulate that protofilament clustering starts increasing before microtubules are stabilised. Both observations reinforce our conclusion that Ndc80 oligomers cause protofilament clustering, which in turn causes the microtubules to get stabilised against shortening (pages 6-7, lines 250-270).

(3) Related to the above point: For the protofilament clustering experiments, the authors rely on control data and measurements from their previously published work rather than generating independent controls for this study. Given that protofilament clustering is a newly documented phenomenon, it seems important to carry out independent controls in this new study.

We have now performed and analysed control experiments with microtubule shortening in absence of any binding proteins – these new data are now used in the manuscript as a control. The new data are comparable to the data previously reported in Kalutskii et al., 2025. The comparison is added below for convenience.

Furthermore, the authors do not specify the distance threshold used to define protofilaments as "clustered," or how this was determined. The manuscript would benefit from additional representative images comparing protofilaments that meet their clustering criteria versus those that are adjacent but do not qualify as clustered, as this would help readers evaluate the data.

We agree with the reviewer that this is an important point and have specified and clarified how we determined protofilament clustering in the text (page 6, lines 257-262) and in Figure 3 and Expanded View Figure EV3. In brief, we used an overlap threshold of 0.1 as in previous work; this threshold indicates that at least 50% of the weighted linear distances between protofilaments deviate by less than 20%. We do now also include a detailed analysis of differences in cluster distributions for a range of thresholds between 0.03 and 0.5 (Expanded View Figure EV3G). The results of these analyses were insensitive to the value of the threshold, as reported previously in Kalutskii et al, 2025. In addition, we provided detailed illustrations to show protofilaments considered clustered and not

clustered, using the threshold of 0.1 which we use throughout the study (Figure 3CD).

(4) The EM images are striking; however, they do not necessarily further our understanding of how these complexes functionally impact MT biology or assemble on MTs. A revealing functional experiment would be to assess the formation of Ska oligomers/aggregates in the presence of the anti-oligomerization mutants which would potentially support their findings (see point #1).

To test this, we generated wt and mutant Ska labelled with AzDye488 (wt) or TMR (oligomerisation-deficient Ska^{SKA1 R236A}) and compared these side-by-side on microtubules using TIRF microscopy. These experiments demonstrate that Ska^{SKA1 R236A} fails to follow the non-uniform distribution of wt Ska on microtubules (Fig. 5D) and prevents envelope formation of wt Ska when present in excess (Fig. 5E). We thank the reviewer for suggesting this elegant experiment.

(5) In Figure 3, the authors conclude from the EM images that NDC80 CH domains occupy two neighboring protofilaments; however, the resolution of the presented images appears insufficient to support this point.

We appreciate the concern about the resolution in our EM images. Our 3D class averages have a resolution of ca. 23Å, judged by the gold-standard 0.5 FSC criterion applied to post-processed half-maps. This resolution should suffice to distinguish individual Ndc80 molecules occupying neighbouring tubulin monomers in the microtubule lattice that are 41 - 43Å apart within the same protofilament, and ca. 60Å apart between neighbouring protofilaments. To further support our conclusion that Ndc80 occupies adjacent protofilaments, we have now included 4-nm thick summed cross-sections through four individual protofilaments in our 3D class average of a microtubule decorated with 10 nM Ndc80 (Figure 2E), showing Ndc80 CH-domain density on protofilaments 1 and 2, but not 3. Protofilament 0 shows a weak CH-domain density arising from a subset of double-protofilament Ndc80 trains localising to the opposite side of the primary train used for alignment.

Notably, individual adjacent protofilaments in a microtubule and linear arrays of Ndc80 CH-domains can also be distinguished without averaging. We now provide videos 1-3 showing non-averaged 0.8-nm slices from three denoised tomograms with Ndc80 oligomer densities on adjacent protofilaments. We have also uploaded 3D class averages we obtained to EMDB: EMD-56085 (10 nM Ndc80), EMD-56086 (3 nM Ndc80), EMD-56087 (1 nM Ndc80 and 10 nM Ska, non-Ndc80 microtubule-bound densities), and EMD-56088 (1 nM Ndc80 and 10 nM Ska, Ndc80 oligomers).

(6) The FRAP data nicely show that the Ska3 C-term plays a role in stabilizing Ska on MTs (independently of Ndc80). Does the stabilizing domain overlap with the NDC80 binding domain?

To address this point directly, we have generated Ska^{SKA3Δ351-377}, a mutant lacking the region predicted to bind Ndc80 by AlphaFold3. We repeated the FRAP assay using this mutant and found that its recovery was similar to full-length Ska (Figure 4D).

(7) In Supp Figure 5A, it is interesting that the dimer population is increased in Ska3 C-term mutant condition compared to WT. Since the C-term was defined in this paper as a Ska-Ska oligomerization stabilizing region, do the authors have an explanation for the puzzling data in the figure?

The percentage of dimers did indeed slightly decrease in the case of Ska^{SKA3ΔC}. Since this difference did not appear statistically significant, we did not speculate about a possible explanation. Please also see our response to the next point raised.

(8) Figure 5: The authors are using residence time as correlative indicator of "direct MT binding" but oligomerization can also affect this parameter (e.g. artificially dimerizing NDC80 complexes with an antibody significantly increasing residence time without altering the binding domains). They state that they are using single molecule conditions (a scenario where considering oligomers would not be necessary), but Supp Figure 5A shows that there are both dimers and monomers present under the experimental conditions.

We thank the reviewer for pointing out the existence of both monomers and dimers of Ska in our experimental conditions. We agree that oligomerization can increase residence time without affecting microtubule-binding domains. For Ska FL, we did not treat monomers and dimers separately since dimers of Ska did not have a longer residence time than monomers ($p = 0.0988$). Using Ska^{SKA3ΔC}, we observed a surprising reverse effect: slightly shorter residence times for dimers. These data are included in the new Expanded View Figure EV6A.

(9) For the cell work in Figure 6, it would be helpful to include some analysis for non-cold treated cells to have some indication of the mitotic phenotype for cells expressing WT Ska1 vs. Ska1 R236A with no perturbation (possibly chromosome alignment phenotype, timing through mitosis, or chromosome segregation errors).

Related to this, the supplemental data for this figure are somewhat confusing. The figure shows two very different phenotypic outcomes in cells presumably depleted of Ska1 and not rescued. Which is more frequently observed? Does one have stable attachments and the other doesn't? Do either of these phenocopy the Ska1 R236A mutant? How, overall, do the phenotypes of Ska1 depletion compare to expression of the mutant?

We thank the reviewer for raising this important point. The SKA1 R236A mutant has been extensively characterised in two independent studies from the Jeyaprakash and the Cheeseman/Grishchuk labs which we discuss in the manuscript (Abad et al., 2014 and Monda et al., 2017). These studies report increased mitotic timing for R236A and R236/245A. In addition, Monda et al. report a slight increase in major chromosome alignment defects for the R236A mutation. We do now more clearly refer the reader to these studies (line 414, page 10). We have also performed additional validation experiments to determine if the expression of GFP-SKA1 R236A (siRNA resistant) in cells treated with siRNA against endogenous SKA1 causes chromosome alignment phenotypes. These data are now included in the new Figure 7D and Expanded View Figure EV7E.

(10) In Figure 7, the authors state that for the 1 nM and 10 nM Ska conditions (with either 0, 0.3, or 1 nM Ndc80 added), the fractions of seeds with extensions are not significantly different from each other, but both are significantly different from the "no Ska added" condition. It is a little hard to see how the 0 vs. 10 nM Ska conditions are different. It would be helpful to add the calculated significance scores to these.

We have changed the colours to provide better contrast. Significance values are provided for a 2-way ANOVA testing the significance of changes in Ndc80 or Ska concentrations: row factor (Ska

concentration) $p = 0.0084$; column factor (Ndc80 concentration) $p = 0.11$.

Minor points:

(1) The rationale for the experiments is not altogether clear. In the introduction, the authors state that "...it is currently unclear whether full-length Ska and Ndc80 can spontaneously oligomerize on microtubules, or they require externally imposed oligomerisation." Is the question if the complexes can oligomerize together in the absence of the NDC80 kinetochore recruitment factors (e.g. is this the "externally imposed oligomerization")? Or, if Ska alone can oligomerize independently of NDC80? It would be helpful if the issue being addressed in the paper was a bit clearer for the reader.

We clarified this sentence to read: "Thus it is currently unclear whether full-length Ska and Ndc80 can spontaneously oligomerise on microtubules, or they require oligomerisation imposed externally by microtubule lattices and inner kinetochore proteins." (lines 82-84). We also re-wrote the concluding paragraph of the introduction (lines 120-130) to clarify the rationale for the experiments conducted in this study.

(2) Figure 2G: I think the significance values are switched (or, alternatively, the description of results doesn't match the figure).

The figure has been updated with new data addressing this reviewer's points above.

(3) In Figure 1E, why does 1 nM Ska enhance MT stabilization with 0.3 nM NDC80C better than 10 nM?

The difference between 1 nM vs 10 nM Ska, both at 0.3 nM Ndc80, is not statistically significant ($p = 0.1065$).

(4) Line 235: bot should be "both"

Corrected

(5) Figure 5: There is no "E" label on the figure

Corrected

(6) Line 298 text: I think this should be the subheading for this section. (Oligomerisation-deficient Ska has intact microtubule binding in single-molecule conditions).

Corrected

(7) In Figure 5, it would be helpful to compare the data for WT, FL Ska and the FL Ska R236 mutant, with the data side-by-side. The WT FL data are included in Panel C, but is tough to see what is going on with the overlaying plots and data points. Also, in Supplement for Figure 5, the replicates for FL Ska are displayed with a single curve fit while the mutant FL R236A is shown with fits for both the long and short. Was this intentional?

Corrected

(8) In Figure 7C, the blue vs. green dots are difficult to distinguish on the graphs.

Corrected

Referee #2:

In this manuscript, Radhakrishnan and colleagues combined in-vitro reconstitution assays with cryoET and TIRF microscopy to address the individual and synergistic effects of Ndc80 and Ska complexes in the stabilization of microtubule plus-ends. The authors showed that neither Ska alone nor non-stabilizing concentration of Ndc80 (<1nM) can prevent the microtubule plus-end shortening, but that combining Ska with non-stabilizing concentration of Ndc80 successfully stabilized microtubule plus-ends. Based on these results and the visualization of Ndc80 and Ska molecules along the neighboring microtubule protofilaments, they propose that the Ndc80-Ska oligomers facilitate lateral contacts between microtubule protofilaments, thereby protecting the microtubule plus-ends from depolymerization. Using site-directed mutagenesis, they also identified a residue of SKA1 (R236) required for Ska oligomerization and showed both in-cellulo and in-vitro that the oligomerization of Ska cooperates with Ndc80 to maintain stable kinetochore-microtubule attachments. In addition to the contribution of the Ndc80:Ndc80 interactions in the stabilization of the kinetochore-microtubule attachments, this study highlights the importance of Ndc80:Ska and Ska:Ska interactions in this important process and provides molecular insight into these interactions. By elucidating the role of these interactions in facilitating lateral contacts between microtubule protofilaments, the study provides a novel insight into how microtubule plus-ends are stabilized at kinetochores. The authors have clearly demonstrated their expertise in the in-vitro reconstitution of kinetochore-microtubule interactions. The manuscript is well-written, and the study is overall well-designed, convincingly executed and properly controlled. I find it suitable for broad readership of the EMBO Journal and support its publication. Below are a few comments that the authors may consider implementing. We thank the reviewer for a positive evaluation of our manuscript.

Major points:

1) It was shown before that SKA3 phosphorylation at T358 and T360 is required for Ndc80:Ska complex formation. The authors used both untreated Ska FL and in-vitro phosphorylated Ska FL to show that these Ska molecules cooperate with Ndc80 to stabilize microtubule plus-ends by facilitating lateral interactions between microtubule protofilaments. An additional control that the authors could use to strengthen their claims that Ndc80:Ska oligomerize to facilitate microtubule stabilization would be to replace SKA3 WT with SKA3 T358A and T360A mutants within the Ska complex and then assess its ability to cooperate with non-stabilizing concentrations of Ndc80 to stabilize the microtubule plus-ends.

We thank the reviewer for suggesting this interesting experiment. We generated Ska^{SKA3 T358A/T360A}, a mutant that was previously characterised in vitro (Huis in 't Veld et al., 2019), and performed microtubule end-stabilisation assays with this mutant and Ndc80 complex added to shortening microtubules. The new data are reported in the text (lines 169-172, 05) in Expanded View Figure EV1CD.

2) Even though both GFP-SKA1 WT and GFP-SKA1 R236A correctly localized at kinetochores, the cold-treated GFP-SKA1 R236A-expressing cells displayed a strong reduction in kinetochore-microtubule attachments. Taking their results together, the authors conclude that this is due to the inability of SKA1 R236A to form oligomers. But is the SKA1 R236A-containing Ska complex functional? The authors could provide SKA3 immunofluorescence staining in GFP-SKA1 WT and GFP-SKA1 R236A cells to partially address this question.

We are grateful to the reviewer for raising this important point. First, we would like to refer to our current Expanded View Figure EV5E that shows that Ska^{SKA1 R236A} forms a stable complex when expressed and purified *in vitro*. Additionally, we show in our current Expanded View Figure EV7C that

siRNA depletion of endogenous SKA1 results in reduced cellular levels of SKA3, which are restored to normal levels equally well by WT SKA1 and SKA1 R236A. These data in combination support our conclusion that SKA1 R236A forms a functional Ska123 complex. To further support this interpretation, we performed immunofluorescence staining of HeLa cells for endogenous SKA3 in presence of transiently expressed GFP-SKA1 with or without the R236A mutation. The new data are reported in the text (lines 404-408, p 10), and in the Expanded View Figure EV7D.

Minor points:

1) Line 167: it should read "we" instead of "were".

Corrected

2) Line 188: remove "of" at the end.

Corrected

3) Line 235: it should read "both" instead of "bot".

Corrected

4) Line 336: it should read "depletion" instead of "depeletion".

Corrected

5) Line 472: it should read "Cells" instead of "Cell".

Corrected

Referee #3:

Accurate segregation of aligned chromosomes during mitosis is essential for eukaryotic cells and ensured by the coordinated action of Ndc80 and Ska proteins. In this study, Radhakrishnan et al. investigated the function of the Ndc80-Ska complex using cryo-electron tomography (cryo-ET), total internal reflection fluorescence microscopy (TIRFM), and in vivo fluorescence imaging. Radhakrishnan et al. found that Ndc80-Ska complex stabilizes the microtubule plus ends by reinforcing the lateral interaction between protofilaments. Their analyses further revealed that the oligomerization of Ska molecules is mediated by the interaction of SKA1 microtubule binding domain (MTBD) and C-terminal tail and the Ska proteins form an envelope-like distribution on microtubules. The authors also identified a residue in SKA1 MTBD which is important for Ska oligomerization but dispensable for microtubule binding. Mutation of this residue in Ska hinders the stable kinetochore-microtubule attachment in cold conditions in vivo as well as microtubule end-tracking ability of Ska complex in vitro. Overall, this research provides insights into the microtubule stabilizing mechanism in mitosis with thorough analyses and proposes the possible role of Ska molecule in stabilizing microtubule ends. However, there are some fundamental issues in the current manuscript, and therefore it cannot be published in its current form. This reviewer thinks that the manuscript could be suitable for publication in The EMBO Journal, provided the authors appropriately address the points listed below.

We thank the reviewer for supporting the publication of a revised version of the manuscript.

Major comments:

In Supplementary Figure 1C and D, the authors claim that the hyperphosphorylation of Ska did not affect the fraction of seeds with extensions. However, the lengths of microtubules in the condition of untreated Ska appear shorter than the hyperphosphorylated condition in Supplementary Fig. 1D. Thus, even the fraction of extensions might be the same, hyperphosphorylation might have effects on the degree of extensions. Did the authors compare the lengths of extensions between these conditions?

We thank the reviewer for pointing out the differences in microtubule lengths in a Supplementary Figure. We quantified the lengths of fluorescent extensions, and also repeated this experiment with Ska^{SKA3 T358A/T360A}, a mutant of FL SKA3 that does not interact with Ndc80 (related to point 1 raised by Reviewer 2, and point 6 raised by Reviewer 1). The new data are reported in Expanded View Figure 1D.

In this work, the authors were not able to show the clear structure of Ska complex bound on the microtubule. There is, however, a previous report using cryo-ET to visualize each Ska molecule bound to the microtubule (Janczyk et al., 2017, "Mechanism of Ska Recruitment by Ndc80 Complexes to Kinetochores, Developmental Cell). This paper is not cited in this manuscript. The authors are encouraged to discuss the reason why similar structures were not obtained in this study with citing the paper.

We are grateful to the reviewer for spotting our regrettable omission of a citation to this important work; this has been rectified in the revised manuscript (line 343, p 8). In the paper in question, Janczyk et al observed densities they interpreted as the Ska complex on the lattice of microtubule in individual micrographs of negatively stained samples. In individual denoised tomograms, we do observe plenty of microtubule-binding densities we interpret as Ska, as shown in Expanded View Figure EV2B, Expanded View Figure EV4A, and Figure 5A. We did not however observe regular densities of Ska after subtomogram averaging, a method that Janczyk et al. did not employ. We therefore do not think there is any contradiction.

In Fig. 2D, it appears that a protofilament sheet is covering a microtubule (MT1) and forms another microtubule (MT2), but this is not mentioned in the main text. Is this something often observed with Ndc80? What is the possible mechanism of the covering of the microtubule and branching? Please describe.

We thank the reviewer for pointing out this interesting observation. We do observe long protofilament clusters/sheets stabilised by Ndc80 oligomers at their plus ends more frequently than in Ndc80-free samples (see examples in Expanded View Figure EV2AC, Figure 3A). However, a situation depicted in the previous Figure 2D with a sheet covering another microtubule is unique. We added a quantitative analysis of the plus-end taper length with and without Ndc80/Ska decoration (lines 244-246 on p 6, and Expanded View Figure EV3F) – we interpret results of this analysis to conclude that Ndc80 oligomers stabilise lateral interactions between protofilaments that would otherwise fall apart.

In the graph in Fig. 4D, the authors compare the recovery of fluorescence of SkaFL and SkaSka3ΔC in the graph. However, only the kymograph of SkaFL is shown. For completeness and to allow readers to directly compare the two conditions, it would be helpful to also include a representative kymograph for SkaSka3ΔC.

We are grateful to the reviewer for pointing out the omission of a kymograph showing recovery of Ska^{Ska3ΔC} after photobleaching. This has been rectified in the revised manuscript (Figure 4D).

In the graph in Fig. 4H, there are some constructs with data from a single experiment without repeated experiments. The authors are encouraged to check at least if these results are reproducible. We thank the reviewer for being diligent regarding the reproducibility of the experimental data presented. We included additional analyses of experimental repeats – now shown in the new Figure 4H.

Regarding Fig. 5, there are several issues that need to be addressed. First, in Fig. 5E, several constructs only have data without repeated experiments and included in the statistical analysis. This case is more serious since the values of constructs like R245A and K223/226A are comparable to some data points of FL Ska constructs. The authors need to perform three independent replicates since this directly affects the conclusion that R236A is the only point mutation that did not affect the long binding events.

We thank the reviewer for their valuable suggestions. In the revised manuscript, we included additional analysis of all experimental replicates performed (Figure 6D). Individual datapoints, in their hundreds, are plotted in Figure 6C and Expanded View Figure EV6B.

Furthermore, the binding events up to 1.0 s are plotted in the graphs in Fig. 5C and D. In the kymographs in Fig. 5B, however, there are some binding events clearly longer than 1 second. The details of the analysis are not described, either. Please describe how binding events exceeding 1 second were treated.

Rare events exceeding 1 s were accounted fully in the analysis, but were excluded from residence time plots to draw attention to where majority of the effects were observed, below 1 s. Description of this analysis has been expanded in the Methods section of the revised manuscript (lines 764-767, p 18).

Another problem with Fig. 5C and D is that, although the short events are indicated as <0.3 s and long events as >0.3 s, the boundaries of shaded areas in the graphs are set around 0.38 s on the X-axis. This must be fixed. Furthermore, in the main text, short events are defined as {less than or equal to}0.3s (Line 314). Please clarify whether the definition is <0.3 s or {less than or equal to}0.3 s.

To address this important point raised by the reviewer, we compared results obtained with a range of thresholds between “long” and “short” events and reported the results of this analysis in the new Expanded View Figure EV6C and in the text (lines 375-389, p 9).

The authors are also saying in the main text that "the residence time of SKA1 MTBD never exceeded 0.4 s (Figure 5B, C)" (Line 307-308) and "the fraction of residence times that were never or very rarely observed for MTBD/LMTBD (long events, >0.3s, Figure 5C)". However, there is clearly one MTBD data point that exceeds 0.4 s in Fig. 5C. In fact, the data presented in Fig. 5C labeled as MTBD (n = 404) looks a lot like LMTBD (n=243) data in Fig. S5B rather than MTBD data (n = 404) in Fig. S5B. Please check if the authors have used the correct dataset when preparing Fig. 5C.

We thank the reviewer for spotting this mistake. Indeed, LMTBD was plotted in Figure 5C instead of MTBD. The manuscript text stands correct. We modified the new Figure 6C and Expanded View Figure EV6B to include one construct per plot, and plotted all of them side by side.

Minor comments:

In contrast to the figures where superscripts are used for the construct names, such as SkaSKA3 Δ C and FL Ska SKA1 R236A, they are not shown as superscripts in the main text as well as in the figure legends. Furthermore, in all cases of MgCl₂, the "2" is not in subscript. It seems that this happened when the authors were preparing the manuscript file for the submission. Please check if these are as the authors intended.

Corrected

For the readers who are not familiar with Ska-Ndc80, the models in Fig. 1B can be improved. First, RWD domain should be indicated in the Ndc80 complex. Second, the schematic of Ska complex is too simplified, and it's hard to understand which part corresponds to SKA1, 2 and 3, respectively. Please indicate which part correspond to which region as well as phosphorylation sites.

Corrected

The authors used many different truncated constructs as well as point mutations to reveal functions of Ska complex. It's easy to lose track which region and/or residues are important for which functions. Nevertheless, there is only a summary figure of whole Ska complex and Ndc80 in Fig. 7D. Adding a summary figure focusing on Ska complex's domains and residues would be helpful for the readers. Since we could not determine the structure of the microtubule-bound Ska, we refrained from providing a schematic that would be more detailed than our data. We believe that the current summary figure (Figure 8D) faithfully captures the essence of our conclusions.

Fig. 1E legend, "1nM Ska" and "10nM Ska": there should be spaces between numbers and units.

Corrected

Fig. 3E is not cited in the main text. It should be mentioned.

Corrected

Fig. 7B legend, "1 nN Ska": this should be 1 nM Ska.

Corrected

Line 20: "cryoET" should be defined as an abbreviation when first mentioned.

Corrected

Line 52 "Yatskevich et al., 2023": did the authors mean "Yatskevich et al., 2024" as in the references?

Corrected

Line 147 "p = 0.0006 for the effect of Ska concentration": this seems to contradict the value in the legend of Fig. 1E (p = 0.0012).

Corrected

Line 182: "TIRF" should be defined as an abbreviation when first mentioned.

Corrected

Line 188 "and found a similar compacted spacing of (Kalutskii et al., 2025) (Supplementary Figure 2F)": the word "of" looks unnecessary here.

Corrected

Line 235 "bot": this should be "both".

Corrected

Line 235 "in which Ndc80 (1nM)": it lacks space between 1 and nM.

Corrected

Line 244 "'envelopes" (Siahaan et al., 2019).': the term "envelope" is not used in Siahaan et al., (2019). Did the author intend to cite Siahaan et al., 2022, "Microtubule lattice spacing governs cohesive envelope formation of tau family proteins", Nat. Chem. Biol.?

Corrected. Both papers are now cited on lines 284-285, p 7 to avoid confusion in terminology.

Line 356 "and found that 1 nM FL SkaSKA1 R236, in presence of 1 nM Ndc80, failed to stabilise microtubules against disassembly": "SkaSKA1 R236" should be "SkaSKA1 R236A".

Corrected.

Line 349: ACA should be defined as an abbreviation when first mentioned.

Corrected.

Line 350: Figure 6D should be Figure 6C.

Corrected.

Line 369: the term "tremendously" is highly subjective and should be avoided.

Corrected.

Line 518 "in an approximate ration of": should be ratio instead of ration.

Corrected.

Line 528 "e. coli BL21(DE3) Rosetta cells": it should be fixed to E. coli with capital E.

Corrected.

Line 651 "4.5g/L Glucose, 2mM L-Glutamine": spaces are necessary between numbers and units.

Corrected.

Line 653: it seems that CO₂'s O is written as 0 here. Furthermore, the "2" should be subscript.

Corrected.

In the "Cloning, expression, and purification of SKA1 fragments." section of Materials and Methods, all the temperature conditions are written as "C" instead of "{degree sign}C". All of them should be fixed. From line 557-574, the temperature conditions are written as "C" instead of "{degree sign}C" again.
Corrected.

Dr. Vladimir A. Volkov
Queen Mary University of London
Centre for Molecular Cell Biology, School of Biological and Behavioural Sciences
Mile End Rd
E1 4NS
United Kingdom

12th Feb 2026

Re: EMBOJ-2025-121896R1
Microtubule end stabilisation by cooperative oligomers of Ska and Ndc80 complexes

Dear Vladimir,

Thank you for submitting a new version of your manuscript on microtubule stabilization by kinetochore complexes to The EMBO Journal. I sent it back to all three original referees, who have now returned their reports (see below). In addition, I also asked an expert Editorial Board Member of our journal for arbitrating input, given the somewhat divided opinions of the original referees. After discussing the combined input within our team, we eventually decided to proceed with publication of this work in The EMBO Journal, following incorporation of the various minor points raised by referees 1 and 3, and additional modifications of certain assertions and conclusions in the spirit of the comments by referee 1 and by our arbitrating expert, also copied below.

Furthermore, there are also a number of editorial points that should be taken care of at this stage:

- Please make sure to re-upload our author checklist with information completed in all columns.
- Please adjust the order of the manuscript sections, and also make sure to use the correct section headers:
Title page with complete author information, Abstract, Introduction, Results, Discussion, Methods, Data Availability, Acknowledgements, Disclosure and Competing Interests Statement, References, Main Figure Legends, Tables, Expanded Figure Legends.
- Please double-check to make sure to all relevant funding information in the manuscript is congruent with the info entered into our submission system. Currently missing in the submission system is: BBSRC grant BB/W019698/1
- Please rename the Conflict of Interest section into "Disclosure and Competing Interests Statement", in accordance with our updated Guide to Authors (<https://www.embopress.org/competing-interests>).
- As we are switching from a free-text author contribution statement towards a more formal statement based on Contributor Role Taxonomy (CRediT) terms, please remove the present Author Contribution section and instead specify each author's contribution(s) directly in the Author Information page of our submission system during upload of the final manuscript. See <https://casrai.org/credit/> for more information.
- In the Data Availability section, please include a direct URL to the respective database (EMDB) in which data generated for this study have been deposited.
- Please carefully go through the reference list and make sure to stick to the journal format:
 - * For references with multiple authors, the first up to 10 authors should be listed, followed by 'et al.' after that
 - * DOIs should not be included, except for preprints, databases, or other items that do not have a formal citation (yet)
 - * some references are incomplete, missing e.g. citation year, volume, and/or page/locator numbers
 - * Please adjust the format for citation of real preprints: The citation in the text should be: "(preprint: NAME1 et al, YEAR)"; and in the reference list: "NAME1, NAME2, ... (YEAR) article title. bioRxiv doi: XXX [PREPRINT]"
- Please move the Statement on AI use into the Methods section, instead of listing it as a separate section.
- Please check the legend for Figure EV2, in which panel (E) seems to be missing (or is wrongly labelled as (B)?)
- Please rename all movies as Expanded View movies (in-text callouts "Movie EV1/2/..."). Their legends should be moved out of the text into individual text files, each of which should be combined with the respective movie file into a separate ZIP file and uploaded as such.
- Please provide suggestions for a short 'blurb' text prefacing and summing up the conceptual aspect of the study in two sentences (max. 250 characters), followed by 3-5 one-sentence 'bullet points' with brief factual statements of key results of the

paper; they will form the basis of an editor-written 'Synopsis' accompanying the online version of the article. Please also upload a synopsis image, which can be used as a "visual title" for the synopsis section of your paper. The image (maybe based on Figure 8D with altered aspect ratio?) should be in JPG format, and please make sure that it remains in the modest dimensions of (exactly) 550 pixels wide and 300-600 pixels high.

- Our routine image checks indicated that a tomography panel seems to appear twice (Fig 2E and Fig EV2D) - please clarify. In case of an intentional re-display, this needs to be explicitly indicated and rationalized in both respective figure legends.

- Finally, during editorial pre-acceptance checks, our data editors have raised the following queries regarding figures, data, and legends; I would appreciate if you briefly answered to them in the cover letter of your final submission, and made the requested text modifications with changes/additions highlighted via the "Track changes" option, to facilitate our final checking":

- 1- Please define the annotated p values ****/***/**/* as well as provide the exact p-values for the same in the legend of figure 2D, 4D, H; 5C, F; 6D, 7B, D; 8C, EV1 C, EV3 E, F; EV6 C, EV7 B as appropriate.
- 2- Please note that the exact p values are not provided in the legend of figure 1E,
- 3- Please indicate the statistical test used for data analysis in the legends of figures EV1 C, EV3 E, F; EV6 C
- 4- Please note that information related to n is missing in the legends of figures EV1 C, EV3 A, C, E, F; EV6 C
- 5- Please note that the error bars are not defined in the legends of figures 2D, 7B, EV1 C, D; EV3 C, EV6 C
- 6- Please note that the white arrows are not defined in the legend of figure 5A. This needs to be rectified.

I am returning the manuscript to you for a final round of minor revision, to allow you to make these remaining modifications and upload the revised files. Once we will have received them, we should be ready to swiftly proceed with formal acceptance and production of the manuscript.

With kind regards,

Hartmut

*** PLEASE NOTE: All revised manuscript are subject to initial checks for completeness and adherence to our formatting guidelines. Revisions may be returned to the authors and delayed in their editorial re-evaluation if they fail to comply to the following requirements. As a first step please read our guidelines for revised submissions:
<https://link.springer.com/journal/44318/submission-guidelines#cms-Revised-submissions>

1) Every manuscript requires a Data Availability section (even if only stating that no deposited datasets are included). Primary datasets or computer code produced in the current study have to be deposited in appropriate public repositories prior to resubmission, and reviewer access details provided in case that public access is not yet allowed.

4) Each main and each Expanded View (EV) figure should be uploaded as individual production-quality files (preferably in .eps, .tif, .jpg formats). For suggestions on figure preparation/layout, please refer to our Figure Preparation Guidelines:
<https://media.springernature.com/original/springer-cms/rest/v1/content/27825798/data/v1>

6) Please complete our Author Checklist, and make sure that information entered into the checklist is also reflected in the manuscript; the checklist will be available to readers as part of the Review Process File.

7) All authors listed as (co-)corresponding need to deposit, in their respective author profiles in our submission system, a unique

ORCID identifier linked to their name. Please see our Guide to Authors for detailed instructions.

8) Please note that supplementary information at EMBO Press has been superseded by the 'Expanded View' for inclusion of additional figures, tables, movies or datasets; with up to five EV Figures being typeset and directly accessible in the HTML version of the article.

9) To facilitate reproducibility and cross-laboratory adoption of methodologies, please structure the Materials & Methods section as outlined in our guide to authors, including a completed Reagents and Tools Table.

10) Digital image enhancement is acceptable practice, as long as it accurately represents the original data and conforms to community standards. If a figure has been subjected to significant electronic manipulation, this must be clearly noted in the figure legend and/or the 'Materials and Methods' section. The editors reserve the right to request original versions of figures and the original images that were used to assemble the figure. Finally, we generally encourage uploading of numerical as well as gel/blot image source data.

In the interest of ensuring the conceptual advance provided by the work, we recommend submitting a revision within 3 months (13th May 2026). Please discuss the revision progress ahead of this time with the editor if you require more time to complete the revisions. Use the link below to submit your revision:

Link Not Available

Referee #1:

Radhakrishnan and colleagues investigate how oligomerization of reconstituted Ska and Ndc80 complexes influences microtubule plus-end stability. Using *in vitro* reconstitution, cryo-electron tomography, and fluorescence-based assays, they confirm that Ska and Ndc80 act synergistically to stabilize dynamic microtubules and associate with microtubule plus ends in oligomeric assemblies. The authors further demonstrate that Ska complexes oligomerize independently on microtubules, and identify domains and residues required for this behavior, as well as link these properties to microtubule stability in cells. While the experiments provide support for a role of Ska and Ndc80 complexes in stabilizing microtubule plus ends, their conclusions do not convincingly support the proposed mechanism of protofilament bridging.

Major Point #1. An initial concern was the use of the envelope assay as a direct readout of oligomerization. The authors cite several studies using other microtubule-binding proteins to support this approach. In the revised manuscript, they further justify the assay by presenting cryoET analyses of WT Ska complexes and complexes containing the Ska1 R236A mutant (Figure 5). Oligomer size was quantified, and the authors conclude that oligomerization is reduced by 60-70% in the mutant. However, the overall binding of complexes on microtubules also appears reduced. Thus, it is difficult to conclude from these data that envelope size directly reflects oligomerization rather than reduced binding or occupancy. While envelope size may correlate with oligomerization, the assertion that envelope size is an equivalent measure of oligomerization should be tempered.

Major Point #2. The central issue here concerns the authors' conclusion that Ndc80/Ska complex oligomerization promotes microtubule end stability by physically tethering adjacent protofilaments. To address the issues raised, the authors performed several new experiments.

In Figure 3, microtubules were induced to depolymerize by tubulin dilution, and the authors report that the fraction of protofilament (PF) clusters increased in the presence of Ndc80 or Ndc80/Ska complexes compared to depolymerizing microtubules alone. However, this observation could also reflect Ndc80- or Ndc80/Ska-mediated stabilization of microtubules, which in turn promotes PF clustering, rather than PF clustering being the direct cause of stabilization.

In a second experiment, the authors examined microtubules in the presence of Ndc80 (3-10 nM) and compared microtubule end shapes when complexes were located within 50 nm of the microtubule end versus farther down the lattice. They found that the presence of Ndc80 complexes near the microtubule end correlated with increased PF clustering. Again, this correlation does not distinguish between PF clustering as a cause versus a consequence of Ndc80-mediated stabilization of plus ends

Finally, the authors performed cryoET of depolymerizing microtubules in the presence of 1 nM Ndc80C, a concentration they found to be non-stabilizing. Under these conditions, protofilaments appeared slightly more clustered than in depolymerizing naked microtubules. The authors interpret these results as evidence that PF clustering precedes and causes microtubule stabilization. However, given the modest magnitude of the effect, low n-values, and the statistical significance indicated on the

graph, the conclusion that Ndc80/Ndc80-Ska oligomers bridge and stabilize adjacent protofilaments is not convincing. Moreover, these data do not clearly support the proposed model that PF clustering results in stabilization, since at 1 nM Ndc80C, PF clustering is observed without detectable stabilization of plus ends, suggesting that these processes are not necessarily coupled.

I do not disfavor the hypothesis that Ndc80 or Ndc80-Ska oligomers bridge protofilaments to promote microtubule end stability; indeed, this is an interesting and plausible mechanism. However, the data presented here are not sufficient to support this conclusion.

While the manuscript reveals other interesting aspects of the Ska complex-such as the identification of a point mutant that perturbs Ska-Ska oligomerization and microtubule stability-I do not feel that these findings alone raise the overall impact of the study to a level appropriate for publication in EMBO Journal.

Minor points:

-The requirement for NDC80 trains to stabilize MTs requires >10 complexes. Is this relevant in terms of kinetochore biology? Or is this mechanism limited to in vitro microtubule dynamics?

-It is still not clear how chromosomes are perfectly aligned with no spindle or detection of microtubules in Figure 7.

-Related to the above point, I don't see any figures supporting the data indicating that a majority of cells (~60%) expressing the R236A mutant fail to properly align chromosomes.

Referee #2:

The authors have properly addressed all my previous concerns. I recommend the revised manuscript for publication in EMBO Journal.

Referee #3:

In this revised manuscript, the authors have extensively performed more experiments and analyses. The manuscript has been significantly improved, and the authors' conclusions are now more convincing. In the revised manuscript, this reviewer's main concerns were resolved. This reviewer recommend the manuscript to be published in the EMBO Journal. There are minor points to be fixed before its publication as below.

Minor points

-The authors added microtubule (MT) length measurement result in Fig. EV1D in response to our comments. They did mention that the MT length with Ska SKA3 T358/360A was shorter, but did not mention about the comparison of treated Ska and untreated Ska in the main text. If they were similar, this point should be mentioned as justification for mixing these two data of treated and untreated Ska samples.

-In revised Fig. 5A, authors added the white arrows in cryo-ET slices presumably in response to our comments. The white arrows are not explained in the figure legend. Please add texts explaining about the white arrows.

-In line 588, MgCl₂'s "2" is not in subscript.

-For the "cryoET" and "TIRF", these abbreviations should be defined when first mentioned. "cryoET" first appears on Line 128, and "TIRF" first appears on Line 204.

-At the last part of the figure legends of Fig. 2, the unit of scale bar is wrong. It should be 10 nm instead of 10 nM. Similarly, the last part of the figure legend of Fig. 3 is also written wrongly as 10 nM.

-In the figure legend of Fig.4, The part for E-F looks incomplete. The sentence "Example kymographs of LZ-LMTBD wt, LZ-LMTBD E236A, and FL SkaSKA1 R236A" is in bracket. Also, R236A is wrongly written as E236A here.

-As for Fig. EV2 figure legend, "(B) Subtomogram averaging of CH-domain trains (left), and non-Ndc80 microtubule decorations in the sample containing 1 nM Ndc80 and 10 nM Ska." This seems to be the legend of Fig. EV2E instead of Fig. EV2B.

-In the part "However, when we repeated the same subtomogram averaging approach, picking particles from non-Ndc80 microtubule decoration in the sample containing both Ndc80 and Ska (Figure 3B, Figure EV2B)", it seems that the figure numbering hasn't been fixed. Please correct it.

-In line 253, it should be "against shortening" rather than "against shortenin".

-As for the sentence "We found that microtubules plus-ends carrying an Ndc80 oligomer directly at the end had fewer single protofilaments and more clusters with {greater than or equal to}3 protofilaments.", it should refer to Fig. 3E at the end.

- "The R236A mutation also suppressed envelope formation in full-length SKA complexes, although we did observe SkaSKA1 R236A accumulation at the shortening microtubule ends (Figure 4F, Figure EV4DE)."
This seems to be (Figure 4F, Figure EV5DE). Please check.

- "We limited further analysis to measuring the fraction of residence times that were only observed for monomeric SKA1 MTBD and LMTBD constructs (short events, {less than or equal to}0.32 s), and the fraction of residence times that were never or very rarely observed for MTBD/LMTBD (long events, >0.32 s, Figure 5C)."
This should be Figure 6C. Please check.

- "Using this method, we observed a strong reduction of kinetochore-bound SKA3 following the SKA1 siRNA treatment (Figure EV7BC). Consistently, we observed an overall reduction in SKA3 levels by immunoblotting (Figure EV7BC)."
Authors might want to refer to Figure EV7B in the first sentence and Figure EV7C in the second sentence.

- "Cells transfected with WT GFP-SKA1 but with poor expression of the protein, as judged by the lack of the GFP signal at kinetochores, were also characterised by unstable kinetochore-microtubule attachments following the cold treatment (Figure EV6D)."

Perhaps authors wanted to refer to Figure EV7E here?

ARBITRATING ADVISOR:

My impression is that the remaining criticism of reviewer 1 is valid in terms of the logic of the argument. S/he does not seem to have concerns about the quality of the data, but about the interpretation and applies strict, but correct reasoning.

Major point 1 essentially asks for separation-of-function mutants that selectively either affect binding affinity to microtubules or oligomerization propensity. Currently the authors can't distinguish whether one or the other or both are affected. In their response letter the authors are often careful in distinguishing between correlation and interpretation/model. Maybe they could do that more explicitly also in the text, and discuss in the Discussion alternative possibilities/limitations of their preferred model, explaining why they prefer their interpretation/model of mechanism.

Major point 2 is a similar issue. There seems to be a clear correlation between Ndc80/Ska complex oligomerization and end stability and protofilament tethering. The authors' interpretation is then that tethering stabilizes microtubules, whereas the referee suggests that stabilization by another mechanism (which could perhaps be affecting microtubule lattice/protofilament structure in some other way) leads to protofilament tethering. Again, I would think that the authors can address this issue by separating clearly observation from interpretation and clearly stating why they favour a particular interpretation. They will say that their 1 nM Ndc80 experiment (Fig. 3E) shows that even under non-stabilizing conditions protofilament tethering happens, but the reviewer finds the effect mild and unconvincing. I would agree that it seems to be on the mild side, but as long as the authors correctly report the data and don't overstate their claim, I would say it is ok.

To me the cryo-ET data are the most original type of data here, and given the slightly controversial state of the literature reporting EM data of dynamic microtubule ends, I would think that what the authors show is within the usual range of what one would consider good quality.

So to me this study makes a valuable contribution combining some advanced ET and TIRF microscopy, probably pushing forward what's currently possible. I also thought that the authors made a good effort addressing the previous constructive criticism of the reviewers. Modifying language to clearly distinguish between observation and interpretation would seem to me a sufficient response to the remaining criticism of the reviewer. Which interpretation in the end will be confirmed - future work will have to show, but these are difficult proteins to work with, so I think the authors did a good job

Dr Vladimir Volkov
Reader in Biochemistry
Queen Mary University of London
School of Biological and Behavioural Science
Centre for Molecular Cell Biology
London E1 4NS
Email: v.volkov@qmul.ac.uk

23rd February 2026

Revision EMBOJ-2025-121896R2

Microtubule end stabilisation by cooperative oligomers of Ska and Ndc80 complexes

Dear Hartmut,

enclosed please find a modified manuscript that incorporates remaining issues raised by the reviewers, and addresses various editorial points identified by you and the editorial team. We sincerely hope that the current version is found suitable for publication at the EMBO Journal.

A point-by-point response to all comments by reviewers and the editorial team is included below (our responses added in blue font).

With best wishes,

Vladimir.

- Please make sure to re-upload our author checklist with information completed in all columns.

Completed author checklist is uploaded

- Please adjust the order of the manuscript sections, and also make sure to use the correct section headers:

Title page with complete author information, Abstract, Introduction, Results, Discussion, Methods, Data Availability, Acknowledgements, Disclosure and Competing Interests Statement, References, Main Figure Legends, Tables, Expanded Figure Legends.

Done

- Please double-check to make sure to all relevant funding information in the manuscript is congruent with the info entered into our submission system. Currently missing in the submission system is: BBSRC grant BB/W019698/1

Additional grant reference added

- Please rename the Conflict of Interest section into "Disclosure and Competing Interests Statement", in accordance with our updated Guide to Authors

(<https://www.embopress.org/competing-interests>).

Corrected

- As we are switching from a free-text author contribution statement towards a more formal statement based on Contributor Role Taxonomy (CRediT) terms, please remove the present Author Contribution section and instead specify each author's contribution(s) directly in the Author Information page of our submission system during upload of the final manuscript. See <https://casrai.org/credit/> for more information.

Author contributions removed from the manuscript and selected in the submission system

- In the Data Availability section, please include a direct URL to the respective database (EMDB) in which data generated for this study have been deposited.

URLs added

- Please carefully go through the reference list and make sure to stick to the journal format:

* For references with multiple authors, the first up to 10 authors should be listed, followed by 'et al.' after that

* DOIs should not be included, except for preprints, databases, or other items that do not have a formal citation (yet)

* some references are incomplete, missing e.g. citation year, volume, and/or page/locator numbers

* Please adjust the format for citation of real preprints: The citation in the text should be:

"(preprint: NAME1 et al, YEAR)"; and in the reference list: "NAME1, NAME2, ... (YEAR)

article title. bioRxiv doi: XXX [PREPRINT]"

Corrected

- Please move the Statement on AI use into the Methods section, instead of listing it as a separate section.

Corrected

- Please check the legend for Figure EV2, in which panel (E) seems to be missing (or is wrongly labelled as (B)?)

Thank you for flagging this mistake, corrected.

- Please rename all movies as Expanded View movies (in-text callouts "Movie EV1/2/..."). Their legends should be moved out of the text into individual text files, each of which should be combined with the respective movie file into a separate ZIP file and uploaded as such. Movies renamed and uploaded as ZIP files, each containing one movie, and a corresponding legend.

- Please provide suggestions for a short 'blurb' text prefacing and summing up the conceptual aspect of the study in two sentences (max. 250 characters), followed by 3-5 one-sentence 'bullet points' with brief factual statements of key results of the paper; they will form the basis of an editor-written 'Synopsis' accompanying the online version of the article. Please also upload a synopsis image, which can be used as a "visual title" for the synopsis section of your paper. The image (maybe based on Figure 8D with altered aspect ratio?) should be in JPG format, and please make sure that it remains in the modest dimensions of (exactly) 550 pixels wide and 300-600 pixels high.

Blurb suggestion:

Microtubule ends are captured by outer kinetochore proteins to ensure proper segregation of chromosomes. This work establishes oligomerising interactions of Ndc80 and Ska complexes that underlie their ability to stabilise microtubule ends against detachment and shortening.

- Microtubule ends stabilised against shortening by Ndc80, or by Ndc80 and Ska are characterised by increased clustering of bent protofilaments
- Ndc80 forms linear trains along two adjacent tubulin protofilaments
- Ska forms irregular oligomers, and helps to further oligomerise Ndc80
- Oligomerisation of Ska occurs separately via SKA3 tail and SKA1 microtubule-binding domain, the latter being necessary for cold-stability of kinetochore-bound microtubules

Summary figure (also uploaded in the manuscript submission system).

- Our routine image checks indicated that a tomography panel seems to appear twice (Fig 2E and Fig EV2D) - please clarify. In case of an intentional re-display, this needs to be explicitly indicated and rationalized in both respective figure legends.

Apologies for including this image twice – in the initially submitted version it served as a single legend for both conditions described in the panel (3 and 10 nM Ndc80). Since the panel was split between two figures, we have now included relevant example images from respective datasets. The image in Figure 2E remains the same; the image in Figure EV2D has been updated with an example of an Ndc80 oligomer at 3 nM.

- Finally, during editorial pre-acceptance checks, our data editors have raised the following queries regarding figures, data, and legends; I would appreciate if you briefly answered to them in the cover letter of your final submission, and made the requested text modifications with changes/additions highlighted via the "Track changes" option, to facilitate our final checking":

- 1- Please define the annotated p values ****/**/**/* as well as provide the exact p-values for the same in the legend of figure 2D, 4D, H; 5C, F; 6D, 7B, D; 8C, EV1 C, EV3 E, F; EV6 C, EV7 B as appropriate.
- 2- Please note that the exact p values are not provided in the legend of figure 1E,
- 3- Please indicate the statistical test used for data analysis in the leg ends of figures EV1 C, EV3 E, F; EV6 C
- 4- Please note that information related to n is missing in the legends of figures EV1 C, EV3 A, C, E, F; EV6 C
- 5- Please note that the error bars are not defined in the legends of figures 2D, 7B, EV1 C, D; EV3 C, EV6 C
- 6- Please note that the white arrows are not defined in the legend of figure 5A. This needs to be rectified.

All updated/corrected. The manuscript file is added with tracked changes as a "Related Manuscript File". However, the line numbering seems different with changes highlighted or accepted. The references to line numbers below are true for the MS version with all changes accepted. In addition, the submission system autoconverts docx files into pdf losing the tracked changes, so we have also uploaded a pdf of the file with tracked changes highlighted.

Referee #1:

Radhakrishnan and colleagues investigate how oligomerization of reconstituted Ska and Ndc80 complexes influences microtubule plus-end stability. Using in vitro reconstitution, cryo-electron tomography, and fluorescence-based assays, they confirm that Ska and Ndc80 act synergistically to stabilize dynamic microtubules and associate with microtubule plus ends in oligomeric assemblies. The authors further demonstrate that Ska complexes oligomerize independently on microtubules, and identify domains and residues required for this behavior, as well as link these properties to microtubule stability in cells. While the experiments provide support for a role of Ska and Ndc80 complexes in stabilizing microtubule plus ends, their conclusions do not convincingly support the proposed mechanism of protofilament bridging.

Major Point #1. An initial concern was the use of the envelope assay as a direct readout of oligomerization. The authors cite several studies using other microtubule-binding proteins to support this approach. In the revised manuscript, they further justify the assay by presenting

cryoET analyses of WT Ska complexes and complexes containing the Ska1 R236A mutant (Figure 5). Oligomer size was quantified, and the authors conclude that oligomerization is reduced by 60-70% in the mutant. However, the overall binding of complexes on microtubules also appears reduced. Thus, it is difficult to conclude from these data that envelope size directly reflects oligomerization rather than reduced binding or occupancy. While envelope size may correlate with oligomerization, the assertion that envelope size is an equivalent measure of oligomerization should be tempered.

We thank the reviewer for their critical comments, which continue to improve the quality of our work. We would like to note that the R236A mutation in SKA1 did not affect residence time of individual molecules of either the dimeric SKA1 fragments, or the full-length Ska123 complex – these data are based on TIRF microscopy and shown in Figures 6 and EV6. We are not aware of methods that can derive binding affinity from cryoET data. We therefore believe that a combination of cryoET to estimate oligomerisation, and single-molecule TIRF microscopy to estimate affinity is a powerful way to assess the properties of microtubule-binding proteins with oligomerisation capacity. Of note, this combination is quite novel due to recent technological advances in cryoET, and thus its description in this study may already present a valuable addition to the literature – previous studies we cite in support of our interpretations relied only on fluorescence microscopy, or only on negative stain EM. Based on this combination of complementary structural and dynamics assays we conclude that the R236A mutation impacts oligomerisation of Ska without directly affecting individual molecules' affinity to microtubules, which apparently remains unchanged thanks to other K/R residues in the SKA1 MTBD binding to tubulin. We added various edits in the text to reinforce that this is our interpretation (such as on lines 362-363), and to clarify this interpretation in the Discussion section, lines 492-495.

Major Point #2. The central issue here concerns the authors' conclusion that Ndc80/Ska complex oligomerization promotes microtubule end stability by physically tethering adjacent protofilaments. To address the issues raised, the authors performed several new experiments.

In Figure 3, microtubules were induced to depolymerize by tubulin dilution, and the authors report that the fraction of protofilament (PF) clusters increased in the presence of Ndc80 or Ndc80/Ska complexes compared to depolymerizing microtubules alone. However, this observation could also reflect Ndc80- or Ndc80/Ska-mediated stabilization of microtubules, which in turn promotes PF clustering, rather than PF clustering being the direct cause of stabilization.

In a second experiment, the authors examined microtubules in the presence of Ndc80 (3-10 nM) and compared microtubule end shapes when complexes were located within 50 nm of the microtubule end versus farther down the lattice. They found that the presence of Ndc80 complexes near the microtubule end correlated with increased PF clustering. Again, this correlation does not distinguish between PF clustering as a cause versus a consequence of Ndc80-mediated stabilization of plus ends

Finally, the authors performed cryoET of depolymerizing microtubules in the presence of 1 nM Ndc80C, a concentration they found to be non-stabilizing. Under these conditions, protofilaments appeared slightly more clustered than in depolymerizing naked microtubules.

The authors interpret these results as evidence that PF clustering precedes and causes microtubule stabilization. However, given the modest magnitude of the effect, low n-values, and the statistical significance indicated on the graph, the conclusion that Ndc80/Ndc80-Ska oligomers bridge and stabilize adjacent protofilaments is not convincing. Moreover, these data do not clearly support the proposed model that PF clustering results in stabilization, since at 1 nM Ndc80C, PF clustering is observed without detectable stabilization of plus ends, suggesting that these processes are not necessarily coupled.

I do not disfavor the hypothesis that Ndc80 or Ndc80-Ska oligomers bridge protofilaments to promote microtubule end stability; indeed, this is an interesting and plausible mechanism. However, the data presented here are not sufficient to support this conclusion.

We thank the reviewer for their critical assessment of our conclusions. Indeed, we can not completely rule out the presence of other, currently unknown, microtubule end-stabilising properties of Ndc80 and Ndc80:Ska oligomers beyond their effect on protofilament clustering. To clarify our assumptions and interpretations, as well as to point out potential existence of these other unknown microtubule end-stabilisation mechanisms, we added a paragraph in the Discussion section on lines 456-476, which substitutes a more strongly worded section of the Discussion which we have removed.

While the manuscript reveals other interesting aspects of the Ska complex-such as the identification of a point mutant that perturbs Ska-Ska oligomerization and microtubule stability-I do not feel that these findings alone raise the overall impact of the study to a level appropriate for publication in EMBO Journal.

Minor points:

-The requirement for NDC80 trains to stabilize MTs requires >10 complexes. Is this relevant in terms of kinetochore biology? Or is this mechanism limited to in vitro microtubule dynamics?

We believe these numbers to be highly relevant and consistent with the currently available estimates of ca. 250 copies of the Ndc80 complex interacting with ca. 10-20 microtubule ends at a human kinetochore. We added references to relevant studies on lines 207-208.

-It is still not clear how chromosomes are perfectly aligned with no spindle or detection of microtubules in Figure 7.

We thank the reviewer for bringing the attention to this point, which has not been raised during the previous revision. We do find it surprising that the metaphase plate of a cell retains its overall alignment after a brief cold treatment that depolymerises majority of the microtubules. We have observed this outcome repeatedly in our experiments, and we report additional images in the Appendix figure S1B. Overall, the mild chromosome alignment phenotype in presence of the SKA1 R236A mutant is consistent with our quantifications reported in Figure 7B, and with earlier literature (Abad et al., 2014; Monda et al., 2017).

-Related to the above point, I don't see any figures supporting the data indicating that a majority of cells (~60%) expressing the R236A mutant fail to properly align chromosomes. As we report in Figure 7B, ≥50% (range of 54-100% in various repeats) of cells expressing the R236A indeed had correct chromosome alignment, consistent with earlier reports (Abad et al., 2014; Monda et al., 2017). Examples of cells we considered as having aligned or

misaligned chromosomes are included in Figure EV7E. We are including additional examples of cells expressing GFP-SKA1 R236A with misaligned chromosomes in the Appendix Figure S1C.

Referee #2:

The authors have properly addressed all my previous concerns. I recommend the revised manuscript for publication in EMBO Journal.

We thank the reviewer for the positive evaluation of our revised manuscript.

Referee #3:

In this revised manuscript, the authors have extensively performed more experiments and analyses. The manuscript has been significantly improved, and the authors' conclusions are now more convincing. In the revised manuscript, this reviewer's main concerns were resolved. This reviewer recommend the manuscript to be published in the EMBO Journal. There are minor points to be fixed before its publication as below.

We thank the reviewer for a positive evaluation of our study, and for their careful reading of our manuscript. All minor points below have been addressed in the revised manuscript.

Minor points

-The authors added microtubule (MT) length measurement result in Fig. EV1D in response to our comments. They did mention that the MT length with Ska SKA3 T358/360A was shorter, but did not mention about the comparison of treated Ska and untreated Ska in the main text. If they were similar, this point should be mentioned as justification for mixing these two data of treated and untreated Ska samples.

Comparison of microtubule lengths between untreated and hyperphosphorylated Ska, as shown in Figure EV1D was added in lines 173-175.

-In revised Fig. 5A, authors added the white arrows in cryo-ET slices presumably in response to our comments. The white arrows are not explained in the figure legend. Please add texts explaining about the white arrows.

Figure 5A legend is now updated.

-In line 588, MgCl₂'s "2" is not in subscript.

Corrected

-For the "cryoET" and "TIRF", these abbreviations should be defined when first mentioned. "cryoET" first appears on Line 128, and "TIRF" first appears on Line 204.

Abbreviations are now spelled out on first mention: cryoET on line 128, TIRF on line 145.

-At the last part of the figure legends of Fig. 2, the unit of scale bar is wrong. It should be 10 nm instead of 10 nM. Similarly, the last part of the figure legend of Fig. 3 is also written wrongly as 10 nM.

Corrected.

-In the figure legend of Fig.4, The part for E-F looks incomplete. The sentence "Example

kymographs of LZ-LMTBD wt, LZ-LMTBD E236A, and FL SkaSKA1 R236A" is in bracket. Also, R236A is wrongly written as E236A here.

Corrected.

-As for Fig. EV2 figure legend, "(B) Subtomogram averaging of CH-domain trains (left), and non-Ndc80 microtubule decorations in the sample containing 1 nM Ndc80 and 10 nM Ska." This seems to be the legend of Fig. EV2E instead of Fig. EV2B.

Corrected.

-In the part "However, when we repeated the same subtomogram averaging approach, picking particles from non-Ndc80 microtubule decoration in the sample containing both Ndc80 and Ska (Figure 3B, Figure EV2B)", it seems that the figure numbering hasn't been fixed. Please correct it.

Corrected.

-In line 253, it should be "against shortening" rather than "against shortenin".

Corrected.

-As for the sentence "We found that microtubules plus-ends carrying an Ndc80 oligomer directly at the end had fewer single protofilaments and more clusters with {greater than or equal to}3 protofilaments.", it should refer to Fig. 3E at the end.

Reference to Figure 3E added.

- "The R236A mutation also suppressed envelope formation in full-length SKA complexes, although we did observe SkaSKA1 R236A accumulation at the shortening microtubule ends (Figure 4F, Figure EV4DE)."

This seems to be (Figure 4F, Figure EV5DE). Please check.

Corrected

- "We limited further analysis to measuring the fraction of residence times that were only observed for monomeric SKA1 MTBD and LMTBD constructs (short events, {less than or equal to}0.32 s), and the fraction of residence times that were never or very rarely observed for MTBD/LMTBD (long events, >0.32 s, Figure 5C)."

This should be Figure 6C. Please check.

Corrected.

- "Using this method, we observed a strong reduction of kinetochore-bound SKA3 following the SKA1 siRNA treatment (Figure EV7BC). Consistently, we observed an overall reduction in SKA3 levels by immunoblotting (Figure EV7BC)."

Authors might want to refer to Figure EV7B in the first sentence and Figure EV7C in the second sentence.

Corrected.

- "Cells transfected with WT GFP-SKA1 but with poor expression of the protein, as judged by the lack of the GFP signal at kinetochores, were also characterised by unstable kinetochore-microtubule attachments following the cold treatment (Figure EV6D)."

Perhaps authors wanted to refer to Figure EV7E here?

We thank the reviewer for spotting this mistake – the figure panel we refer to here has been

inadvertently removed during revision (it was present as Supplementary Figure 6E in the original submission). We added this panel back as Appendix Figure S1A.

ARBITRATING ADVISOR:

My impression is that the remaining criticism of reviewer 1 is valid in terms of the logic of the argument. S/he does not seem to have concerns about the quality of the data, but about the interpretation and applies strict, but correct reasoning.

Major point 1 essentially asks for separation-of-function mutants that selectively either affect binding affinity to microtubules or oligomerization propensity. Currently the authors can't distinguish whether one or the other or both are affected. In their response letter the authors are often careful in distinguishing between correlation and interpretation/model. Maybe they could do that more explicitly also in the text, and discuss in the Discussion alternative possibilities/limitations of their preferred model, explaining why they prefer their interpretation/model of mechanism.

We thank the reviewer for their balanced and impartial assessment of our conclusions and interpretations. Addressing point 1, we have modified the manuscript text to clarify that we interpret SKA1 R236A as the separation-of-function mutant: it affects the oligomerisation both of the Ska complex, and of the dimeric SKA1 MTBD constructs, without a direct effect on their microtubule binding in single molecule conditions. Please also see our more detailed response to reviewer 1 above.

Major point 2 is a similar issue. There seems to be a clear correlation between Ndc80/Ska complex oligomerization and end stability and protofilament tethering. The authors' interpretation is then that tethering stabilizes microtubules, whereas the referee suggests that stabilization by another mechanism (which could perhaps be affecting microtubule lattice/protofilament structure in some other way) leads to protofilament tethering. Again, I would think that the authors can address this issue by separating clearly observation from interpretation and clearly stating why they favour a particular interpretation. They will say that their 1nM Ndc80 experiment (Fig. 3E) shows that even under non-stabilizing conditions protofilament tethering happens, but the reviewer finds the effect mild and unconvincing. I would agree that it seems to be on the mild side, but as long as the authors correctly report the data and don't overstate their claim, I would say it is ok.

To me the cryo-ET data are the most original type of data here, and given the slightly controversial state of the literature reporting EM data of dynamic microtubule ends, I would think that what the authors show is within the usual range of what one would consider good quality.

So to me this study makes a valuable contribution combining some advanced ET and TIRF microscopy, probably pushing forward what's currently possible. I also thought that the authors made a good effort addressing the previous constructive criticism of the reviewers. Modifying language to clearly distinguish between observation and interpretation would seem to me a sufficient response to the remaining criticism of the reviewer. Which interpretation in

the end will be confirmed - future work will have to show, but these are difficult proteins to work with, so I think the authors did a good job

We thank the reviewer for appreciating our efforts to address the constructive and valuable criticism of our previous submission. We have addressed the point 2 raised by reviewer 1, and discussed here, by toning down the strength of our assertions in various places in the Results section, as well as by including a more balanced description of our assumptions and interpretations in the Discussion. Please also see our detailed response to reviewer 1 above.

Dr. Vladimir A. Volkov
Queen Mary University of London
Centre for Molecular Cell Biology, School of Biological and Behavioural Sciences
Mile End Rd
E1 4NS
United Kingdom

27th Feb 2026

Re: EMBOJ-2025-121896R2
Microtubule end stabilisation by cooperative oligomers of Ska and Ndc80 complexes

Dear Vladimir,

Thank you for submitting your final revised manuscript for our consideration. I am pleased to inform you that we have now accepted it for publication in The EMBO Journal.

You may qualify for financial assistance for your publication charges - either via a Springer Nature fully open access agreement or an EMBO initiative. Check your eligibility: <https://link.springer.com/journal/44318/how-to-publish-with-us>

With kind regards,

Hartmut

Please note that it is The EMBO Journal policy for the transcript of the editorial process (containing referee reports and your response letters) to be published as an online supplement to each paper. If you should prefer removal of any referee-only figures included in the point-by-point response(s), e.g. because they may still be used for future publication or because they have been reproduced from published work by others, please do let us know immediately via response email.

More information is available here: <https://link.springer.com/partners/embo-press/editorial-policies#Peer%20review>